# A Novel Pathway of Atmospheric Sulfate Formation Through Carbonate Radical

Yangyang Liu[1,2], Yue Deng[1,2], Jiarong Liu[3], Xiaozhong Fang[1], Tao Wang[1], Kejian Li[1], Kedong Gong[1], Aziz U. Bacha[1], Iqra Nabi[1], Qiuyue Ge[1],  Xiuhui Zhang[3], Christian George[4], and Liwu Zhang[1,2]

[1]Shanghai Key Laboratory of Atmospheric Particle Pollution and Prevention, Department of Environmental Science and Engineering, Fudan University, Shanghai, 200433, P. R. China.
[2]Shanghai Institute of Pollution Control and Ecological Security, Shanghai, 200092, Peoples' Republic of China.
[3]Key Laboratory of Cluster Science, Ministry of Education of China, School of Chemistry and Chemical Engineering, Beijing Institute of Technology, Beijing 100081, P. R. China
[4]Univ. Lyon, Université Claude Bernard Lyon 1, CNRS, IRCELYON, F-69626, Villeurbanne, France.

*Correspondence to*: Liwu Zhang (zhanglw@fudan.edu.cn)

**Abstract.** Carbon dioxide is considered an inert gas that rarely participates in atmospheric chemical reactions. Nonetheless, we show here that $CO_2$ is involved in some important photo-oxidation reactions in the atmosphere through the formation of carbonate radicals ($CO_3^{.-}$). This potentially active intermediate $CO_3^{.-}$ is routinely overlooked in atmospheric chemistry concerning its effect on sulfate formation. Present work demonstrates that $SO_2$ uptake coefficient is enhanced by 17 times on mineral dust particles driven by $CO_3^{.-}$. Importantly, upon irradiation mineral dust particles are speculated to produce gas-phase carbonate radical ions when the atmospherically relevant concentration of $CO_2$ presents, thereby potentially promoting external sulfate aerosol formation and oxidative potential in the atmosphere. Employing a suite of laboratory investigations of sulfate formation in the presence of carbonate radical on the model and authentic dust particles, ground-based field measurements of sulfate and (bi)carbonate ions within ambient PM, together with density functional theory (DFT) calculations for single electron transfer processes in terms of $CO_3^{.-}$-initiated S(IV) oxidation, a novel role of carbonate radical in atmospheric chemistry is elucidated.

## 1. Introduction

Atmospheric composition changes are subjected to highly reactive light-induced radicals, such as hydroxyl ($\cdot OH$), hydroperoxyl ($HO_2\cdot$), or nitrate radicals ($NO_3\cdot$), which are able to alter not only compositions but also physical and chemical properties of particulate matter (Liu et al., 2022a; Mahajan et al., 2021; Stevenson et al., 2020). However, when atmospheric chemical reactions occur over humified particles at ambient conditions, which creates a locally enriched aqueous medium of unique chemical activity, other radicals might likewise gain importance. The carbonate radical ($CO_3^{.-}$) is typically such an active radical. The lifetime of $CO_3^{.-}$ ranges from a microsecond to even a few milliseconds and its concentration can be two

orders of magnitude higher than that of hydroxyl radicals over the water surface (Chandrasekaran and Thomas, 1983; Goldstein et al., 2001; Shafirovich et al., 2001; Sulzberger et al., 1997). In addition, the one-electron reduction potential of $E^0$($CO_3^-$/$CO_3^{2-}$) couple is 1.78 V vs. NHE at neutral pH, leaving $CO_3^-$ a strong oxidant in aquatic chemistry (Bisby et al., 1998; Cope
et al., 1973 ; Merouani et al., 2010). Previous studies concerning carbonate radical in aqueous media demonstrate that it reacts rapidly with some organic compounds (Merouani et al., 2010), especially for those electron-rich compounds amines (Stenman et al., 2003; Yan et al., 2019). Also, it has been pointed out that the scavenging of hydroxyl radicals by (bi)carbonate species leads to the formation of $CO_3^-$ ions (Graedel and Weschler, 1981; Xiong et al., 2016). Besides, a high second order of rate constant, lying at $10^9$ $M^{-1}$ $s^{-1}$, has been reported for the reaction of $CO_3^-$ with porphyrins (Ferrer-Sueta et al., 2003), indicating
that this radical ion has great oxidation capability that may trigger atmospherically relevant chemical reactions, i.e. secondary inorganic species formation. However, it is only regarded as a marginal intermediate in tropospheric anion chemistry so far (Beig and Brasseur, 2000; Ge et al., 2021; Herrmann et al., 2000; Lehtipalo et al., 2016) and its underlying role as an active oxidant for heterogeneous reaction in the atmosphere is barely explored. Very recently, our group observed the promotional effect of $CO_3^-$ on atmospheric nitrate formation (Fang et al., 2021). Motivated by this finding, attempts were made to further
explore its role in other important atmospherically-relevant reactions.

It is well documented that sulfate ($SO_4^{2-}$) is also a key constituent of aerosols in the atmosphere (Huang et al., 2015; Su et al., 2016). It is able to serve as the precursors of efficient cloud condensation nuclei, with optical properties leading to a cooling effect (Wang et al., 2011). As a consequence, the mechanism aspect of secondary sulfate formation was the focus of numerous studies over the past decades (Hung et al., 2018; Stone, 2002; Zhang et al., 2015b). There is a consensus that high-valence
sulfur (VI), produced from the oxidation of anthropogenic $SO_2$, is the dominant source of atmospheric secondary sulfate. However, a remarkable missing sulfate budget emerges for the atmospheric modeling (Huang et al., 2019; Itahashi et al., 2018; Liu et al., 2021), which significantly underpredicts $SO_4^{2-}$ with respect to observational results when heterogeneous aerosol chemistry is not considered (Feng et al., 2018; Wu et al., 2021; Zheng et al., 2015). This indicates that the heterogeneous sulfate production pathway is a crucial process and exploring the unrecognized heterogeneous mechanism is very likely to
narrow the gap between observations in lab studies, field measurements, and numerical modelings. Due to the missing chemical mechanism that initiated fast $SO_2$ oxidation, atmospheric models fail to capture the key feature of atmospheric observations of high sulfate production during dust storm episodes in the troposphere (Dong et al., 2016; Huang et al., 2014; Yu et al., 2020), where an evident increase of $Ca^{2+}$ (Li et al., 2013; Wang et al., 2005), carbonate-containing particles with high alkalinity (Abou-Ghanem et al., 2020; Li et al., 2014; Tang et al., 2016) as well as photoactive mineral components (Nie et al., 2012; Ta
et al., 2003) are prevalent. Air mass is usually in low relative humidity, reported being 25-35% (Al-Salihi and Mohammed, 2015) (Csavina et al., 2014; Najafpour et al., 2020) in these events, during which the photochemical process is able to alter atmospheric constituents (Liu et al., 2022b). Consequently, there are unknown heterogeneous reaction pathways and previously unconsidered promoters that have great potential to accelerate sulfate formation in the dust storm relevant conditions.

Due to the high stability of $CO_2$ under ambient conditions (Hossain et al., 2020), there are rare studies concerning the
influence of $CO_2$ in atmospheric chemical processes (Deng et al., 2020; Liu et al., 2020c; Xia et al., 2021). $CO_2$ is demonstrated

to form (bi)carbonate species over humidified dust particles (Baltrusaitis et al., 2011; Nanayakkara et al., 2014) and reduced to CO under solar illumination (Deng et al., 2020). Nonetheless, its impact on atmospheric heterogeneous reactions remains poorly characterized. Our early laboratory study illustrates that $CO_2$ decreases the sulfate formation on aluminum oxide particles in the dark (Liu et al., 2020c) while upon solar illumination its role in $SO_2$ oxidation over mineral dust surfaces is still an open question. In addition, carbonate salt is abundant in authentic dust particles (Cao et al., 2005) and is reported to reach over 10% wt. of Asian dust particles (McNaughton et al., 2009). It is generally accepted that $CO_3^{2-}$ affects atmospheric chemistry and aerosol characteristics mainly through its intrinsic alkalinity, which buffers aerosol acidity and increases $SO_2$ adsorption and corresponding sulfate production in the presence of oxidants (Al-Hosney and Grassian, 2005; Bao et al., 2010; Kerminen et al., 2001; Li et al., 2007; Yu et al., 2018). In fact, either $CO_2$ or carbonate salt is able to produce the active $CO_3^{-}$ under the ambient circumstance (Ervens et al., 2003; Graedel and Weschler, 1981) and prone to increase the oxidative capacity in the atmosphere. Our early study shows that carbonate radicals serve as an active oxidant to accelerate $NO_2$ oxidation over mineral dust particles (Fang et al., 2021), allowing us to consider the possibility that fast heterogeneous $SO_2$ oxidation can be triggered by this active intermediate as well. Nevertheless, to the best of our knowledge, no work has ever considered how and to what extent the carbonate radical influences $SO_2$ heterogeneous oxidation in the atmosphere.

In the current study, through laboratory studies, we present that carbon dioxide and calcium carbonate, working as the precursor of carbonate radicals, have the ability to accelerate sulfate formation over authentic particles in the atmosphere. Together with quantum chemistry calculations, a detailed molecular mechanism regarding a single electron transfer (SET) process between carbonate radical and sulfite ions is elucidated. Furthermore, ground-based observations validate some findings from the laboratory-based simulations.

## 2. Experimental methods

### 2.1 Laboratory Studies

**Methodology for uptake coefficient estimation.** The reaction uptake coefficient was estimated by the following Eqs. **1**-**3**, as suggested by the previous work (Kong et al., 2014):

$$\gamma = \frac{d[SO_4^{2-}]/dt}{Z} \tag{1}$$

$$Z = \frac{1}{4} \times A_s \times [SO_2] \times v \tag{2}$$

$$v = \sqrt{8RT/\pi M_{SO2}} \tag{3}$$

where $v$ is the mean molecular velocity of $SO_2$, $A_s$ the effective sample surface, R the gas constant, T the temperature, $M_{SO2}$ the molecular weight of $SO_2$, and a total number of surface collisions per unit time ($Z$). To be precise, the formation rates

($d[SO_4^{2-}]/dt$) in the equation were determined by ion chromatography (IC) measurements, followed by a conversion factor calculation through linear regression analysis for the integrated absorbance of sulfate bands and corresponding sulfate concentrations. By employing this method, a conversion factor of $f = 6.34\times10^{15}$ ions integrated absorption units$^{-1}$ was obtained and corresponding $SO_2$ uptake coefficients in the "$TiO_2$+($CO_2$)+$SO_2$" system were thus estimated using seven timepoints during the heterogeneous reaction. For $SO_2$ uptake in the "$TiO_2$+($CaCO_3$)+$SO_2$" system, we estimated $SO_2$ uptake coefficients using three conversion curves established for various types of dust particles. For this purpose, we mixed the known proportion of $K_2SO_4$ and dust particles of concern, and thus obtained relationship curves between the integrated absorbance of DRIFTS (i.e. diffuse reflectance Fourier transformed infrared spectroscopy) sulfate bands and corresponding theoretical sulfate content through linear fitting (Fig. S4). Ten timepoints during the heterogeneous reaction were applied for these kinetics calculations.

**Preparation for clay and dust membranes and investigation of sulfate formation on those authentic particles during the daytime and nighttime.** Each particle suspension ATD (2.5 mg~0.5 mL), IMt-2 (10 mg~ 0.5 mL) and K-GA (10 mg~ 0.5 mL) were first dispersed into water through ultrasonic bath for 5 min. After that, sample suspensions were transformed onto the cleansed round quartz films (d = 2 cm) using a pipette and subsequently sent to the infrared drying oven for 10 min to prepare dust membranes. Once taken out from the oven, samples were quickly sealed into a desiccator and cooled down to room temperature before starting experiments. A membrane sample was then mounted at the center of the reaction chamber (the top half made of quartz and the bottom half made of teflon). Before each set of experiments, a gas flow (dry air) of 300 mL min$^{-1}$ was introduced to the chamber for 5 min where a prepared membrane sample was installed. Afterward, samples were exposed to $4.91\times10^{14}$ molecule cm$^{-3}$ $SO_2$ (+$2.46\times10^{18}$ $CO_2$ when necessary) /$N_2$+$O_2$ mixture in the absence and presence of irradiation (Light intensity (I) = 30 mW cm$^{-2}$) for 15 min before the sample were transferred to a beaker (scale = 10 mL) with 2 mL of 2% vol. isopropanol leaching solution, extracting ions in the ultrasonic tank for 5 min using 0.22 µm PTFE membrane filter, followed by sending the sample into IC. Noting that dust and clay particles possess a considerable amount of sulfate ion background, we thus measured the background ions for each batch of synthesized dust particle membranes following the procedures described above. All data demonstrated in Fig. 6 were obtained after subtraction of background ions.

**Determination of gas-phase reactive oxygen species (ROS) production in the flow-cell reactor.** To measure the concentration of ROS released from $TiO_2$ particles in various reaction systems, an experimental approach using the probe molecule aniline was applied in this study. This is because compound aniline is reported to react rapidly with $\cdot OH$ radicals and $CO_3^{\cdot-}$ radicals, which are also evidenced to be two major active ROS species that are responsible for the $SO_2$ oxidation over mineral dust particles. The method applied in this study was almost implemented as the same to that of the previous study (Behrman, 2018), with slight modification. Briefly, the degradation rate of aniline in various reaction systems were monitored through High-Performance Liquid Chromatography (HPLC, LC-10AD, SHIMADZU, Japan). A Zorbax SB C18 (4.6 mm × 150 mm, 5 µm) reverse phase column at 25 ℃ was used with a UV detector at 236 nm to measure the aniline concentration. The mobile phase consisted of acetonitrile/water = 55:45 (V/V) with a flow rate of 1 mL/min.

TiO$_2$ suspension (5 mg TiO$_2$ per 100 uL deionized water) was deposited onto the glass substrate (0.13 - 0.17 mm in thickness) using a pipette and then dried in an oven for 10 min to obtain a TiO$_2$-coated film. Dilute aniline solution, using 67 mM phosphate buffer solution (pH = 7.0), was prepared and placed below the TiO$_2$-coated film, with an intervening gap between TiO$_2$ film and solution surface around 2 mm (Fig. 1). This short distance essentially guarantees gaseous ROS (e.g. ·OH radicals or CO$_3$·$^-$ radicals) to diffuse and react with aniline molecule (Rodriguez et al., 2013).

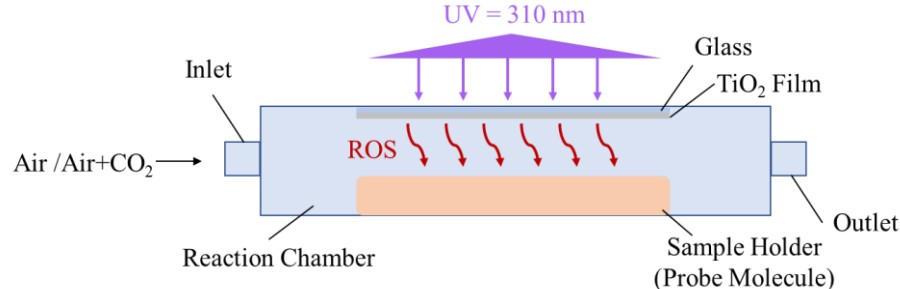

**Figure 1.** The schematic of the flow-cell reaction chamber for gaseous ROS determination.

In the reaction system containing TiO$_2$ film upon irradiation (the UV wavelength = 310 nm) in the presence of humidified air (RH = 30%), when operated in a continuous mode, the overall degradation rate of probe molecules in the presence of TiO$_2$ film can be described by Eq. **4** (Wang et al., 2004):

$$k_{obs} = \frac{d[An]}{dt} = r_A + r_U + r_{ROS} = r_A + r_U + k_{ROS, AN}[ROS][An] \tag{4}$$

Where $k_{obs}$ is the observed degradation rate of aniline, [An] is the concentration of aniline, denoted as [An] hereafter, and $r_A$, $r_U$, $r_{ROS}$ stand for aniline removal rates resulting from air stripping, UV photolysis, ROS oxidation. $k_{ROS, An}$ are the overall second-order reaction rate constants for An with ROS.

Reference experiments without the introduction of ROS were also conducted to measure $r_A + r_U$ in each reaction system. Upon irradiation, the dust proxy TiO$_2$ produces hole-electron pairs, further forming ·OH radicals and superoxide radicals (O$_2$·$^-$) in the presence of absorbed water and oxygen molecules. Thus, an experiment using N$_2$ was adopted to investigate the role of O$_2$·$^-$ in consuming aniline. As illustrated in Fig. 7c, a slight change in the degradation rate of aniline after stripping oxygen from the air, indicating that O$_2$·$^-$ shows quite a smaller contribution than ·OH. This result agrees well with the finding reported by Durán et al. (Duran et al., 2019), where the removal of O$_2$·$^-$ by adding benzoquinone (BQ) into TiO$_2$ suspension results in the negligible change of An degradation rate.

Taken above, ·OH radicals are assumed to be the only active ROS that accounts for the An degradation. Hence, the maximum steady concentration of ·OH radicals can be given by the following equation:

$$-\frac{d[An]}{dt} = k_{exp}[An] = k_{·OH, An}[·OH]_{ss-max}[An] \tag{5}$$

Integration of Eq. **5** yields

$$-\ln\frac{[An]_t}{[An]_0}=k_{exp}t \tag{6}$$

$$k_{exp}=k_{\cdot OH,An}[\cdot OH]_{ss\text{-}max} \tag{7}$$

Together with the reported second-order rate constant ($k_{\cdot OH, An} = 6.5\times10^9$ M$^{-1}$s$^{-1}$) (Samuni et al., 2002), the steady-state OH radical concentration $[\cdot OH]_{ss\text{-}max}$ in buffered An solution can be calculated from Eq. **7**. The observed degradation rate constant of $k_{exp}$ can be obtained from the slope of the semi-log plot of An degradation as illustrated in Eq. **6**. The maximum steady-state aqueous concentration of $\cdot OH$ supplied by the partitioning process from gas-phase $\cdot OH$ was thus estimated to be $2.15\times10^{-15}$ M for the "TiO$_2$+Air" system.

When CO$_2$ (400 ppm, atmospheric relevant concentration) is introduced into a flow-cell chamber, an increased degradation rate of An is seen, which is very likely to be the generation of active carbonate radical ions (Fig. 9). Similar to the method we adopted for the estimation of $[\cdot OH]_{ss\text{-}max}$, reference experiments were conducted to determine the rates for air stripping and UV photolysis processes in the "TiO$_2$+Air+CO$_2$" system. In the next step, we quenched the hydroxyl radicals by adding tertiary butanol (TBA). This is because it reacts rapidly with hydroxyl radicals (Li et al., 2020) $k_{\cdot OH,TBA}= (6\times10^8$ M$^{-1}$ s$^{-1}$) while shows a rather low reaction rate with carbonate radicals (Liu et al., 2015) ($k_{CO_3^{\cdot-},TBA}<1.6\times10^2$ M$^{-1}$ s$^{-1}$). Subsequently, we determined $[CO_3^{\cdot-}]_{ss}$ using the previous protocol (Huang and Mabury, 2000) with known $k_{CO_3^{\cdot-},AN}$ ($5.4\times10^8$ M$^{-1}$ s$^{-1}$) (Wojnarovits et al., 2020). In the extreme case, assuming that all hydroxyl radical ions were fully trapped by absorbed and dissolved HCO$_3^-$/CO$_3^{2-}$, the maximum steady-state CO$_3^{\cdot-}$ concentration was determined to be $1.39\times10^{-13}$ M for "TiO$_2$+Air+CO$_2$" system, matching well with the early study where the concentration of CO$_3^{\cdot-}$ is reported to be two orders of magnitudes than $\cdot OH$ over the water surface (Sulzberger et al., 1997).

## 2.2 Quantum Chemical Calculation

We employed density functional theory (DFT) calculations in the term of the single electron transfer (SET) process using Gaussian 09 package to investigate this novel route, detailed in Supplementary text 11 and 20.

## 2.3 Field Observations

**Sampling.** Our sampling for atmospheric particle matter was launched on the roof of the environmental science and engineering department, Fudan University (Jiangwan Campus, 31.340661°N, 121.506747°E, 16 km away from the city downtown. More geographical information for sampling has been described in detail elsewhere (Liu et al., 2020c). Observations for water-soluble ionic components of particulate matter were performed using an 8-stage non-viable-cascade-impactor type sampler (TISCH TE Inc., USA), size gradient of which are in the sequence of < 0.43, 0.43- 0.65, 0.65-1.1, 1.1-2.1, 2.1- 3.3, 3.3- 4.7, 4.7- 5.8 5.8- 9.0 and 9.0~ μm. These sizes represent the effective cutoff diameter at each level for unit density spherical particles. In our sampling, the estimations for (bi)carbonate ions in atmospheric particulate matter is conducted for the initial four stages (3.3~ μm). Atmospheric airflow from the head was maintained at the constant rate of 28.3

L min$^{-1}$ to meet the operation criterion required for the Anderson-type sampler. Quartz filters (81 mm in diameter, Whatman, GE Healthcare, UK) were applied for samplings, and membranes were rinsed with ultrapure water (electrical resistivity= 18.2 MΩ) no less than three times; that is kept in the ultrasonic cleaning tank for 40 min and then rinsed with ultrapure water for twice, before being sent into the infrared drying oven, followed by packing them in aluminum foils prior to field sampling. We carried out this procedure to eliminate the water-soluble background ions as much as possible and to ensure the balance calculation for (bi)carbonate ions. Additionally, following the same aforementioned pretreatments, we measured the background concentrations of ions in blank membranes. We separated our daily samplings into two periods, 11 hours for each, to give an insight into the influence of potential photo-induced reactions on secondary sulfate formation in the atmosphere.

**(Bi)carbonate estimation.** The concentration of bi(carbonate) ions were estimated following the protocol reported in the early works (Fang et al., 2021; Liu et al., 2020c; Palmer and Cherry, 1984; Zhang et al., 2011). Two assumptions were made to simplify the estimation for bi(carbonate) concentrations within the system: a. dominating cations in each system are $H^+$, $Li^+$, $Na^+$, $NH_4^+$, $K^+$, $Mg^{2+}$ and $Ca^{2+}$ whereas those transition and heavy metal ions were out of consideration considering their limited contents of dissolve cations in the atmospheric particulate matter; b. three typical organic acid ions ($CH_3COO^-$, $COOH^-$ and $C_2O_4^{2-}$, major soluble organic acid ions in the atmosphere) were taken into account for ionization balance and the rest of the charge gap in each system was assumed to originate from (bi)carbonate ions. Then we established balance equations (Eqs. **8-11**) for each sample on the basis of charge conservation and ionization equilibrium constant of carbonic acid ($K_1$= 4.47×10$^{-7}$ and $K_2$= 4.69×10$^{-11}$ at 273 K).

$$[H^+]+ [Li^+]+ [Na^+]+ [NH_4^+]+ [K^+]+ 2[Mg^{2+}]+ 2[Ca^{2+}]=[OH^-]+ [F^-]+ [CH_3COO^-]+[COOH^-]+ [NO_2^-] +[Cl^-]+ [NO_3^-]+$$

$$3[PO_4^{3-}]+ 2[SO_4^{2-}]+ 2[C_2O_4^{2-}]+ [HCO_3^-]+ 2[CO_3^{2-}] \tag{8}$$

$$[HCO_3^-]= \frac{K_1[H_2CO_3]}{[H^+]} \tag{9}$$

$$[CO_3^{2-}]= \frac{K_1K_2[H_2CO_3]}{[H^+]^2} \tag{10}$$

It is worth mentioning that [X] is referred to charge concentration for ions (Coulomb·M). Additionally, the temperature factor was also considered to correct the equilibrium constant for (bi)carbonate ions using the following equation:

$$Ln \frac{K_{x'}}{K_x} = -\frac{\Delta H}{R} (\frac{1}{T_{x'}} - \frac{1}{T_x}) \tag{11}$$

where ΔH is the temperature variation (K), R the ideal gas constant (8.31451 J·mol$^{-1}$·K$^{-1}$), and T the temperature (K) during

pH measurements. We then solved those equations to obtain a series of $[HCO_3^-]$, which were eventually corrected by subtracting blank values.

### 2.4 Other measurements and analysis

In addition to the above descriptions of measurements and calculation methodologies, more information including heterogenous reaction setup, kinetics reaction order determination, pretreatment of mineral dust, DRIFTS, IC, Raman measurements and analysis, etc. are available in the supplementary text 1-13.

### 3.    Results and discussion

### 3.2.1 Accelerated sulfate production in the presence of carbonate.

The physico-chemical properties of employed mineral dust proxies including BET surface area, crystal phase, and structure were first characterized (Fig. S1), consistent with early studies (Balachandran and Eror, 1982; Shang et al., 2010; Su et al., 2008), with additional discussion in supplementary text 14. The spectral irradiance of the solar simulator applied in the present study is well covered by natural sunlight (Fig. S2), as much as possible having experimental results from the lab simulate the real atmosphere. Upon solar irradiation under RH of 30% $SO_2/N_2+O_2$ flow ($[SO_2]$ =$2.21\times10^{14}$ molecules $cm^{-3}$), the sulfate production on $TiO_2+CaCO_3$ mixture particles (50 wt.% $TiO_2$ and 50 wt.% $CaCO_3$), measured by IC, is significantly enhanced by 7 times and 23 times compared to that of pristine $TiO_2$ and $CaCO_3$ (Fig. 2a), respectively. In stark contrast, there is a negligible increase in sulfate production detected in the $TiO_2+CaCO_3$ mixture relative to that of pristine $CaCO_3$ and $TiO_2$ in dark experiments (Fig. S3). Great discrepancies in sulfate production over $TiO_2+CaCO_3$ particles between dark and light experiments suggest that carbonate salt may play a different role in these two scenarios. However, the alkalinity of carbonate salt favors $SO_2$ adsorption (Al-Hosney and Grassian, 2005; Yu et al., 2018) and the photo-oxidation process assisted by $TiO_2$ particles is able to strengthen the oxidation efficiency of adsorbed $SO_2$ (Chen et al., 2012; Shang et al., 2010), which is a plausible explanation for increased sulfate production over $TiO_2+CaCO_3$ particles. Following this speculation, two types of mixtures $TiO_2+CaCO_3$ and $TiO_2+CaO$ were employed. In the dark experiments (Fig. 2b), both $TiO_2+CaO$ and $TiO_2+CaCO_3$ almost yield an identical concentration of sulfite and sulfate. On the contrary, $TiO_2+CaCO_3$ particles produce nearly two times of sulfate than $TiO_2+CaO$ particles once irradiated, along with a sharp decrease of S(IV) species on the surface of $TiO_2+CaCO_3$ surfaces. Besides, $CaCO_3$ tends to show relatively humble physical properties including BET surface area, surface pH, as well as solubility, etc. relative to CaO (see detailed discussion in supplementary text 15). The above results allow us to assert that the carbonate-containing system contains an alternative important mechanism for sulfate formation beyond the production of an alkaline environment (additional discussion available in supplementary text 16). Fig. 2c and 2d illustrate that the DRIFTS features of S(IV) and S(VI) species (Nanayakkara et al., 2014; Wu et al., 2011) increase over time on theoretical and experimental $TiO_2+CaCO_3$ mixtures (wt./wt. = 50/50) upon irradiation. The "theoretical" is calculated based on the DRIFTS

experiments of pristine $TiO_2$ and $CaCO_3$ particles through a simple linear superposition whereas the "experimental" is directly derived from the DRIFTS experiment of $TiO_2+CaCO_3$ (wt./wt. = 50/50) particles. These results suggest a synergistic effect presented in this mixture for sulfate formation under solar irradiation.

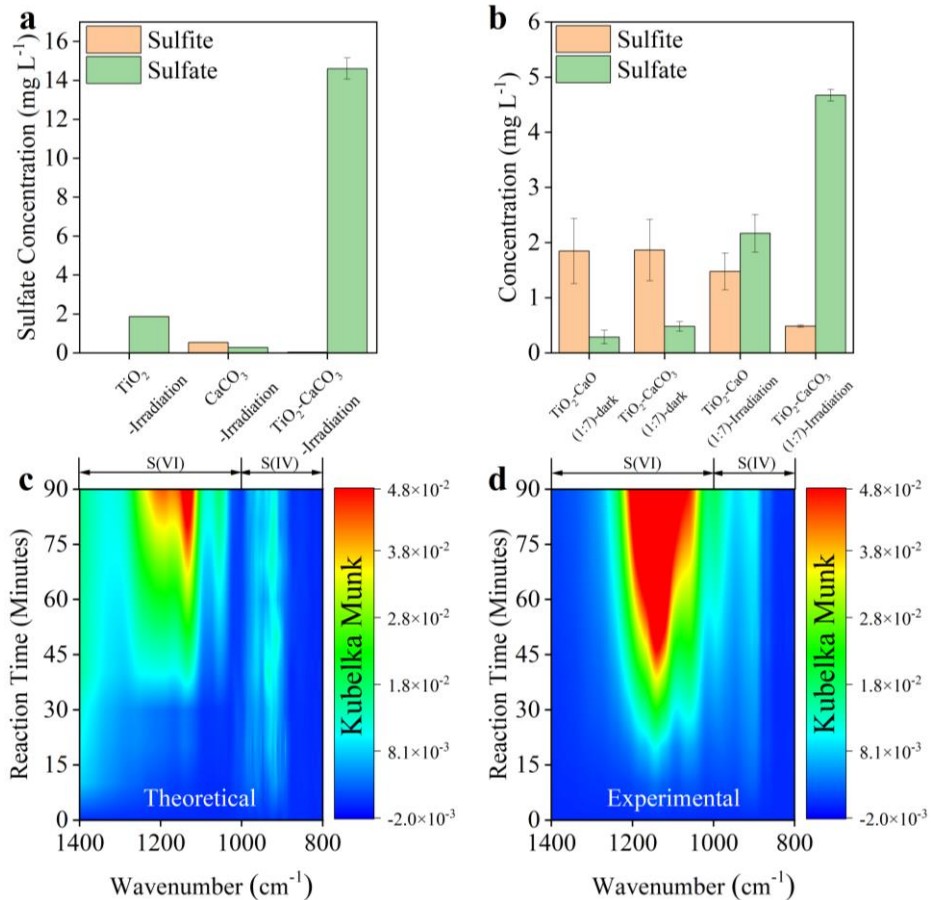

**Figure 2.** (**a**) Sulfate or sulfite concentration quantified by IC on $TiO_2$, $CaCO_3$, $TiO_2+CaCO_3$ particles (wt./wt. = 50/50) after exposure to gaseous $SO_2$ under irradiation for 60 min. (**b**) Sulfate and sulfite concentration quantified by IC on mineral dust particles of concern after exposure to $SO_2$ under dark or irradiation for 20 min. *In situ* DRIFTS of S(IV) and S(VI) production on theoretical (**c**) and (**d**) experimental $TiO_2+CaCO_3$ mixtures (wt./wt. = 50/50) upon irradiation for 90 min. All spectra were processed by the Kubelka-Munk (K-M) algorithm. Noting that the production of sulfur species in theoretical $TiO_2+CaCO_3$ mixtures refers to 0.5 × K-M bands of sulfur species of $TiO_2$ + 0.5 × K-M bands of sulfur species of $CaCO_3$ while that for experimental $TiO_2+CaCO_3$ mixtures refers to 1 × K-M bands of sulfur species of $TiO_2+CaCO_3$ mixtures (wt./wt. = 50/50). Reaction conditions: RH = 30%, Light intensity (I) = 30 mW cm$^{-2}$, Total flow rate = 52.5 mL min$^{-1}$ and $SO_2$ = 2.21×10$^{14}$ molecules cm$^{-3}$.

Combining DRIFTS experiments with the obtained calibration curve (Fig. S4), we estimated that the uptake coefficient of $TiO_2+CaCO_3$ mixture (50wt. % $CaCO_3$) is increased by about a factor of 17 as compared to that of pure $CaCO_3$ or $TiO_2$ (Table S1). More importantly, upon irradiation, $SO_2$ uptake coefficients for these dust mixtures lie at the order of magnitudes of 10$^{-4}$, which is proven to gain importance in overall sulfate production by numerical modeling investigations (Wang et al., 2014; Zhang and Carmichael, 1999). Hence, the photochemical pathway associated with carbonate species is likely a potential

driving force to increase sulfate production in the atmosphere. Meanwhile, the reaction order of $SO_2$ in "$TiO_2$+$CaCO_3$+$SO_2$"

reaction system in the range of 400-20000 ppb is determined to be 0.80 (Fig. S5), indicating that the uptake coefficients

255    obtained at the ppm level of $SO_2$ would somewhat overestimate the real one obtained at atmosphere relevant $SO_2$ concentration

level (several or a few tens ppb). While we note that the difference between the lab and atmospheric conditions regarding $SO_2$

concentration remains even after considering the 400 ppb case, employing hundreds of ppb $SO_2$ in the laboratory simulation

to obtain the kinetic parameter of sulfate formation is acceptable (Liu and Abbatt, 2021; Liu et al., 2020b). Therefore, we

tentatively believe that uptake coefficients estimated in this work would be valid after being calibrated.

260    High-resolution transmission electron microscopy (HRTEM) analysis of $TiO_2$ (50 wt.%) +$CaCO_3$ (50 wt.%) particles after

reaction, in combination with energy dispersive spectrometer mapping measurements of sulfur component, was conducted to

investigate the synergistic effect between $TiO_2$ and carbonate ions (Fig. 3 a-d and Fig. S6). A region with a relatively high

density of sulfur species was selected for further observation (Fig. 3a and 3b) and the distribution of each component was

determined by fast Fourier transformation (FFT) and inverse FFT analyses (panel d) of the selected HRTEM image in high

265    resolution with lattice fringes shown in Fig. 3c. Observation of crystalline $Ti(SO_4)_2$ and $CaSO_4$ on the interface of $TiO_2$ and

$CaCO_3$ components imply that the synergistic effect on sulfate production likely originates from interplays of those two types

of components under solar illumination.

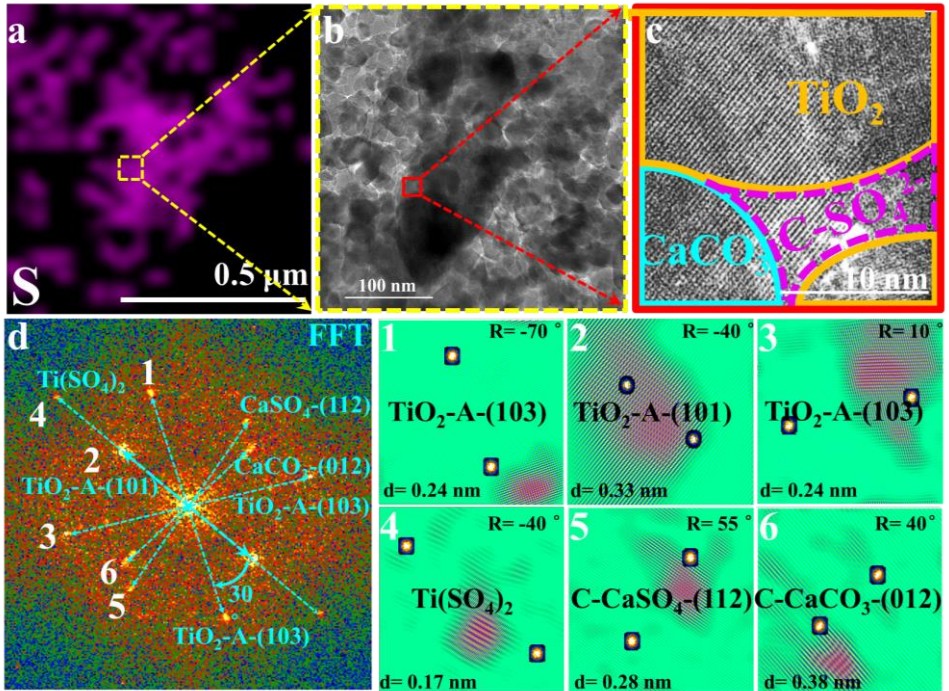

**Figure 3.** (**a**) Energy Dispersive Spectroscopy (EDS) mapping of sulfur. (**b**) Selected HRTEM region containing a high density of sulfur for
270    further observation and the red rectangle refers to the region shown in panel c. (**c**) The HRTEM image in high resolution with lattice fringes
and (**d**) corresponding FFT power spectra, lattice indexing, and (1-6) inverse FFT analysis of lattice signal shown in panel d. In panel c, the
term C-$SO_4^{2-}$ stands for crystalline $SO_4^{2-}$, i.e. $CaSO_4$ and $Ti(SO_4)_2$. Particles employed for the HRTEM measurement refer to $TiO_2$ (50 wt.%)
+$CaCO_3$ (50 wt.%) mixture particles upon exposure to the $4.42\times10^{14}$ molecules cm$^{-3}$ $SO_2/N_2$+$O_2$ for 60 min while other reaction conditions

 are as same as that of above sulfate quantification experiments in Fig.1. Reaction conditions: RH = 30%, Light intensity (I) = 30 mW cm$^{-2}$, Total flow rate = 52.5 mL min$^{-1}$.

We further assessed the importance of interfacial contact between $TiO_2$ and $CaCO_3$ in sulfate production by two synthesis approaches in which the interface abundance is modulated for comparison. Typically, a "grinding" method was used to make $TiO_2$+$CaCO_3$ mixture with compact contact between those two components, thus leading to abundant interfaces. Meanwhile, the "shaking" method is designed to create a $TiO_2$+$CaCO_3$ mixture of loose contact, leaving relatively fewer amounts of interfaces within the mixtures. The resulting mixing statuses of the two samples meet our expectations, evidenced by the scanning electron microscope (SEM) technique (Fig. S7). IC quantification analysis suggests that particles with considerable junctions exhibit a more pronounced promotion for sulfate production than those having relatively few junctions (Fig. S8). These results emphasize the importance of an indispensable interface contact between $TiO_2$ and $CaCO_3$ in fast production upon irradiation.

**Table1.** Chemical compositions of mineral dust simulants.

| | SiO$_2$ (wt. %) [†] | Al$_2$O$_3$ (wt. %) [†] | CaCO$_3$ (wt. %) [*,†] | TiO$_2$ (wt. %) [†] | [‡]Ca:Al (Fe:Al) |
|---|---|---|---|---|---|
| SiO$_2$+Al$_2$O$_3$ | 89.46 | 10.54 | - | - | - |
| SiO$_2$+Al$_2$O$_3$+CaCO$_3$ | 82.47 | 9.72 | 7.81 | - | 0.73 (-) |
| SiO$_2$+Al$_2$O$_3$+TiO$_2$ | 88.42 | 10.42 | - | 1.15 | - |
| SiO$_2$+Al$_2$O$_3$+TiO$_2$ +CaCO$_3$ | 81.59 | 9.62 | 7.73 | 1.06 | 0.73 (-) |
| ATD | 71.27 | 8.4 | - | 0.93 | 0.73 (0.17) |

*In the present study, $CaCO_3$ was taken as representative of alkaline-earth metal oxide in our proxies for the authentic dust.

[†]The mass ratio of 4 components (if any) in the simulants were controlled in the ratio of SiO$_2$:Al$_2$O$_3$:CaCO$_3$: TiO$_2$= 81.59: 9.62:7.73:1.06. For instance, the mass ratio of SiO$_2$ to Al$_2$O$_3$ is 81.59: 9.62 in the SiO$_2$+Al$_2$O$_3$ simulant whereas is 81.59: 9.62:7.73 in the SiO$_2$+Al$_2$O$_3$+CaCO$_3$ mixture. The ratios of each component are derived from the EDS mapping analysis of ATD dust particles.

[‡]This column refers to the molar ratio.

The increased sulfate production was further probed by employing mineral dust simulants where two dominant crust constituents $SiO_2$ and $Al_2O_3$ were introduced into $TiO_2$+$CaCO_3$ particles to mimic the authentic mineral dust particles in the atmosphere, with specific component and corresponding ratio information shown in Table 1. It is worth mentioning that the determination of the ratio of each component in the simulants relies upon the EDS mapping results of ATD particles. In Fig. 4, the introduction of $TiO_2$ components (1%wt.) into $SiO_2$+$Al_2O_3$ leads to 81.6% enhancement of sulfate production because of photolabile ROS. On the other hand, merely 24.8% wt. increase of sulfate yield was observed once ~8% wt. of $CaCO_3$ was incorporated into $SiO_2$+$Al_2O_3$ dust particles. This can be attributed to the alkaline environment created by $CaCO_3$, which is

thought to increase SO$_2$ adsorption (Al-Hosney and Grassian, 2005) and sulfate production accordingly. Surprisingly, mixing of ~1% mass fraction of TiO$_2$ and ~8% wt. of CaCO$_3$ into SiO$_2$+Al$_2$O$_3$ gives rise to a 235% increase in sulfate formation

relative to that of SiO$_2$+Al$_2$O$_3$. It represents nearly an extra 100% enhancement of sulfate production due to the presence of TiO$_2$ and CaCO$_3$ of an atmospherically relevant mass fraction. These results lead to the hypothesis that the observed synergistic effect on heterogeneous oxidation of SO$_2$ is likely to take effect in the atmosphere.

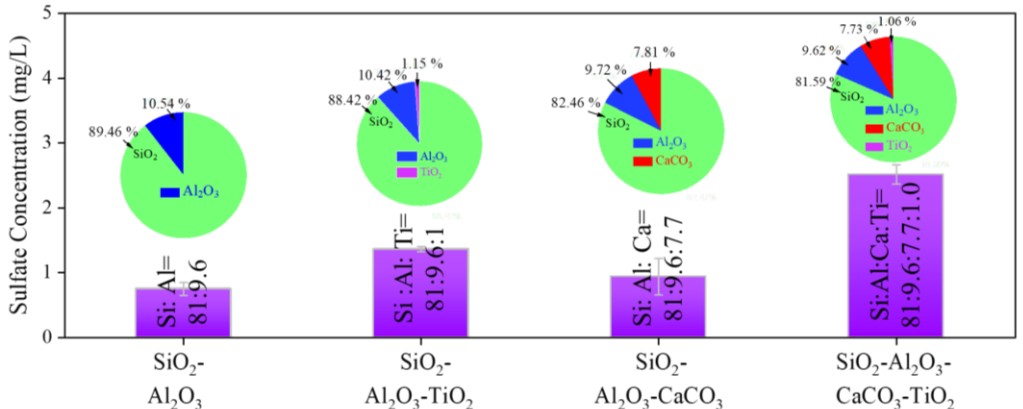

**Figure 4.** Sulfate concentration quantified by IC. Sulfate concentration was measured by IC on mineral dust simulants after

exposure to gaseous SO$_2$ (2.46×10$^{14}$ molecule cm$^{-3}$) under irradiation. Noting that SiO$_2$: Al$_2$O$_3$: CaCO$_3$: TiO$_2$ refers to the mass fraction ratios of the components in simulants. Experiments were all conducted at RH of 30% and Light intensity (I) of 30 mW cm$^{-2}$.

Fe$_2$O$_3$ is also one of the crucial components found in authentic mineral dust (El Zein et al., 2013), and it has been reported to produce electron-hole pairs under solar irradiation (Li et al., 2019), thus likely involving the reaction channel considered in

this work (see detailed discussion in the later section). Similar to the protocol applied for synthesizing the TiO$_2$+CaCO$_3$ mixture, αFe$_2$O$_3$+CaCO$_3$ are accordingly prepared. In Fig. S9a, our results show that αFe$_2$O$_3$ can not trigger fast SO$_2$ oxidation in the presence of carbonate ions upon irradiation, which is distinguished from the results we derived from TiO$_2$+CaCO$_3$ mixture. This can be explained by the fact that Fe$_2$O$_3$ shows a lower redox activity relative to TiO$_2$ (Fig. S9b), where its strong redox capability essentially enables photo-induced electrons and holes to produce O$_2^-$ and ·OH radical ions. In stark contrast, the

valence band and conduct band of Fe$_2$O$_3$ lie at -0.18 and at 1.68 V vs. NHE (pH = 7), lower than the redox potential required for generating O$_2^-$, ·OH as well as carbonate-containing ROS (Li et al., 2016). Hence, no promoted sulfate production is seen for αFe$_2$O$_3$+CaCO$_3$ particles under irradiation. We also note the inconsistency between our study and the previous literature with regard to the response of SO$_2$ oxidation to solar irradiation over αFe$_2$O$_3$ particles, which has been interpreted in supplementary text 17.

Overall, we show that upon irradiation atmospherically relevant content of TiO$_2$ (nearly 1%) in mineral dust simulants is able to interact with carbonate ions to launch an increased sulfate production, which is beyond the conventional regime of

alkaline neutralization of $H_2SO_4$. Unlike $TiO_2$, $\alpha Fe_2O_3$ lacks the ability to initiate fast $SO_2$ oxidation by generating carbonate-containing ROS due to its limited photo-chemical activity although ferric chemistry is important in secondary sulfate formation in the atmosphere (Sullivan et al., 2007; Yermakov and Purmal, 2003).

### 3.2.2 Accelerated sulfate production in the presence of $CO_2$.

Atmospheric $CO_2$ is also an important source of (bi)carbonate. Its influence on photochemical $SO_2$ uptake on mineral dust was thus studied. Distinguishing S(VI) from S(IV) species over $TiO_2$ particles relies upon the position of the IR bands according to the assignment of previous literature (Nanayakkara et al., 2014), and S(VI) and S(IV) build up as heterogeneous reactions proceed in all cases (Fig. 5). In the presence of atmospherically relevant $CO_2$ ($9.83\times10^{15}$ molecules cm$^{-3}$), sulfate yield was increased under irradiation as compared to the $CO_2$-free case (Fig. 5a and b). We cautiously examined the net effect of formed (bi)carbonate on sulfate production by time-resolved DRIFTS spectra (Fig. 5c and d) using dark experiments. $CO_2$ suppresses both S(IV) and S(VI) products in the dark probably because of the competitive adsorption effect, as we observed over $Al_2O_3$ particles (Liu et al., 2020c).

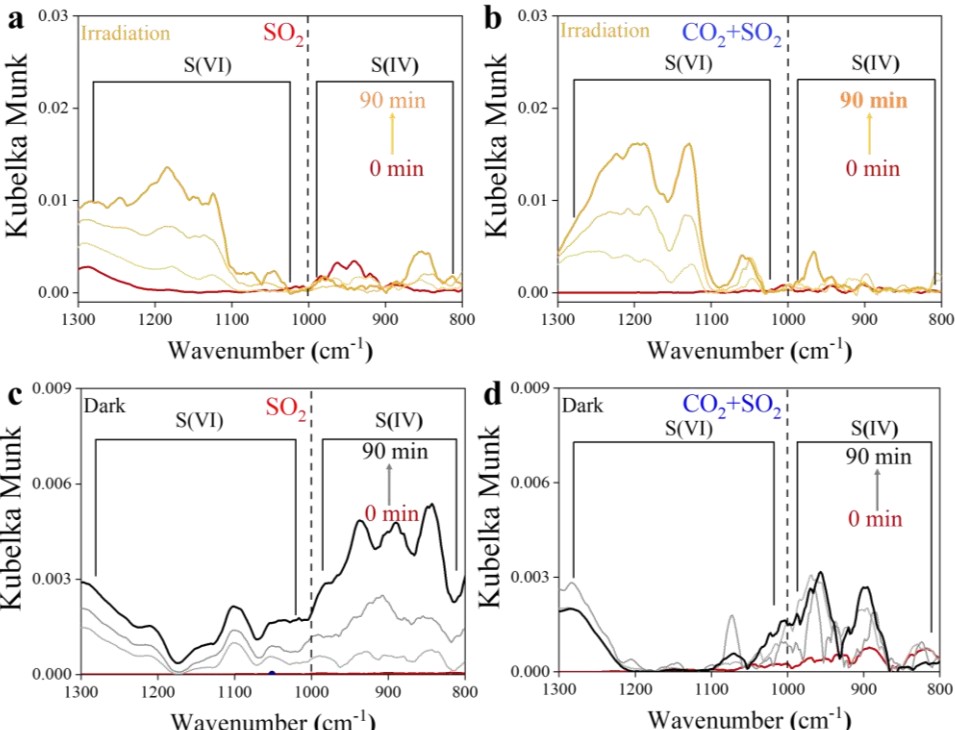

**Figure 5.** Time-resolved DRIFTS of S(IV) and S(VI) products over $TiO_2$ particles after exposure to $SO_2/N_2+O_2$ in the absence and presence of $CO_2$ upon irradiation (**a** and **b**) and those reactions under dark (**c** and **d**). Reaction conditions: RH = 30%, Light intensity (I) = 30 mW cm$^{-2}$, total flow rate = 52.5 mL min$^{-1}$ and $SO_2$ = $7.37\times10^{13}$ molecules cm$^{-3}$.

Above observation leads us to speculate that the active intermediate derived from (bi)carbonate species upon irradiation is a plausible force to drive rapid sulfate formation. Besides, a nearly 50% increase in $SO_2$ uptake coefficient is observed for the

mineral dust proxy $TiO_2$ after being exposed to $9.83\times10^{15}$ molecules $cm^{-3}$ (400 ppm) $CO_2+SO_2/N_2+O_2$ mixture (Table S1). The pseudo-first reaction order (1.13) was also determined in the selected concentration range of 400-20000 ppb (Fig. S5b), which satisfies the prerequisite for uptake coefficients derived from laboratory chambers potentially being generalized to the atmosphere condition, as we expounded in the early context.

**Table 2.** Chemical Compositions of the ATD Dust and Standard Clays.

| Minerals | ATD (%)* | IMt-2 (%)* | KGa-2 (%)* |
|---|---|---|---|
| $SiO_2$ | 78.11 | 59.57 | 56.93 |
| $Al_2O_3$ | 7.19 | 19.47 | 37.49 |
| $Fe_2O_3$ | 2.57 | 7.95 | 1.81 |
| FeO | n.d. [†] | 0.05 | 0.15 |
| MgO | 1.22 | 2.42 | 0.03 |
| CaO | 3.03 | 0.37 | 0.01 |
| $Na_2O$ | 1.39 | 0.08 | n.d. [†] |
| $K_2O$ | 2.06 | 8.72 | 0.06 |
| $TiO_2$ | 0.46 | 0.99 | 3.43 |
| $P_2O_5$ | 0.10 | 0.07 | 0.05 |
| MnO | 0.04 | 0.03 | n.d.[†] |
| S | n.d. [†] | 0.03 | 0.02 |
| Total | 99.21 | 99.75 | 99.98 |
| Total A.E. [‡] | 7.70 | 11.59 | 0.10 |

*Chemical compositions of the dust and clays were determined by XRF results.

[†]n.d. refers to not detected.

[‡]A.E. refers to alkaline earth metal oxide.

    As another step toward a real scenario in the atmosphere, experimental trials employing authentic mineral dust particles, i.e. K-Ga-2 (Kaolin, Georgia, USA), Arizona test dust (ATD), and clays IMt-2 (Illite, Mont., USA) were implemented, with

component analysis results shown in Table 2. In Fig. 6, the pronounced increase in sulfate yield (by nearly 100% increased sulfate production in the $CO_2$-involved case under irradiation) is best seen in K-Ga-2 clay (panel a). The promotional effect of $CO_2$ on sulfate formation under irradiation, nonetheless, is less evident for IMt-2 (the content of $TiO_2 \approx 0.99\%$) and ATD (the content of $TiO_2 \approx 0.46\%$) as compared to K-Ga-2 particles. This may correlate to their higher mass fraction of alkaline earth metal oxide (denoted as A.E.), which enables dust particles to possess a substantial number of (bi)carbonate species within the

natural environment where they have experienced long-term exposure to atmospheric $CO_2$ during the regional transport. Therefore, the aforementioned synergetic effect takes effect over IMt-2 and ATD particles even without exposure to $CO_2$

presumably due to the presence of abundant alkaline carbonate formed, and a relatively moderate increase in sulfate production was thus observed. On the other hand, $TiO_2$ content is not necessarily an accurate predictor of photoreactivity, the content and proportion of the active phase of $TiO_2$ in K-Ga-2 altogether contribute to a more pronounced increase in sulfate production

relative to the other two clays (see detailed discussion in supplementary text 18).

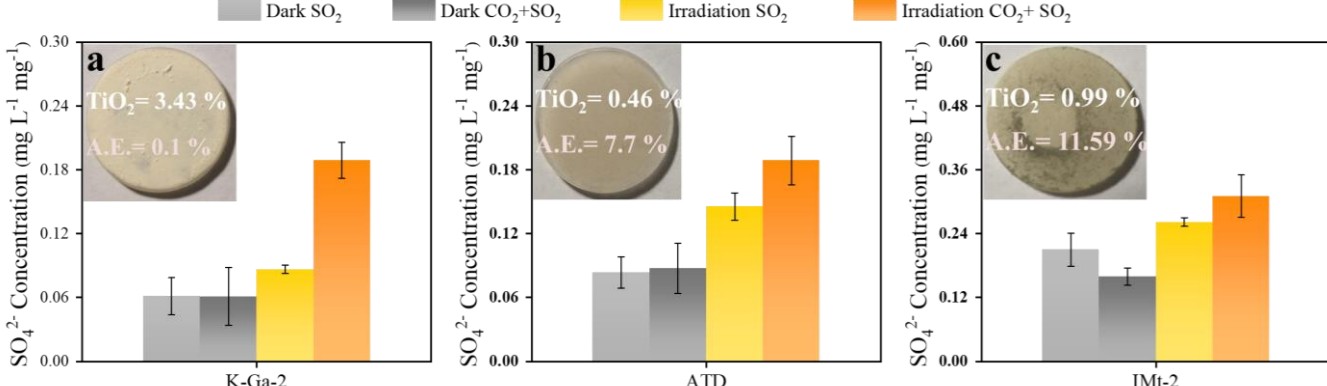

**Figure 6.** Laboratory studies of sulfate production on authentic dust and clay membranes **(a)** K-Ga-2 **(b)** ATD as well as **(c)** IMt-2 under the dark and irradiation (30 mW cm$^{-2}$) upon exposure to $4.91 \times 10^{14}$ molecules cm$^{-3}$ $SO_2/N_2+O_2$ and $2.46 \times 10^{18}$ molecules cm$^{-3}$ $CO_2+ 4.91 \times 10^{14}$ molecules cm$^{-3}$ $SO_2/N_2+O_2$ at RH of 30%. Noting that sulfate yield in three cases was normalized by the mass of dust particles employed

for heterogeneous reaction.

### 3.2.3 Reaction Mechanism.

The heterogeneous reaction of $SO_2$ on dust particles in the atmosphere is a complicated process, covering a series of reactions taking place in both homogeneous and heterogeneous ways. At a sufficiently low RH condition (normally below 10% RH), water readily dissociates on the surface of metal oxide under ambient atmospheric conditions, where the metal oxide surface

is terminated by hydroxyl groups that hydrogen bond to adsorbed water molecules (Cwiertny et al., 2008). In this case, $SO_2$ oxidation over dust particles is dominated by the reaction regime where the resulting hydroxyl groups react with gaseous $SO_2$ to form adsorbed $S(IV)_{ad}$ species. Afterward, $S(IV)_{ad}$ will be oxidized by oxidants in the atmosphere or photo-induced active intermediates produced from the dust surface upon irradiation. As the RH increases beyond 10%-15%, multilayer water coverage occurs, reaching approximately two monolayers at RH of 30 % (Mogili et al., 2006). Under these circumstances, the

amount of water adsorbed onto the surface of the dust particles is believed to be sufficiently large that it is liquid-like in corresponding physical and chemical properties (Cwiertny et al., 2008) (Peters and Ewing, 1997). In this work, heterogenous $SO_2$ oxidation over mineral dust proxies proceeds at the RH of 30%, and two water layers are likely to attach to dust particles. Thus, radical ions are anticipated to play a key role in fast $SO_2$ oxidation over humidified mineral dust and mechanism studies performed in the aqueous phase are persuasive to some extent.

Our preliminary sulfate quantification results (Fig. 2a and b) suggest that the presence of (bi)carbonate ions under solar light contributes to increased sulfate yield. In this carbonate-containing reaction system, a plausible intermediate is the active $CO_3^{\cdot-}$. It is readily produced via the following two pathways. First of all, carbonate anion can be directly oxidized by produced photo-induced holes from typical mineral dust upon solar irradiation (**R1** and **R2**), as the redox potential of $CO_3^{\cdot-}/CO_3^{2-}$ is 1.78 V (vs NHE, at pH = 7), which is lower than the $TiO_2$ valence band (VB) potential of 2.67 V (vs NHE, at pH = 7)  (Li et al., 2016;

Xiong et al., 2016):

$$\text{Mineral dust} + h\nu \rightarrow h^+ + e^- \tag{R1}$$

$$h^+ + CO_3^{2-} \rightarrow CO_3^{\cdot-} \tag{R2}$$

      In the second pathway, carbonate radicals evolve through the reaction of (bi)carbonate anion with formed hydroxyl radicals ·OH over mineral dust surfaces (Zhang et al., 2015a) (**R3** and **R4**).

$$h^+ + H_2O_{ad} \rightarrow \cdot OH + H^+ \tag{R3}$$

$$\cdot OH + HCO_3^-/CO_3^{2-} \rightarrow CO_3^{\cdot-} + H_2O/OH^- \tag{R4}$$

      The above assumptions are supported by nanosecond transient absorption spectra (NTAS), in which signal (ΔOD) of carbonate radical $CO_3^{\cdot-}$ at 600 nm (Bhattacharya et al., 1998) only emerges for dust suspension containing (bi)carbonate species (Fig. 7a). An increased degradation rate of aniline observed in $TiO_2$ suspension due to the presence of carbonate ions produces

additional evidence of the formation of active $CO_3^{\cdot-}$ ions and strengthened oxidation capability of $TiO_2$ (Fig. 7b, see additional discussion in supplementary text 19). The $CO_3^{\cdot-}$-induced chemistry was further evidenced by ·OH scavenging experiments using tertiary Butyl Alcohol (TBA) and isopropanol (i-PrOH) as they show lower reaction rates with $CO_3^{\cdot-}$ ($k_{CO_3^{\cdot-},i\text{-PrOH}} < 4.0$ $\times10^4$ M$^{-1}$ s$^{-1}$ and $k_{CO_3^{\cdot-},TBA} < 1.6 \times 10^2$ M$^{-1}$ s$^{-1}$) relative to that with ·OH ($k_{i\text{-PrOH}\sim\cdot OH} < 1.9 \times10^9$ M$^{-1}$ s$^{-1}$ and  ($k_{\cdot OH,TBA} = 6$ $\times10^8$ M$^{-1}$ s$^{-1}$) (Buxton et al., 2009; Liu et al., 2015) (Liu et al., 2015) (Li et al., 2020). TBA dramatically decreases the yield

of sulfate on $TiO_2$ surface by nearly 70%, with sulfite ions being the dominant sulfur species (Fig. 7c). Meanwhile, a great loss of sulfate yield when $TiO_2$ suspension was added with i-PrOH (Fig. 7d). This is in strong contrast to the result of a carbonate-involved system where the reactivity is sustained as carbonate radicals offer an alternative reaction pathway for $SO_2$ oxidation. This is plausible since the carbonate ions are excellent ·OH scavenger, and $CO_3^{\cdot-}$ becomes the predominant species in a relatively strong alkaline aqueous-like environment in the presence of carbonate salt. This is supported by the previous work

(Sun et al., 2016), in which adding 0.1 M of bicarbonate salt into the UV/$H_2O_2$ system ($H_2O_2$ = 0.3 mM) was sufficient to suppress ·OH concentration to around $10^{-15}$ M, creating a carbonate radical dominated regime ([$CO_3^{\cdot-}$] = $8.64 \times 10^{-12}$ M). In our experiments (Fig. 7b), 0.2 M of carbonate salt was employed, and the reaction rate of $CO_3^{2-}$ with ·OH is nearly two orders of magnitude higher than that of $HCO_3^-$, thus giving rise to carbonate radical being the substitute for hydroxyl radical in the reaction. The above results suggest that ·OH is a major contributor to sulfate yield on $TiO_2$ particles in the absence of carbonate

ions while $CO_3^{\cdot-}$ ions dominate $SO_2$ oxidation over humidfied carbonate-containing $TiO_2$ particles upon irradiation. In addition to experimental investigations, the carbonate radical formation process is proved to be thermodynamically favorable, supported by density functional theory (DFT) calculations (Fig. S10 and supplementary text 20).

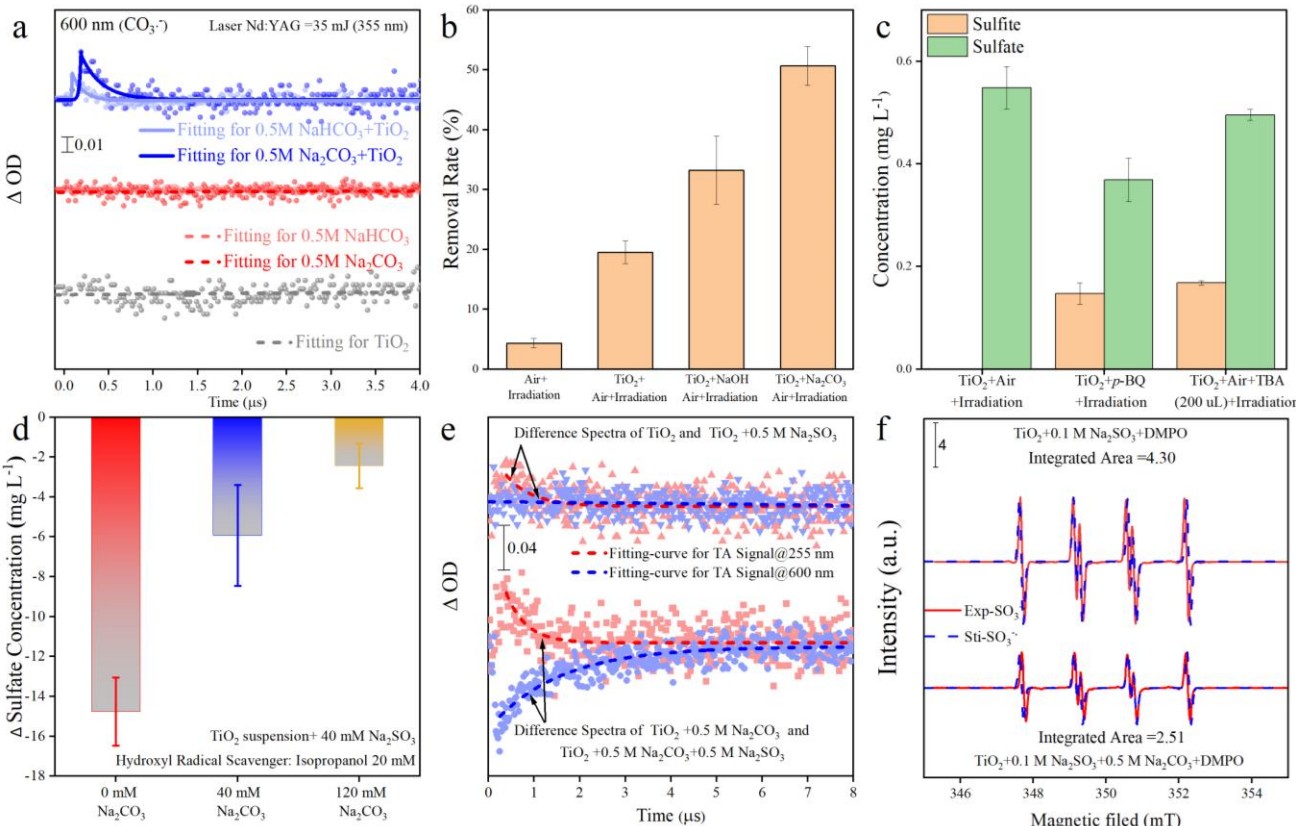

**Figure 7.** (**a**) Single-wavelength transient absorption spectra of various aqueous solutions. (**b**) The removal rate of aniline after exposure to airflow under irradiation in the absence and presence of mineral dust particles for 300 seconds. Reaction conditions: RH = 30%, Light intensity (I) = 30 mW cm$^{-2}$, Total flow rate = 52.5 mL min$^{-1}$. Noting that an adequate amount of NaOH was introduced into the TiO$_2$ suspension system to achieve a pH environment condition identical to that of the TiO$_2$+Na$_2$CO$_3$ suspension system. (**c**) Determination of sulfite and sulfate concentration after exposure to air flow under irradiation in the absence and presence of mineral dust particles for 20 min. Reaction conditions: RH = 30%, Light intensity (I) = 30 mW cm$^{-2}$, Total flow rate = 52.5 mL min$^{-1}$. (**d**) Sulfate formation change Δ(SO$_4^{2-}$) is determined by different sulfate concentrations with and without the addition of isopropanol as hydroxyl radical scavenger. (**e**) The difference in transient absorption kinetics of sulfite radical and carbonate radical at the various aqueous solutions and their corresponding growth-decay fit curves. ΔA-signal was recorded at 255 and 600 nm after pulsed 355 nm laser excitation. (**f**) ESR spectrometry of [DMPO–SO$_3^{--}$] intermediate formed in a solution of d TiO$_2$ (3 mg~4 mL) + 0.1 M Na$_2$SO$_3$ and TiO$_2$ (3 mg~4 mL) + 0.5 M Na$_2$CO$_3$ + 0.1 M Na$_2$SO$_3$. For clarity, the integrated areas of ESR profiles were also presented for direct comparison. Exp. and Sti. stand for experimental results and corresponding fitting results using software Isotropic Radicals.

On the other hand, early studies (Chameides and Davis, 1982; Das, 2001; Neta and Huie, 1985) agree with the key role of sulfite radical (SO$_3^{--}$) in rapid sulfate production in an aqueous medium, and the present reaction system creates a localized environment where SO$_3^{--}$ can be readily produced from the TiO$_2$ and S(IV) species upon solar illumination (Salama et al., 1995). Consequently, probe light of NTAS at wavelength 255 nm (ascribed to sulfite radical) and 600 nm (ascribed to carbonate

radical) were simultaneously monitored (Ghalei et al., 2016; Goldstein et al., 2001; Hayon et al., 1972). A weak signal of sulfite radical was observed in the system of $TiO_2+Na_2SO_3$ suspension under irradiation (Fig. 7e). On the contrary, the sulfite radical signal is strengthened after the introduction of carbonate ions into the $TiO_2+Na_2SO_3$ suspension, along with a significant decrease in the signal for carbonate radical. Electron spin resonance (ESR) data (Fig. 7f) further confirms the increase of $SO_3^{\cdot-}$ after 2 min UV irradiation in the presence of carbonate ion. Based on the above results, one may deduce that the interplay

between carbonate radical and sulfite ions is a crucial step giving rise to the increased $SO_3^{\cdot-}$, which is reported to account for rapid atmospheric sulfate formation through chain propagation reactions that involve $SO_4^{\cdot-}$ and $SO_5^{\cdot-}$ intermediates (Hung and Hoffmann, 2015; Hung et al., 2018). Additionally, this sulfite radical ion chemistry is believed to drive fast sulfate formation over mineral dust particles as well (Gankanda et al., 2016; Rubasinghege et al., 2010). Nevertheless, there are two possibilities that might explain the aforementioned interaction. One way is the oxygen transfer and the other route is electron transfer,

which needs further clarification.

We first examined the oxygen transfer path through $^{18}O$ isotope labeling experiments. $TiO_2$ particles were initially exposed to $C^{16}O_2/N_2$ and $C^{18}O_2/N_2$, followed by the exposure of $SO_2/N_2+O_2$ under irradiation (Fig. 8a). Bidentate carbonate band centered at 1573 $cm^{-1}$ appears after the introduction of $C^{16}O_2/N_2$, while this band shifts to 1558 $cm^{-1}$ when $C^{18}O_2/N_2$ is introduced, indicating the incorporation of $^{18}O$ into bidentate carbonate species, in accord with the previous report (Liao et al.,

2002). However, no shift of IR features at 1269, 1219, and 1159 $cm^{-1}$, assigned to (bi)sulfate species on $TiO_2$ particles, were observed throughout the reaction. This implies that the oxygen transfer path does not account for the rapid $SO_2$ oxidation on particles of concern.

In light of the above analysis, the electron transfer might be a plausible pathway to explain the fast oxidation within the reaction system. DFT calculations provide an accessible approach to study the electron transfer pathway. The result in Fig. 8b

illustrates $SO_3^{\cdot-}$ formation is a SET process of $CO_3^{\cdot-}$ and $SO_3^{2-}$, where O atom in $SO_3^{2-}$ transfers an electron to O atom in $CO_3^{\cdot-}$ to form $SO_3^{\cdot-}$ and $CO_3^{2-}$. This SET reaction is a thermodynamically favorable process, with the difference of Gibbs free energy between reactant and product lying at -24.09 kcal $mol^{-1}$. We note that the insufficient $O_2$ supply in aqueous media may be an underlying constraint to the proposed $CO_3^{\cdot-}$-initiated $SO_2$ oxidation pathway. Therefore, careful estimation of both oxygen consumption and supply rates were conducted, revealing that oxygen supply flux can be sufficiently larger than corresponding

consumption (see additional details in the supplementary text 21). This enables us to deduce that considered chain reactions can continually proceed. Taken above results and discussions altogether, the following reactions are proposed accordingly (**R5-R8**):

$$CO_3^{\cdot-} + SO_3^{2-} \rightarrow CO_3^{2-} + SO_3^{\cdot-} \qquad\qquad \textbf{[R5]}$$

$$SO_3^{\cdot-} + O_2 \rightarrow SO_5^{\cdot-} \qquad\qquad \textbf{[R6]}$$

$$SO_5^{\cdot-} + SO_3^{2-} \rightarrow SO_4^{\cdot-} + SO_4^{2-} \qquad\qquad \textbf{[R7]}$$

$$SO_4^{\cdot-} + SO_3^{2-} \rightarrow SO_4^{2-} + SO_3^{\cdot-} \qquad\qquad \textbf{[R8]}$$

Another important pathway needs to be considered as well; that is $SO_3^{\cdot-}$ can also be formed via the conventional reaction of $\cdot OH$ and $SO_3^{2-}$ (**R9**).

$$\cdot OH + SO_3^{2-} \rightarrow SO_3^{\cdot-} + OH^- \qquad [\textbf{R9}]$$

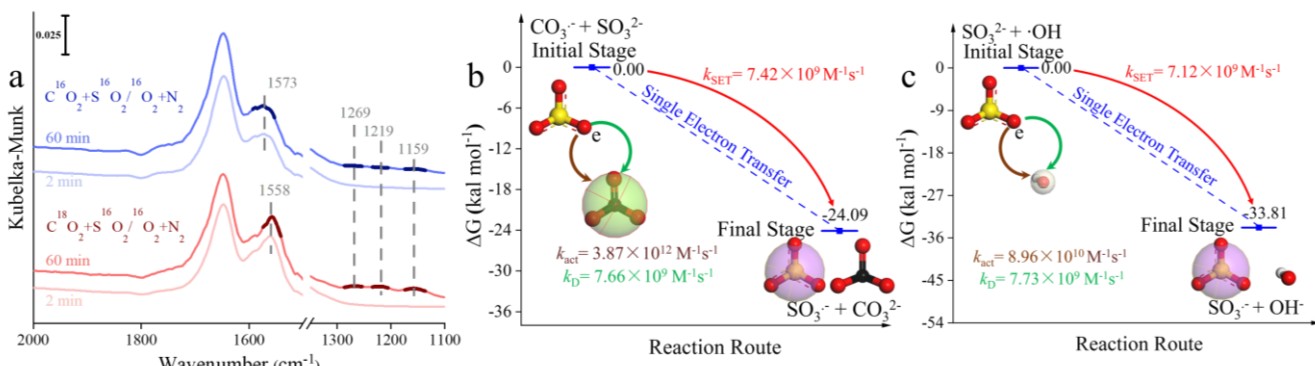


**Figure 8.** (**a**) *In situ* DRIFTS of heterogeneous reaction of $SO_2$ on the $TiO_2$ particles for 2 and 60 min after being exposed to $C^{16(18)}O_2/N_2$ for 20 min under irradiation. (**b**) Reaction pathway of interaction between carbonate radical ($CO_3^{\cdot-}$) and sulfite ($SO_3^{2-}$) and (**c**) Interaction between hydroxyl radical ($\cdot OH$) and sulfite ($SO_3^{2-}$) through the SET process at the CCSD (T)-F12/cc-PVDZ-F12//M06-2X/6-311++G (3df, 3pd) level and $\Delta G_0^{SET}$ represents the difference in Gibbs free energy between reactant and product. The white, black, yellow, and red spheres
represent H, C, S, and O atoms, respectively. In order to visualize the variation of surface products in oxygen isotope experiments (panel a), DRIFTS features of these concerned species were highlighted in dark colors. For interpretation of the references to color in the legends of panels b and c, the reader is referred to the Web version of this article.

In this SET process, electron donor $SO_3^{2-}$ reacts spontaneously with electron acceptor $\cdot OH$ (Fig. 8c) and the calculated activation free energy barrier $\Delta G^{\neq}_{SET}$ for this SET reaction is 2.50 kcal mol$^{-1}$. Hence, the reaction process of $\cdot OH$ with $SO_3^{2-}$ is
diffusion-controlled, and the total rate constant $k_{SET-2}$ was calculated to be $7.12 \times 10^9$ M$^{-1}$s$^{-1}$. In comparison, the rate constant $k_{SET-1}$ of the diffusion-controlled SET process for $CO_3^{\cdot-}$ and $SO_3^{2-}$ was estimated to be $7.42 \times 10^9$ M$^{-1}$s$^{-1}$. Despite a slight net increase of the rate, the distinguishable concentration of $CO_3^{\cdot-}$ and $\cdot OH$ should also be taken into account for the rate comparison in varied reaction paths. To visualize the difference, relative rates were calculated according to Eq. **12**:

$$r = \frac{v_{CO_3^{\cdot-}+SO_3^{2-}}}{v_{\cdot OH+SO_3^{2-}}} = \frac{k_{SET-1}[CO_3^{\cdot-}][SO_3^{2-}]}{k_{SET-2}[\cdot OH][SO_3^{2-}]} \qquad [\textbf{12}]$$

Where $r$ is the ratio of two reaction rates, $[CO_3^{\cdot-}]$, $[SO_3^{2-}]$, and $[\cdot OH]$ refer to the concentration of corresponding reactants. Previous literature suggests the concentration of carbonate radicals is able to show two orders of magnitude higher than that of hydroxyl radicals at the surface of the water under solar irradiation (Chandrasekaran and Thomas, 1983; Goldstein et al., 2001; Sulzberger et al., 1997), consistent with concentration gap between carbonate radicals and hydroxyl radicals through partitioning process from gas-phase determined in our reaction system (Fig.9). While the net concentrations of carbonate and
hydroxyl radicals in the water layers of humified particles are very likely to be different from that found in the bulk aqueous

media, concentration inputs of two radicals with the gap of two orders somehow could reflect the relative contribution of carbonate radicals and hydroxyl radicals to sulfate production based on literature results and our experimental trails. The concentrations of $CO_3^{\cdot-}$ and $\cdot OH$ were set in the range from $1.0 \times 10^{-10}$ to $1 \times 10^{-12}$ mol $L^{-1}$ and from $1.0 \times 10^{-12}$ to $1 \times 10^{-14}$ mol $L^{-1}$ (Sulzberger et al., 1997) and $r$ value could thus reach to $1.04 \times 10^4$ at most (Fig. S11). As a result, we speculate that the formation pathway of $SO_3^{\cdot-}$ via interaction between $CO_3^{\cdot-}$ and $SO_3^{2-}$ is a more efficient route, corresponding well with our experimental results.

In addition to the pathway launched by photo-generated holes, the sink of photo-generated electrons is as well considered. In our reaction system, $O_2$ is thought to be an electron trap and produce the superoxide radical ions ($O_2^{\cdot-}$), which is reported to play a non-negligible role in sulfate formation (Shang et al., 2010) and should be taken into account to give a whole picture of reaction scheme in triggering sulfate formation on the surface of $TiO_2$-containing mineral dust particles. $p$-benzoquinone is a commonly-used $O_2^{\cdot-}$ scavenger for trapping the $O_2^{\cdot-}$ radical ions (Yan et al., 2018). Our data shows that adding an excess amount of $p$-benzoquinone into $TiO_2$ particles reduces the sulfate yield by 32% along with the appearance of sulfite ions over $TiO_2$ particles upon exposure to $SO_2$ (Fig.7c). Notably, the decrease in sulfate yield by around 30% in the presence of $O_2^{\cdot-}$ scavenger $p$-benzoquinone is almost complementary to that added with $\cdot OH$ scavenger using TBA (70%), pointing toward a minor sulfate formation pathway contributed by $O_2^{\cdot-}$ relative to the major pathway by $CO_3^{\cdot-}$ when carbonate ions are presented to efficiently capture $\cdot OH$ ions. Following Shang's work (Shang et al., 2010), $O_2^{\cdot-}$ involved $SO_2$ oxidation can be given as **R10-R12**:

$$e^- + O_2 \rightarrow O_2^{\cdot-} \qquad\qquad\qquad\qquad\qquad\qquad\qquad\qquad\qquad\qquad\qquad [\textbf{R10}]$$

$$SO_2 + O_2^{\cdot-} \rightarrow SO_3 + O^- \qquad\qquad\qquad\qquad\qquad\qquad\qquad\qquad\qquad [\textbf{R11}]$$

$$SO_3 + H_2O \rightarrow H_2SO_4 \qquad\qquad\qquad\qquad\qquad\qquad\qquad\qquad\qquad\qquad [\textbf{R12}]$$

Where intermediates $SO_3$ formed via the interaction between $SO_2$ and $O_2^{\cdot-}$, subsequently couple with water molecules to produce sulfate species as a final product. pH is an important factor within aqueous chemical reaction processes and is in preference to alter the dominated regime for sulfate production. Yet so far adjusting the pH of particle surfaces is quite tough, and exploring the role of dust surface pH in the reactivity of $CO_3^{\cdot-}$ is not easily achieved. Notwithstanding, the increase of pH in $TiO_2$ suspension was observed to promote the production of $CO_3^{\cdot-}$, further strengthening the oxidation capability of dust particles (Fig. 7b). In contrast, decreasing pH is expected to reduce the yield of $CO_3^{\cdot-}$ since the reaction rate of $CO_3^{2-}$ with $\cdot OH$ is nearly two orders of magnitude higher than that with $HCO_3^-$. On this basis, a question arises whether the surface pH of mineral dust can be sustained to maintain fast $SO_2$ oxidation triggered by $CO_3^{\cdot-}$ in the typical lifespan of mineral dust.

Considering that $SO_2$ concentration employed in this work is higher than that in the real atmosphere, the concept of "equivalent exposure time" is introduced to evaluate the influence of pH on the $SO_2$ oxidation pathway initiated by $CO_3^{\cdot-}$ (see a more detailed discussion on determining equivalent exposure time in supplementary text 22). The heterogeneous sulfate production over $TiO_2$ and $TiO_2+CaCO_3$ particles versus equivalent exposure time were plotted (Fig. S12). Clearly, the sulfate yield builds up steadily during the two-week equivalent exposure time, suggesting that the regime of $CO_3^{\cdot-}$ initiated $SO_2$

oxidation over $TiO_2$ and $TiO_2+CaCO_3$ particles is slightly affected by the possible decrease of surface pH because of the accumulation of sulfate production over entire reaction course. In the atmosphere, the lifetime of mineral dust particles ranges from several days to weeks (Bauer and Koch, 2005), and the equivalent exposure time considered in this study (nearly 2 weeks) falls right within the characteristic lifespan range of mineral dust particles. Besides, 20 ppb is assumed to be an atmospherically relevant concentration to calculate "equivalent exposure time" in this study whereas even low $SO_2$ concentrations (several or a few tens ppb of $SO_2$) were monitored in the field observation (He et al., 2014; Watanabe et al., 2020). Therefore, the reduction of dust surfaces pH would be more moderate than we now considering and even little influence of surface pH on our proposed reaction scheme would have. Therefore, persistent growth of sulfate shows a negligible effect on $CO_3^-$ initiated $SO_2$ oxidation scheme proposed in this work.

Additionally, dust particles are reported to eject the radical ions from the surface under solar light irradiation, severing as an underlying pathway for sulfate aerosol formation in the atmosphere (Chen et al., 2021; Dupart et al., 2012), as described as:

Mineral Dust + $hv \rightarrow$ ROS (g)                                                                                      [R13]

ROS (g)+ humidified Air+ $SO_2 \rightarrow$ Sulfate(g)                                                              [R14]

Where ROS (g) stands for the active intermediates in the gas phase. Over 400 ppm of $CO_2$ is universal in the atmosphere, and it is expected to form (bi)carbonate ions once enters the quasi-liquid layer of humified particles. Bi(carbonate) ions are then prone to react with hydroxyl radical ions to form carbonate radicals. Following this line of reasoning, we attempt to monitor the plausible gas ROS species that are formed in the presence of $CO_2$ (see detailed discussion in the experimental section).

When $CO_2$ (atmospheric relevant concentration) is introduced into the homemade flow-cell chamber, with the intervening gap between $TiO_2$-coated film and probe molecule solution fixing at nearly 2 mm, the short distance of which guarantees gaseous ROS to diffuse and react with aniline (None, 2013). An increased degradation rate of this probe molecule was seen, which can be speculated to be the generation of active carbonate radical ions (Fig. 9). Aqueous $CO_3^-$ is believed to be supplied by partitioning processes from $CO_3^-$(g) that comes from humified dust particles, and its maximum steady-state concentration was determined to be $1.39 \times 10^{-13}$ M for "$TiO_2+Air+CO_2$" system, which is over 1.8 orders of magnitudes higher than that of ·OH for "$TiO_2+Air$" system ($2.15 \times 10^{-15}$ M). This observation matches with the early study where the concentration of carbonate radical can be nearly two orders of magnitudes than ·OH over the water surface (Sulzberger et al., 1997).

Overall, the above results suggest that the photochemistry that involves carbonate ions, more precisely $CO_3^-$ radicals, increases sulfate production. This finding broadens the prevailing view that acceleration of $SO_2$ oxidation over the carbonate salt is merely due to the favorable neutralization of $H_2SO_4$ over an alkaline surface. To be important, upon irradiation active component $TiO_2$ in mineral dust produce carbonate radicals in the gas phase when $CO_2$ presents, therefore potentially promoting sulfate aerosol formation in the atmosphere. Overall, it could be speculated that (bi)carbonate species strengthen the oxidative capacity of $TiO_2$-containing dust particles with regard to $SO_2$ oxidation.

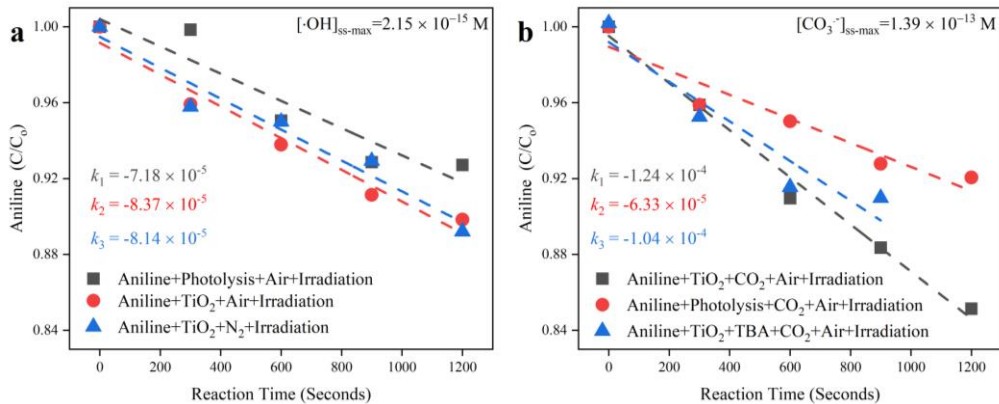

**Figure 9.** The degradation rate of aniline after exposure to air flow under irradiation in the absence **(a)** and presence **(b)** of $CO_2$ over mineral dust proxy particles $TiO_2$ as function of the reaction time. Reaction conditions: RH = 30%, Light intensity (I) = 30 mW cm$^{-2}$, Total flow rate = 52.5 mL min$^{-1}$.

### 3.2.4 Field Measurements of Sulfate and (Bi)carbonate Ions.

Complement field sampling and analysis were further conducted to examine our hypothesis that intermediates $CO_3^{\cdot-}$ may play role in secondary sulfate formation in the atmosphere. We first considered the meteorological condition wind speed, which is an important parameter determining whether the local chemical process gains importance in affecting secondary sulfate formation. Meteorological information was collected from the open-access database (https://www.aqistudy.cn/). During the sampling period, the wind scale mainly varies from 0 to 1, corresponding to the wind speed ranging from 0 to 1.5 m s$^{-1}$ (Fig. S13). All plots shown in Fig. S13 give rise to a statistical wind speed of 0.76 ± 0.73, which represents the weak dispersion of pollutants at low wind speed (not exceeding 2.5 m s$^{-1}$)(Liu et al., 2020a; Witkowska et al., 2016), indicating that local source is a dominant contributor to the air pollution.

Under stagnant meteorological conditions (wind speed < 1.5 m s$^{-1}$), for the coarse-mode (2.5 μm ~ 10 μm) of sulfate, the heterogeneous reaction of $SO_2$ on the dust surfaces is thought to be a major contributor (Liu et al., 2017). This correlates to the fact that a large mass fraction of mineral dust is abundant in coarse-model particulate matter (PM) (Fang et al., 2017; Miller-Schulze et al., 2015), in which $TiO_2$ was found at mass mixing ratios ranging from 0.1 to 10% depending on the exact location where particles were uplifted (Chen et al., 2012; Hanisch and Crowley, 2003). Therefore, PM with relatively larger size dimensions is expected to contribute to secondary sulfate formation via heterogeneous reactions, which is supported by the recent field study where the carbonate fraction of coarse PM is evidenced to promote secondary sulfate production (Song et al., 2018). Considering this, rather than determining the concentration of water-soluble ions in all stages, special attention is focused on PM collected in stages 1-4 (particles with their dimension ⩾ 3.3 μm). As (bi)carbonate ions are known as key precursors in producing $CO_3^{\cdot-}$ and accelerating sulfate formation, quantifications of those relevant water-soluble ions were thus conducted (see details in the experimental section and supplementary text 13).

We further consider the relationships between sulfate ions and (bi)carbonate ions by means of linear regression analysis. However, under the low wind speed ($0.76 \pm 0.73$), correlation coefficients $R^2$ obtained for the relationship between bi(carbonate) and sulfate ions are not promising, 0.56 (sulfate vs carbonate) and 0.61 (sulfate vs bicarbonate) for $PM_{3.3}$-$PM_{9.0}$ during daytime hours (Fig. 10). A plausible explanation is that in spite of little significance, local primary emission source also brings bias and uncertainty to the correlation results. Shanghai is a coastal city, and sulfate species such as $K_2SO_4$ and $Na_2SO_4$

from the sea salt contribute to the local sulfate emission as well (Long et al., 2014). On the other hand, this novel $SO_2$ oxidation channel is yet to be in the infant stage, and only active mineral dust components have been considered in this work whereas other components found in the coarse mode of PM such as organic matter, elemental carbon as well as sea salt (Cheung et al., 2011) are likely to involve this mechanism and alter the response of sulfate yield to $SO_2$ heterogeneous uptake. In addition, the water-soluble ions determined in these samples may not come from the net contribution of heterogenous reaction processes

in absolute daytime and nighttime periods. In other words, some of the collected samples, experiencing heterogeneous reaction that occurs during day(nigh)-night(day) shifts periods, inevitably being assigned to the sulfate ions measured in separate sampling hours, thus reducing the correlation coefficients.

        For those large particles (LP), which refer to the particles with a diameter large than 9 µm in this work, sulfate ions show a rather weak or even no correlation to (bi)carbonate ions during the nighttime and day-time hours (Fig. S14). This is likely due

to the short lifetime of LP. Generally, the aerosol lifetime is on the order of less than an hour to days (Koelemeijer et al., 2006), highly depending on particle size. For example, the lifetime of $PM_{10}$ ranges from minutes to hours, and its travel distance, in general, is less than 10 km (Agustine et al., 2018). As a consequence, secondary sulfate formation through chemical reaction over LP is not significant with respect to *in situ* emissions. When PM downsizes to 2.5 µm, $PM_{2.5}$ has a lifetime prolonged to nearly one day or longer (Wu et al., 2020). Therefore, $PM_{3.3}$-$PM_{9.0}$ are expected to have a relatively long lifetime, on the order

of several hours on average, which enables the heterogeneous reaction process to become a more important contributor to overall sulfate measured in $PM_{3.3}$-$PM_{9.0}$ than that in $PM_{\geq 9.0}$. This is supported by our observations where during the daytime hours the correlation coefficients for $PM_{3.3}$-$PM_{9.0}$, i.e. 0.56 (sulfate vs carbonate) and 0.61 (sulfate vs bicarbonate), are higher than that of $PM_{\geq 9.0}$, i.e. 0.489 (sulfate vs carbonate) and 0.36 (sulfate vs bicarbonate), respectively. Likewise, higher correlation coefficients are also observed for $PM_{3.3}$-$PM_{9.0}$ than $PM_{\geq 9.0}$ in the sample collected during the nighttime periods.

While we note that the correlations between sulfate and (bi)carbonate are not high in this work, ground-based field measurements of sulfate and (bi)carbonate ions shed light on their distinct correlations during the daytime and nighttime hours. In Fig. 10 and Fig. S14, the negative correlations between the mass concentrations of sulfate ions and (bi)carbonate ions are observed in the nighttime hours, consistent with the suppression of sulfate formation by $CO_2$ in the dark experiments. This is also supported by lab-based observations where $CO_2$-derived (bi)carbonate species are demonstrated to suppress sulfate

production over two dominant mineral dust components aluminum oxide (Liu et al., 2020c) and silicon dioxide (Fig. S15 and supplementary text 23). Alternatively, while $CO_2$-derived (bi)carbonate may slightly affect sulfate accumulation over PM with high water content in the dark scenario, fresh PM is usually dry when emitted into the atmosphere. Due to the competitive

adsorption, the occurrence of suppression of $SO_2$ adsorption and subsequent sulfate formation is possible in the early emission stage before PM becomes wet, thus contributing to the overall negative correlation.

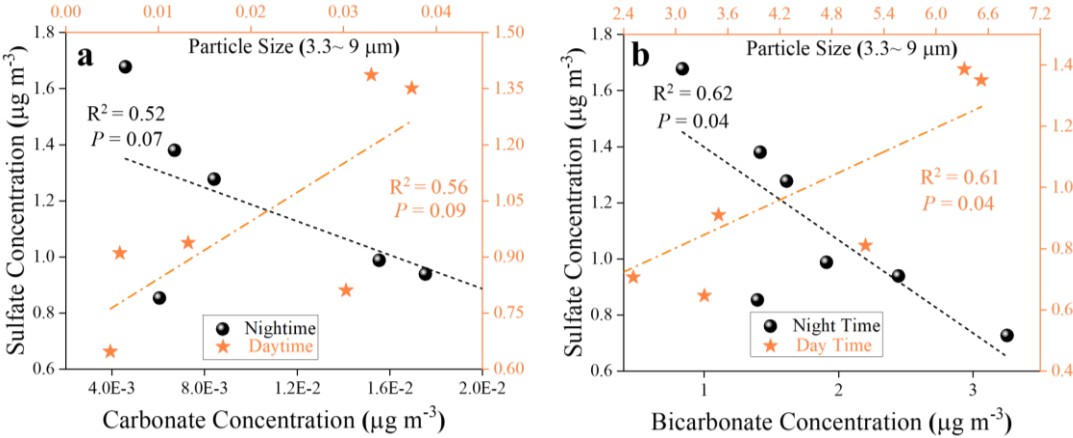


**Figure 10.** Field observation for the relationship between carbonate and sulfate ions during daytime and night-time hours. Linear relationship analyses for measured sulfate ions and estimated carbonate ions (**a**) and for measured sulfate ions and estimated bicarbonate ions (**b**) during the daytime and night-time hours, with particle sizes of PM ranging from 3.3 to 9 μm.

Instead, positive correlations were seen for those ions within PM sampled during the daytime hours regardless of size ranges

and carbonate types ($HCO_3^-/CO_3^{2-}$). This matches with the scenarios in which sulfate production upon irradiation in the presence of (bi) carbonate ions is increased over both model and authentic dust particles. Except the case (nighttime period, size larger than 9 μm), most of the significance $P$ values for their correlations were smaller than 0.1, specifically with significance $P$ values below 0.5 determined for bicarbonate vs sulfate, implying the plausible underlying connection between sulfate and (bi)carbonate ions. In fact, preceding ground-based observations of a highly correlated relationship between $Ca^{2+}$

and $SO_4^{2-}$ water-soluble ions (Wu et al., 2020) during the carbonate-enriched dust storm episodes, together with persistent reports on the significant role of photochemical channels in elevating the sulfate concentration level during the daytime hours (Kim et al., 2017; Wei et al., 2019; Wu et al., 2017) indirectly reflects the possibility of accelerated $SO_2$ oxidation triggered by photo-generated active intermediates associated with carbonate species.

Overall, this is the first time that relationships between those ions are explored separately in these two periods. Taken

together, carbonate radical is likely to promote sulfate production in the atmosphere during daytime hours. Detailed and systematic $SO_2$ oxidation channel triggered by $CO_3^-$ needs further investigations to enable a better interpretation of correlations between these inorganic ions at the given meteorological conditions of sampling and physicochemical properties of PM.

## 4. Conclusion

On the basis of the experimental and theoretical results derived from this work, we for the first time propose a novel reaction

channel for fast $SO_2$ oxidation over mineral dust particles due to the formation of carbonate radical ions. A schematic chart for

the sulfate formation in the presence of carbonate radicals upon solar light or bi(carbonate) ions under dark conditions is summarized and elucidated in Fig. 11. During the nighttime hours at 298 K (ambient temperature) $CO_2$-derived (bi)carbonate species are prone to have a slightly negative effect on sulfate formation presumably due to the competitive adsorption between $CO_2$ and $SO_2$. For alkaline carbonate salt, it favors sulfate formation through the neutralization process. On the other hand, in

the daytime, both $CO_2$-derived (bi)carbonate species and carbonate salt work as the precursor of $CO_3^{\cdot-}$, which promotes sulfate formation. Especially, uptake coefficients for carbonate salt containing mineral dust can be increased by 17 times, which is more pronounced than the increase due to the neutralization regime in the dark condition. Consistent with the findings reported in the early studies (Chen et al., 2021; Dupart et al., 2012), we speculate the production of gas-phase $CO_3^{\cdot}$ ions when mineral dust particles are irradiated in the presence of $CO_2$ (atmospherically relevant concentration 400 ppm). This observation

potentially implies that the increased sulfate yield in part comes from increased external secondary sulfate aerosol triggered by $CO_3^{\cdot-}$ (g).

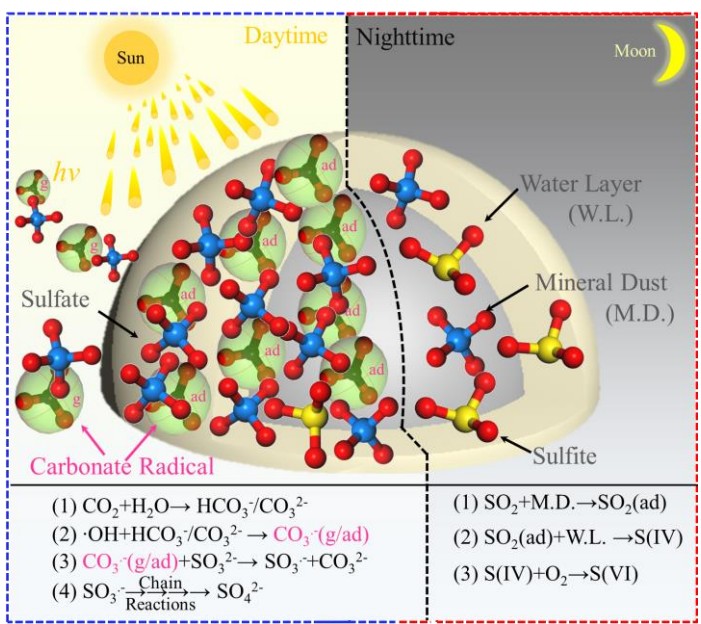

**Figure 11.** Schematic of the sulfate formation in the presence and absence of carbonate radical. Noting that g and ad represent gas-phase and adsorbed carbonate radical ions, respectively.

By means of ROS scavenger experiments, direct observation of carbonate radical using NTAS analysis, oxygen isotope assay, ESR measurement as well as DFT calculations, $CO_3^{\cdot-}$-initiated S(IV) oxidation involving single electron transfer process are elucidated. While carbonate radical ions are mainly responsible for rapid sulfate formation, superoxide radical ions are likely to serve as a minor pathway over $TiO_2$-containing mineral dust particles (not shown on the schematic chart to direct readers' focus on $CO_3^{\cdot-}$). In addition, a weak correlation between sulfate ions and (bi)carbonate ions observed for $PM_{3.3}$-$PM_{9.0}$

in this work reasonably correlates to non-chemical primary emission and the complicated nature of $CO_3^{\cdot-}$ regime of sulfate

production in the atmosphere. Nonetheless, complement field sampling of ambient PM and analysis of sulfate and (bi)carbonate ions in this study unfold their distinct correlations during the daytime and nighttime hours, these two tendencies of which agree with the experimental observations.

In this work, only atmospheric secondary sulfate formation was considered, whereas the oxidation of primary organic species yet has not been investigated. In fact, carbonate radical ions are prone to rapidly react with electron-rich organics amines (Stenman et al., 2003; Yan et al., 2019) as well as phenol (Busset et al., 2007; Xiong et al., 2016), and it may potentially serve as the key oxidants that drive the fast formation of SOA in the atmosphere. Besides, observation of strengthened photochemistry launched by carbonate radicals suggests that such chemistry may be amplified on atmospherically relevant reactions that occur in cloud droplets as well as fog water where they often contain hydroxyl radicals and water-soluble (bi)carbonate ions.

To be important, gas-phase carbonate radical ions are speculated to be formed in the atmospherically relevant $CO_2$ concentration (400 ppm) when mineral dust is irradiated. This will help the formation of external sulfate aerosol formation. Since both sulfate aerosol and $CO_2$ are well known to regulate the radiation budget and solar energy balance on the earth (Cheung et al., 2011; Möller, 1964), coupled with the $CO_2$-intiated promoted sulfate pathway found here, their overall influence on the global climate needs further investigation. Therefore, our study highlights the necessity for a comprehensive understanding of the $CO_3^{-}$ relevant chemistry in the underlying impacts of fine PM concentration, human health, and climate. All these assumptions need to be investigated in further detail. This study provides the indication that carbonate radical not only plays a role as a marginal intermediate in tropospheric anion chemistry but also as a strong oxidant for surfacial processing of trace gas in the atmosphere.

*Data availability.* The data that support the results are available from the corresponding author upon request.

*Author contributions.* Y.L., Y.D. and L.Z. initially proposed the idea; Y.L. and Y.D. together designed the experiments, and Y.L. performed the most of the experiments; J.L. performed DFT calculations; Y.L., X.Z. and T.W. contributed to field samplings and data analysis; K.L., K.G., A.B., I.N., Q.G., X.Z., C.G., and L.Z. provided suggestions on the experiments and paper writing; All authors wrote the manuscript.

*Competing interests.* The authors declare that they have no conflict of interest.

*Acknowledgements.* We greatly appreciate Dr. Yang Yang and Prof. Keli Han from Dalian institute of chemical physics for NTAS and some helpful discussions.

*Financial support.* This work was supported by the National Natural Science Foundation of China (No. 21976030 and No. 21677037), National key research and development program of China (2016YFE0112200 and 2016YFC0202700), the Natural Science Foundation of Shanghai (No. 19ZR1471200 and No. 17ZR1440200).

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
