# Peer review of "A Novel Pathway of Atmospheric Sulfate Formation Through Carbonate Radical"

_Atmospheric Chemistry and Physics, 2021_

## Referee Comment (RC1)

Comments on ACP-2021-564

In this work, Liu et al. studied the formation of sulfate on mixed mineral dust particles and found a synergistic effect between $TiO_2$ and carbonate in promoting sulfate formation upon illumination. They proposed a novel mechanism in which carbonate radical ($\cdot CO_3^-$) was considered as an important intermediate. This $\cdot CO_3^-$ was assumed to oxidize $SO_3^{2-}$ to $\cdot SO_3^-$ and then promote sulfate formation. Lots of methods were used to prove the existence of $\cdot CO_3^-$ and its interaction with other species in this reaction system. Furthermore, analysis of samples collected in field observation and quantum chemical calculation were also used to show this synergistic effect between $TiO_2$ and carbonate in promoting sulfate formation in the atmosphere. The formation mechanism of sulfate is an important research topic in atmospheric chemistry as well as the occurrence of high concentration of fine particles during haze episodes. This work provided a new and interesting perspective for synergistic effect in the formation of atmospheric sulfate aerosol. However, the flaws in the hypothesis of reaction mechanism make its scientific and environmental significance questionable. In addition, the manuscript is not well organized and a little hard to read. So, I think it may not be accepted in current version.

Main concerns.

1. It is not reasonable to exclude the buffering effect of carbonate in promoting the formation of sulfate. The results that $CaCO_3$ did not promote the formation of sulfate under dark conditions (Page 6 line 155) can not extend to confirm its effect in promoting sulfate formation under illumination. It has been well known that the conversion of $SO_2$ to S(VI) hardly happen under ambient conditions without strong oxidants or illumination. On the other hand, several recent studies reported that mineral dust photochemistry can induce the formation of $H_2SO_4$ (PNAS, 2012, 109, 20842–20847; EST, 2021, 55, 14, 9784-9793). In the present study, it seems the increase in sulfate concentration under illumination condition is most likely due to the enhanced condensation or neutralization of $H_2SO_4$ in the presence of $CaCO_3$ or carbonates. I think

the $CO_2$+$SO_2$ experiments may not rule out the buffering effect of carbonate effect in $TiO_2$+$CaCO_3$ system since the enhancement effect of $CO_2$ is significantly lower than that on $CaCO_3$. Moreover, why $SO_2$ concentration in Figure 2 is quite lower than those in Figure 1? And could you show the IR peaks due to $CO_2$ adsorption? Did their intensities change during the photooxidation of $SO_2$?

2. Consistency and comparability of experimental system. The heterogeneous reactions of $SO_2$ were studied on mineral particles while some characterization experiments for supporting evidences were conducted in solutions. Although some water layers may be formed on mineral dust particle surface at 30%RH, however, this situation may far from the liquid state. So, it is unreasonable to assume that all reaction mechanisms are ionic reactions in liquid phase.

3. mechanism: Firstly, only hole was consumed which resulted in the formation of •OH and •$CO_3^-$ (eq 2 and 3). However, the consumption of photogenerated electron is not mentioned. According to eq 5-8, $O_2$ is the key oxidant for the oxidation of $SO_3^{2-}$ to sulfate. However, the content of $O_2$ in solution is limited due to its low solubility if aqueous reactions were assumed. In addition, $O_2$ can react with photogenerated electron to form $O_2^-$ and then oxidize $SO_3^{2-}$ on particles surface. The authors need to compare the effect of these two processes on the formation of sulfate. Secondly, according to eq 6-8 (in eq 7, $SO_4$ should be $SO_4^{2-}$?), it seems that the oxidation of $SO_3^{2-}$ by $O_2$ could be a catalytic reaction while •$SO_3^-$ acted as catalyst. If so, the amounts of sulfate formed through photooxidation on $TiO_2$ should be the same (at least close to) in the presence of carbonates and $CO_2$ since •$CO_3^-$ only contribute to the formation of •$SO_3^-$. Is it? Thirdly, what's the pH effect on the reaction? As seen in eq 3 and 4, only the reaction of •OH with $HCO_3^-$/$CO_3^{2-}$ was considered. What's about the reactions between $H^+$ and $HCO_3^-$/$CO_3^{2-}$? As the oxidation of $SO_2$ or sulfite increased, the pH should decrease and then affect $HCO_3^-$/$CO_3^{2-}$.

Other concerns:

1. The concentrations of $SO_2$ used are much higher than the ambient atmospheric concentration.

2. Line 228: it is difficult to understand this sentence "ESR data (Fig. 3d) further confirms the increase of $SO_3^{\cdot-}$ after 2 min UV irradiation in the presence of carbonate ion" since the change is not very obvious.

3. sampling in field observation. The samples collected in daytime and nighttime did not mean they are always in dark and illuminated conditions. The samples collected in nighttime may also have undergone multiple daytime photochemical processes. In addition, as proposed by Sullivan et al. (Atmos. Chem. Phys., 7, 1213–1236, 2007), oxidation of S(IV) to S(VI) by iron in the aluminosilicate dust is a possible explanation for the enrichment of sulphate in Asian mineral dust. So, how to exclude the effect of Fe in this study?

4. Conclusion and atmospheric implications: this study only found the enhanced sulfate formation in mixed $TiO_2$ and $CaCO_3$ particles compared to individual $TiO_2$ or $CaCO_3$ particles. However, the hypotheses of $CO_2$-derived carbonate species and carbonate salt works as the precursor of $\cdot CO_3^-$ is exaggerated. As seen in this study, $TiO_2$ is necessary but its content in atmospheric particulate matter is very low. Considering the unclear role of $\cdot CO_3^-$, as well as the high concentration of $SO_2$ used, its implications even on sulfate formation is limited. Consequently, the extension of its atmospheric implications to fine PM concentration, human health, and climate is not meaningful.

---

## Author Comment (AC1)

**Response Letter (For Referee 1)**

**Comment from Referee 1:**

"In this work, Liu et al. studied the formation of sulfate on mixed mineral dust particles and found a synergistic effect between $TiO_2$ and carbonate in promoting sulfate formation upon illumination. They proposed a novel mechanism in which carbonate radical ($\bullet CO_3^-$) was considered as an important intermediate. This $\bullet CO_3^-$ was assumed to oxidize $SO_3^{2-}$ to $\bullet SO_3^-$ and then promote sulfate formation. Lots of methods were used to prove the existence of $\bullet CO_3^-$ and its interaction with other species in this reaction system. Furthermore, analysis of samples collected in field observation and quantum chemical calculation were also used to show this synergistic effect between $TiO_2$ and carbonate in promoting sulfate formation in the atmosphere. The formation mechanism of sulfate is an important research topic in atmospheric chemistry as well as the occurrence of high concentration of fine particles during haze episodes. This work provided a new and interesting perspective for synergistic effect in the formation of atmospheric sulfate aerosol. However, the flaws in the hypothesis of reaction mechanism make its scientific and environmental significance questionable. In addition, the manuscript is not well organized and a little hard to read. So, I think it may not be accepted in current version."

**Author general reply:**

Thanks for your valuable suggestions, which greatly helped us to improve the manuscript. According to your comments, we have noted the flaws and shortcomings in the argument, especially for the controversial role of carbonate ions in sulfate formation under irradiation. We thus supplied a series of experiments to further improve and modify the reaction scheme proposed in the previous submission, and revised the manuscript to provide more convincing explanations to the readers. Besides, the role of superoxide radical ions, more precisely the sink of photo-generated electrons and its contribution to sulfate formation relative to carbonate radical ions have been discussed in the revised version of the manuscript. Further, we supplied a more detailed discussion to connect each paragraph on a common string of reasoning, which will guide the readers to capture the whole picture of the manuscript and to get the take-home message. We carefully consider all your comments posted to the previous version of the manuscript, and the detailed point-by-point revisions are presented as follows.

Main concerns.

**$Q_1$-A:** It is not reasonable to exclude the buffering effect of carbonate in promoting the formation of sulfate. The results that $CaCO_3$ did not promote the formation of sulfate under dark conditions (Page 6 line 155) can not extend to confirm its effect in promoting sulfate formation under illumination. It has been well known that the conversion of $SO_2$ to S(VI) hardly happen under ambient conditions without strong oxidants or illumination. On the other hand, several recent studies reported that mineral dust photochemistry can induce the formation $H_2SO_4$ (PNAS, 2012, 109, 208842–20847; EST, 2021, 55, 14, 9784-9793). In the present study, it seems the increase in sulfate concentration under illumination condition is most likely due to the enhanced condensation or neutralization of $H_2SO_4$ in the presence of $CaCO_3$ or carbonates.

**Response to $Q_1$-A:** Thanks for your question. We note that dark experiments may not fully rule out the possibility where photo-generated oxidants accelerate sulfate formation due to the favorable neutralization over alkaline surfaces. Therefore, several supplementary experiments were performed to validate our findings, as shown below:

**1. $SO_2$ oxidation over $TiO_2+CaCO_3$ mixture and $TiO_2+CaO$ mixture**

Indeed, it remains puzzle for the role of carbonate salt in sulfate formation either by favoring neutralization of $H_2SO_4$ on alkaline surfaces or by serving as the precursor of active $CO_3^-$ to trigger the fast sulfate formation as we proposed in this work. Therefore, we employed two types of mixtures $TiO_2+CaCO_3$ and $TiO_2+CaO$. According to the EDS mapping analysis of relevant component contents of the Arizona test dust (ATD) (Table S1), the mass fraction ratio of $TiO_2$ to $CaCO_3/CaO$ are thus fixed at 1:7 to mimic the synergistic effect that is likely to take place over the authentic dust particles. In the dark experiments, both $TiO_2+CaO$ and $TiO_2+CaCO_3$ almost yield identical concentration levels of sulfite and sulfate, indicating that they show similar surface properties, e.g. alkalinity and the number of surface sites. Once irradiated, $TiO_2+CaCO_3$ particles produce nearly two times of sulfate than $TiO_2+CaO$ particles, along with a sharp decrease of S(IV) species on the surface of $TiO_2+CaCO_3$ surfaces. This result confirms the existence of the formation of active intermediates that drive the fast $SO_2$ oxidation when carbonate salt is presented. In addition, the total sulfur content, i.e. S(IV)+S(VI), in $TiO_2+CaCO_3$ particles is quite higher than that in $TiO_2+CaO$ particles upon irradiation, whereas they are almost identical in the dark experiments. Consequently, the difference between two mixtures regarding sulfate yield under illumination can be mainly attributed to the formation of additional reaction channels that have been not previously considered. Another plausible explanation is that carbonate radical ions promote sulfate formation by forming the carbonate radical in the gas phase (see results in Fig. R3) and thus yield more secondary sulfate aerosol in the gas phase. Part of them will then condense back onto particle surfaces to increase total sulfate yield.

[Figure]

**Fig. R1.** Sulfate and sulfite concentration quantified by IC on mineral dust proxies after exposure

to gaseous $SO_2$ under irradiation or dark for 20 min. Reaction conditions: RH = 30 %, Light intensity (I) = 30 mW cm$^{-2}$, Total flow rate = 52.5 mL min$^{-1}$ and $SO_2$ = 2.21×10$^{14}$ molecules cm$^{-3}$.

***Detailed correction in the manuscript:***

*"While great discrepancies in sulfate yield between dark and irradiation experiments, it remains unclear for the role of carbonate salt in promoting sulfate formation. There is a prevailing view that neutralization of $H_2SO_4$ accounts for rapid $SO_2$ oxidation over carbonate salt particles, which needs careful consideration. Following this speculation, two types of mixtures $TiO_2$-$CaCO_3$ and $TiO_2$-$CaO$ were employed. In the dark condition (Fig. S3), both $TiO_2$-$CaO$ and $TiO_2$-$CaCO_3$ almost yield an identical concentration of sulfite and sulfate as they are likely to present similar physical and chemical properties, e.g. surface pH and neutralization capability. Once irradiated, $TiO_2$-$CaCO_3$ particles produce nearly two times of sulfate than $TiO_2$-$CaO$ particles, along with a sharp decrease of S(IV) species on the surface of $TiO_2$-$CaCO_3$ surfaces (see additional discussion in the supplementary text 2). These results allow us to assert that the carbonate-containing system contains another important mechanism for sulfate generation beyond the production of an alkaline environment."* ***(Main Text, Page 4, Line 108-117)***

**2. Observation of strengthend oxidation capbility in carbonate-containing $TiO_2$ suspension**

To validate the formation of carbonate radical strengthening oxidative capacity of carbonate-containing $TiO_2$ particles under irradiation, aniline is used as probe molecular, which is reported to have a high reaction rate with carbonate radical ions ($k._{OH,aniline}$ = 6.5×10$^9$ M$^{-1}$s$^{-1}$) (Samuni et al., 2002). In Fig. R2, a difference between "air + irradiation" system and "$TiO_2$ + air + irradiation" system mainly comes from the contribution of hydroxyl radical instead of from other intermediates (e.g. superoxide radical, see discussion in the next subsection "Determination of gas-phase ROS production in the flow-cell reactor"). When carbonate ions are introduced into $TiO_2$ suspension, leaving the pH of the reaction system at 11, the removal rate of aniline is evidently increased. We noted that increasing pH favors the formation rate of ·OH radical, which has been well verified in numerous works (Chavadej et al., 2008; Kansal et al., 2008; Tang and An, 1995). To examine the net contribution of carbonate radical ions to the increased oxidation capability of the carbonate-containing $TiO_2$ system beyond the increased pH environment, we performed the reference experiment using "$TiO_2$+ air +NaOH" reaction system. In detail, an adequate amount of NaOH was added into $TiO_2$ suspension to have the pH of $TiO_2$ suspension identical to that of $TiO_2$+$Na_2CO_3$. Indeed, it shows a higher removal rate than "$TiO_2$+ air" but a lower removal rate than "$TiO_2$+ $Na_2CO_3$+air". It seems to suggest that higher alkaline carbonate salt favors in promoting sulfate formation in part due to the increased OH yield. Nevertheless, one should note that carbonate radical ions are predominant species in a relatively alkaline environment in the presence of carbonate since the carbonate ion is an excellent ·OH scavenger.

The previous work (Sun et al., 2016) shows that adding 0.1 M of $NaHCO_3$ into the UV/$H_2O_2$ system ($H_2O_2$ = 0.3 mM) were sufficient to suppress ·OH concentration to around 10$^{-15}$ M, creating a carbonate radical dominated reaction system ([$CO_3^-$] = 8.64 × 10$^{-12}$ M). In our supplementary experiments, 0.2 M of $Na_2CO_3$ was employed, and the reaction rate of $CO_3^{2-}$ with ·OH is over one order of magnitude higher than that with $HCO_3^-$, thus giving rise to carbonate radical being the substitute of hydroxyl radical in the reaction system and responsible for enhanced aniline degradation.

Overall, when (bi)carbonate ions are present, carbonate radical ions that are enriched on the surface of $TiO_2$ increase the overall oxidation capability of $TiO_2$ particles. Besides, the increase of pH favors the production of $CO_3^-$, further strengthening the oxidation capability of dust particles.

[Figure]

**Fig. R2.** The removal rate of aniline after exposure to air flow under irradiation in the absence and presence of mineral dust particles for 300 seconds. Reaction conditions: RH = 30 %, Light intensity (I) = 30 mW cm$^{-2}$, Total flow rate = 52.5 mL min$^{-1}$. Noting that an adequate amount of NaOH was introduced into $TiO_2$ suspension system to achieve a pH environment condition comparable to that of $TiO_2$-$Na_2CO_3$ suspension system.

***Detailed correction in the manuscript:***

*"A promoted degradation of aniline in $TiO_2$ suspension due to presence of carbonate ions presents additional evidence of the formation of active $CO_3^-$ ions and strengthened oxidative capability of $TiO_2$ (Fig. S11, see more discussion in supplementary text 4)." **(Main Text, Page 10, Line 246-248)***

[revised manuscript text omitted]
$_3^{2-}$ over TiO$_2$ film and gaseous water molecular in the humidified air flow, the maximum steady-state CO$_3^{.-}$ radical concentration was determined to be 1.39 × 10$^{-13}$ M for TiO$_2$+Air+CO$_2$ system, matching well with the earlier study where the concentration of carbonate radical can be two orders of magnitudes than ·OH over the water surface (Sulzberger et al., 1997b).

The above supplementary results suggest that the photochemistry that CO$_3^{.-}$ radical increases sulfate formation. This finding broadens the previous prevailing view that acceleration of SO$_2$ oxidation over the carbonate salt is merely due to the favorable neutralization of H$_2$SO$_4$ over alkaline surfaces. To be important, upon irradiation active component TiO$_2$ in mineral dust will produce carbonate radical in the gas phase when the atmospherically relevant concentration of CO$_2$ is presented, therefore potential promoting sulfate aerosol formation in the atmosphere. Overall, it could be deduced that carbonate radical ions strengthen the oxidative capability of dust particles TiO$_2$, and consequently accelerate SO$_2$ oxidation.

[Figure]

**Fig. R3.** The degradation rate of aniline after exposure to air flow under irradiation in the absence (a) and presence (b) of $CO_2$ over mineral dust proxy particles $TiO_2$ as function of the reaction time. Reaction conditions: RH = 30 %, Light intensity (I) = 30 mW cm$^{-2}$, Total flow rate = 52.5 mL min$^{-1}$.

***Detailed correction in the manuscript:***

*"Additionally, dust particles are reported to eject the radical ions from the surface under solar light irradiation, showing a non-negligible contribution to sulfate aerosol formation (Chen et al., 2021; Dupart et al., 2012), as described as:*

*Mineral Dust + hv → ROS (g)*                                              *(Eq.14)*

*ROS (g)+ humidified Air+ $SO_2$→Sulfate(g)*                         *(Eq.15)*

*Where ROS (g) stands for the active intermediates in the gas phase. Over 400 ppm of $CO_2$ is in the atmosphere, and it is expected to form (bi)carbonate ions once enters into the atmospheric aqueous media such as aerosol water, cloud droplets as well as fog environment. Bi(carbonate) ions are then prone to react with hydroxyl radical ions to form carbonate radical ions. Following this line of reasoning, we attempt to monitor the plausible gas ROS species that are formed in the presence of $CO_2$ (see detailed discussion about the measurement approach and experimental setup in supplementary text 7 and Fig. S16).*

*When $CO_2$ (atmospheric relevant concentration) is introduced into the homemade flow-cell chamber, with the intervening gap between $TiO_2$-coated film and probe molecule solution fixing at nearly 2 mm, and a short distance of which allows possible gaseous ROS to diffuse and react with aniline molecular (None, 2013). An increased degradation rate of aniline was seen, which can be attributed to the generation of active carbonate radical ions (Fig. S17). The maximum concentration of steady-state $CO_3^-$ radical ions supplied by partition processes between gas phase and solid-liquid phases (humified dust particles) was determined to be $1.39 \times 10^{-13}$ M for the $TiO_2+Air+CO_2$ system, which is over one order of magnitudes higher than that of ·OH for the $TiO_2+Air+$system ($2.15 \times 10^{-15}$ M). This observation matches with the earlier study where the concentration of carbonate radical*

*can be two orders of magnitudes than ·OH over the water surface (Sulzberger et al., 1997a).*

*The above results suggest that the photochemistry that involves carbonate ions, more precisely $CO_3^-$ radical, increases sulfate formation. This finding broadens the prevailing view that acceleration of $SO_2$ oxidation over the carbonate salt is merely due to the favorable neutralization of $H_2SO_4$ over an alkaline surface. To be important, upon irradiation active component $TiO_2$ in mineral dust produce carbonate radical in the gas phase when $CO_3^-$ precursor $CO_2$ is presented, therefore potentially promoting sulfate aerosol formation in the atmosphere. Overall, it could be deduced that carbonate radical ions strengthen the oxidative capability of $TiO_2$-containing mineral dust particles, and consequently accelerates $SO_2$ oxidation." **(Main Text, Page 14, Line 365-387)**

**Q$_1$-B:** I think the $CO_2$+$SO_2$ experiments may not rule out the buffering effect of carbonate effect in $TiO_2$+$CaCO_3$ system since the enhancement effect of $CO_2$ is significantly lower than that on $CaCO_3$. Moreover, why $SO_2$ concentration in Figure 2 is quite lower than those in Figure 1? And could you show the IR peaks due to $CO_2$ adsorption? Did their intensities change during the photooxidation of $SO_2$?

**Response to Q1-B**: Thanks for your question. As reported by the previous study (Czapski et al., 1999), the reaction rate of hydroxyl radical with carbonate ions is nearly one order magnitude higher than that with bicarbonate ions. Therefore, carbonate salt can produce much more carbonate radical ions than $CO_2$ does, and consequently promote $SO_2$ oxidation more evidently.

[Figure]

**Fig. R4.** Time-resolved DRIFTS of carbonate products over $TiO_2$ particles after exposure to $SO_2/N_2+O_2$ in the presence of $CO_2$ upon irradiation. Reaction conditions: RH = 30 %, Light intensity (I) = 30 mW cm$^{-2}$, total flow rate = 52.5 mL min$^{-1}$ and $SO_2$ = $7.37×10^{13}$ molecules cm$^{-3}$.

In Figure 1, we employed the HRTEM technique to analyze the sulfate distribution on the $TiO_2$-$CaCO_3$ particles to elucidate the reaction mechanism for the increased sulfate yield. While this benefits us to figure out the scheme behind the observation, one limitation appears for this methodology, that is a relatively high sulfate concentration required to overcome its relatively low detection limit. To address this issue, we elevated the $SO_2$ concentration for heterogenous reaction

to monitor sulfate feature in HETEM images, i.e. high resolution with lattice fringes, and we also carefully consider the influence of $SO_2$ concentration on the reaction kinetics by examining the reaction order of $SO_2$ on the dust particles. Within the range of 400-20000 ppb $SO_2$ (Fig. S10), the heterogeneous reaction of $SO_2$ on mineral dust particles follows the pseudo-first-order. Therefore, despite some inconsistency of $SO_2$ concentration applied in different reaction systems, it would not give fundamental influence on our findings and observations where the increased sulfate formation comes from the formation of carbonate radical ions due to the synergistic effect between $TiO_2$ and $CaCO_3$.

Time-resolved IR bands assigned to $CO_2$ adsorption over $TiO_2$ particle surface during the heterogeneous photo-oxidation of $SO_2$ have been presented in Fig. R4. For $TiO_2$ dust particles, the absorption at 1533 $cm^{-1}$ is specifically assigned to the vibrational modes of the adsorbed monodentate carbonate (Baltrusaitis et al., 2011; Nanayakkara et al., 2015). There is a decrease in the intensity in the O-C=O region as heterogeneous reaction proceeds during 180 min. Following Jiang's work (Jiang et al., 2019), we introduced the concept of "equivalent exposure time" in analyzing the DRIFTS data. The "equivalent exposure time" refers to the theoretical exposure time of $SO_2$ at an atmospherically-relevant concentration that $TiO_2$ and $TiO_2$-$CaCO_3$ particles are exposed to. The equivalent exposure time is calculated by multiplying the reaction time in the lab with the scale factor, which is the ratio of $SO_2$ concentration applied in DRIFTS experiments to $SO_2$ concentration possible in the atmospherically-relevant condition (20 ppb). Nevertheless, considering a large $SO_2$ concentration gap between the lab simulations and field observations, direct extrapolation of equivalent exposure time into such low $SO_2$ concentration may not be appropriate. Therefore, the reaction kinetics of $SO_2$ on mineral dust particles $TiO_2$ was investigated, and it is evidenced to follow the pseudo-first-order in the $SO_2$ concentration range of 400 ppb-20000 ppb, which covers all $SO_2$ concentrations applied in this study. While a concentration gap between lab studies and field observation remains, we tentatively assume this gap has a marginal impact on the kinetics considered. On this basis, we plotted integrated areas of the IR peak at 1615-1550 $cm^{-1}$ versus equivalent exposure time (Fig. R4b).

The mean lifetime of mineral dust particles in the atmosphere is nearly one week (Bauer and Koch, 2005). In our study, nearly one week of equivalent exposure of atmospheric $SO_2$ gives rise to the reduction of 4.9 % of carbonate species. Therefore, this moderate consumption of absorbed carbonate slightly affects the abundance of carbonate ions and the production of $CO_3^{\cdot-}$.

**$Q_2$:** Consistency and comparability of experimental system. The heterogeneous reactions of $SO_2$ were studied on mineral particles while some characterization experiments for supporting evidences were conducted in solutions. Although some water layers may be formed on mineral dust particle surface at 30 % RH, however, this situation may far from the liquid state. So, it is unreasonable to assume that all reaction mechanisms are ionic reactions in liquid phase.

**Response to $Q_2$:** Thanks for your question. We understand your concern about the consistency issue for reaction systems in the adsorbed water layer and aqueous media. On one hand, the experiments for carbonate radical detection were conducted in solution, which is unavoidable since it is a requirement for the nanosecond transient absorption technique and ESR measurements. One the other hand, at a sufficiently low RH condition (normally below 10 % RH), water readily dissociates on the surface of metal oxide under ambient atmospheric conditions, where metal oxide surface is terminated by hydroxyl groups (Cwiertny et al., 2008). In this case, $SO_2$ oxidation over dust particles

is dominated by the heterogenous pathway, where the resulting hydroxyl groups capture $SO_2$ in the gas phase first and then stabilize it to adsorbed $S(IV)_{ad}$ over dust surfaces. Afterward, $S(IV)_{ad}$ will be oxidized by oxidants in the atmosphere or photo-induced active intermediates produced from the dust surface upon irradiation.

As the RH increases beyond 10 % - 15 %, multilayer water coverage occurs, reaching approximately two monolayers at RH of 30 % (Mogili et al., 2006). Under these circumstances, the amount of water adsorbed onto the surface of the dust particles is believed to be sufficiently large that it is liquid-like in its physical and chemical properties (Cwiertny et al., 2008) (Peters and Ewing, 1997). In this work, heterogenous $SO_2$ oxidation over mineral dust proxies proceeds at the RH of 30 %, and two water layers absorb on dust particles. Thus, radical ions are expected to play a key role in fast $SO_2$ oxidation. Taken above, the mechanism studies performed in solution phase are persuasive to some extent.

***Detailed correction in the manuscript:***

*"At a sufficiently low RH condition (normally below 10 % RH), water readily dissociates on the surface of metal oxide under ambient atmospheric conditions, where metal oxide surface is terminated by hydroxyl groups that hydrogen bond to adsorbed water molecules (Cwiertny et al., 2008). In this case, $SO_2$ oxidation over dust particles is dominated by the heterogenous pathway, where the resulting hydroxyl groups capture $SO_2$ in the gas phase first and then stabilize it as adsorbed $S(IV)_{ad}$. Afterward, $S(IV)_{ad}$ will be oxidized by oxidants in the atmosphere or photo-induced active intermediates produced from the dust surface upon irradiation. As the RH increases beyond 10 % -15 %, multilayer water coverage occurs, reaching approximately two monolayers at RH of 30 % (Mogili et al., 2006). Under these circumstances, the amount of water adsorbed onto the surface of the dust particles is believed to be sufficiently large that it is liquid-like in its physical and chemical properties (Cwiertny et al., 2008) (Peters and Ewing, 1997). In this work, heterogenous $SO_2$ oxidation over mineral dust proxies proceeds at the RH of 30 %, and two water layers absorb on dust particles. Thus, radical ions are expected to play a key role in fast $SO_2$ oxidation and mechanism studies performed in solution phase are persuasive to some extent."*
***(Main Text, Page 9, Line 222-233)***

**Q$_3$-A:** Mechanism: Firstly, only hole was consumed which resulted in the formation of •OH and •$CO_3^-$ (eq 2 and 3). However, the consumption of photogenerated electron is not mentioned. According to eq 5-8, $O_2$ is the key oxidant for the oxidation of $SO_3^{2-}$ to sulfate. However, the content of $O_2$ in solution is limited due to its low solubility if aqueous reactions were assumed. In addition, $O_2$ can react with photogenerated electron to form $O_2^-$ and then oxidize $SO_3^{2-}$ on particles surface. The authors need to compare the effect of these two processes on the formation of sulfate.

**Response to Q$_3$-A:**
**The sink of $O_2$ and role of $O_2^{\cdot-}$**
Thank you for your valuable comments. We considered the sink of photo-generated electrons and the role of superoxide radical ions ($O_2^{\cdot-}$) in sulfate formation. In addition, we supplied the ·OH scavenger experiment to monitor the contribution of $O_2^{\cdot-}$ and ·OH to sulfate formation over $TiO_2$ particles. *p*-benzoquinone is a commonly-used $O_2^{\cdot-}$ scavenger for trapping the $O_2^{\cdot-}$ radical ions (Yan et al., 2018). Our supplementary data shows that adding an excess amount of *p*-benzoquinone into

$TiO_2$ particles reduces the sulfate yield by 32 % along with the appearance of sulfite ions found over $TiO_2$ particles upon exposure to $SO_2$. On the other hand, Tertiary Butyl Alcohol (TBA), ·OH scavenger, sharply decreases the yield of sulfate on $TiO_2$ surface by nearly 70 %, with sulfite ions being the dominant sulfur species. Notably, the decrease in sulfate yield by around 30 % in the presence of $O_2^{·-}$ scavenger *p*-benzoquinone is almost complementary to that added with ·OH scavenger using TBA (70 %), pointing toward a minor sulfate formation pathway contributed by $O_2^{·-}$ relative to the major pathway by $CO_3^{·-}$ when carbonate ions are presented to efficiently to capture ·OH ions. Nevertheless, $O_2^{·-}$ plays a non-negligible role in sulfate formation and should be incorporated to give a whole picture of the reaction scheme in triggering sulfate formation on the surface of $TiO_2$ particles. We have added this pathway into the main text.

[Figure]

**Fig. R5.** Determination of sulfite and sulfate concentration after exposure to air flow under irradiation in the absence and presence of mineral dust particles for 20 min. Reaction conditions: RH = 30 %, Light intensity (I) = 30 mW cm$^{-2}$, Total flow rate = 52.5 mL min$^{-1}$.

***Detailed correction in the manuscript:***

*"In addition to the pathway launched by photo-generated holes, we also considered the sink of photo-generated electrons. In our reaction system, $O_2$ is believed to be an electron trap and produce the superoxide radical ions ($O_2^{·-}$), which is reported to play a non-negligible role in sulfate formation (Shang et al., 2010a) and should be taken into account to give a whole picture of reaction scheme in triggering sulfate formation on the surface of $TiO_2$-containing mineral dust particles. p-benzoquinone is a commonly-used $O_2^{·-}$ scavenger for trapping the $O_2^{·-}$ radical ions (Yan et al., 2018). Our supplementary data shows that adding an excess amount of p-benzoquinone into $TiO_2$ particles reduces the sulfate yield by 32 % along with the appearance of sulfite ions over $TiO_2$ particles upon exposure of $SO_2$ (Fig.S12). Interestingly, the decrease in sulfate yield by around 30 % in the presence of $O_2^{·-}$ scavenger p-benzoquinone is almost complementary to that added with ·OH scavenger using TBA (70 %), pointing toward a minor sulfate formation pathway contributed by $O_2^{·-}$ relative to the major pathway by $CO_3^{·-}$ when carbonate ions are presented to efficiently capture ·OH ions.*

*Following Shang's work (Shang et al., 2010a), $O_2^{-}$ involved $SO_2$ oxidation can be given as Eqs. 11-13:*

*$e^{-}+O_2\rightarrow O_2^{-}$* (Eq.11)

*$SO_2+O_2^{-}\rightarrow SO_3+O^{-}$* (Eq.12)

*$SO_3+H_2O\rightarrow H_2SO_4$* (Eq.13)

*Where intermediates $SO_3$ formed via the interaction between $SO_2$ and $O_2^{-}$ subsequently couple with water molecules to produce sulfate species as a final product." (**Main Text, Page 14-, Line 337-352**)*

**Oxygen supply and consumption:**

In addition to the consideration of the role of $O_2^{-}$ in sulfate formation over $TiO_2$ particles. We supplied estimation of oxygen consumption rates and oxygen supply rates in the reaction systems considered in this study. When (bi)carbonate ions are introduced into the reaction, they serve the excellent ·OH scavenger to form $CO_3^{-}$, leaving two major active intermediates $O_2^{-}$ and $CO_3^{-}$ responsible for fast sulfate formation.

For $CO_3^{-}$ pathway:

$$h^{+}+H_2O\rightarrow\cdot OH$$
$$\cdot OH+HCO_3^{-}/CO_3^{2-}\rightarrow CO_3^{-}$$
$$CO_3^{-}+SO_3^{2}\rightarrow SO_3^{-}$$
$$SO_3^{-}+O_2\rightarrow SO_5^{-}$$
$$SO_5^{-}+SO_3^{-}\rightarrow SO_4^{-}+SO_4^{2-}$$
$$SO_4^{-}+SO_3^{2}\rightarrow SO_4^{2-}+SO_3^{-}$$

For $O_2^{-}$ pathway:

$$e^{-}+O_2\rightarrow O_2^{-}$$
$$SO_2+O_2^{-}\rightarrow SO_3+O^{-}$$
$$SO_3+H_2O\rightarrow H_2SO_4$$

As we discussed above, the relative contribution of $CO_3^{-}$ and $O_2^{-}$ to overall sulfate formation over $TiO_2$ particles could be assumed to be 0.7 and 0.3, respectively. Together with the major $SO_2$ oxidation reaction channel considered above, one may note that 1 mole of oxygen contributes to 1.7 moles of sulfate, and $H_2O$ provides an additional oxygen source compensating for the oxygen deficit. Given the measured $SO_2$ uptake coefficient in the DRIFTs chamber, the sulfate formation rates are thus determined to be 0.33 $\mu M\ s^{-1}\ m^{-2}$ for geo surface of $TiO_2$ particles, known as the upper limit of uptake capability, corresponding to the maximum oxygen consumption rate of nearly 0.19 $\mu M\ s^{-1}\ m^{-2}$. For $TiO_2+CaCO_3$ particles, the sulfate formation rate is 2.01 $\mu M\ s^{-1}\ m^{-2}$. We applied the relation (1 mole of oxygen ~ 1.7 moles of sulfate) for calculating the oxygen consumption rate in the $TiO_2+CaCO_3$ system. This operation leads to a conservative estimation of oxygen consumption rate since the relative contribution of carbonate radical ions to sulfate formation are expected to be even more predominant in the "$TiO_2+CaCO_3+Air$" system (1 mole of oxygen ~ 2 moles of sulfate) than that in "$TiO_2+CO_2+Air$" system (1 mole of oxygen ~ 1 mole of sulfate). Therefore, the oxygen consumption rate is not likely to exceed 1.18 $\mu M\ s^{-1}\ m^{-2}$

We further considered the oxygen supply capability over water layers attached to the dust particles. A steady-state of gas diffusion is described as a state where the diffusion flux density $J_s$, stays

constant and by integration from 0 to *l*. Fick's first law can be expressed in the following form (Nguyen et al., 1992):

$$J_s = -D\frac{\Delta C}{l} \tag{R5}$$

where, $\Delta C$ is the concentration difference between saturation and the system at a given time, and $D$ the mass transfer coefficient (0.021 millimeters$^2$/s), and $l$ the distance between water layers with gradient oxygen concentration. At RH = 30 %, nearly two aqueous-like water layers are believed to absorb onto the dust particle surface (Mogili et al., 2006) (Peters and Ewing, 1997), and around 0.3 nm, known as the typical thickness feature for mono water layer (Ali et al., 2015; Gao et al., 2020; Ruiz-Agudo et al., 2013), and 0.6 nm is thus adopted for $l$. For degassed single water layer devoid of $O_2$ in our system, the flux of $O_2$ supplied across the two aqueous-like water layers is at a rate of 17.08 M s$^{-1}$ m$^{-2}$, which is several orders of magnitude higher than that of oxygen consumption determined for both "TiO$_2$+CO$_2$+Air" and "TiO$_2$+CaCO$_3$" systems. Therefore, oxygen is sufficient in the reaction, allowing the considered chain reactions to continually proceed.

***Detailed correction in the manuscript:***
"*However, we noted that the insufficient $O_2$ supply in aqueous media may be an underlying constraint to the proposed $CO_3^{\cdot-}$-initiated $SO_2$ oxidation pathway. Hence, we estimated both oxygen consumption and supply rates, and oxygen supply flux can be several orders of magnitude larger than corresponding consumption flux (see detailed discussion in the supplementary text 5). Therefore, oxygen is sufficient in the reaction, allowing the considered chain reactions to continually proceed.*"
***(Main Text, Page 12, Line 300-304)***

**Q$_3$-B-1:** Secondly, according to eq 6-8 (in eq 7, SO$_4$ should be SO$_4^{2-}$?)
**Response to Q$_3$-B-1:** Thanks for your careful review and it should be SO$_4^{2-}$, which has been revised and highlighted in the current version of the manuscript.

**Q$_3$-B-2:** It seems that the oxidation of SO$_3^{2-}$ by O$_2$ could be a catalytic reaction while •SO$_3^-$ acted as catalyst. If so, the amounts of sulfate formed through photooxidation on TiO$_2$ should be the same (at least close to) in the presence of carbonates and CO$_2$ since •CO$_3^-$ only contribute to the formation of •SO$_3^-$. Is it?
**Response to Q$_3$-B-2:** Carbonate ions capture ·OH more efficiently than bicarbonate ions (more than one order of magnitude). Consequently, carbonate ions can yield more CO$_3^{\cdot-}$ than bicarbonate ions do. Carbonate salt and CO$_2$ are known to dissociate into (bi)carbonate ions in the aqueous medium, in which carbonate ions are dominant in the aqueous system containing carbonate salt while bicarbonate ions are major speices in the CO$_2$-aerated solution. Therefore, over humidified dust particels, carbonate radical ions are expected to be enriched in "TiO$_2$+carbonate salt" scenario, more abundant than in "TiO$_2$+CO$_2$" one.

   Besides, DFT calculation provides theoretical evidence that carbonate radical ions decrease the energy barrier for SO$_3^-$ formation, and its reaction with SO$_3^{2-}$ is thus faster than that with hydroxyl radical (Fig. 4 b and c). Considering this, CO$_2$ and carbonate ions severs as a precursor of CO$_3^{\cdot-}$, and thus increases sulfate yield. Therefore, sulfate yield over TiO$_2$ particles in the presence of CO$_2$ and carbonate ions is higher than that over pristine TiO$_2$.

**Q₃-C:** Thirdly, what's the pH effect on the reaction? As seen in eq 3 and 4, only the reaction of •OH with $HCO_3^-/CO_3^{2-}$ was considered. What's about the reactions between $H^+$ and $HCO_3^-/CO_3^{2-}$? As the oxidation of $SO_2$ or sulfite increased, the pH should decrease and then affect $HCO_3^-/CO_3^{2-}$.

**Response to Q₃-C:** pH ($H^+$ concentration) is an important factor within the aqueous chemical reaction process. Yet so far adjusting the pH ($H^+$ concentration) of particle surfaces is quite tough, and exploring the role of dust surface pH ($H^+$ concentration) in the reactivity of $CO_3^{.-}$ is not easily achieved. Notwithstanding, as we discussed in the section "Q1-2. Observation of carbonate radical formed in carbonate-containing $TiO_2$ suspension", the increase of pH in $TiO_2$ suspension was observed to promote the production of $CO_3^{.-}$, further strengthening the oxidation capability of dust particles. In contrast, decreasing pH is expected to reduce the yield of $CO_3^{.-}$ since the reaction rate of $CO_3^{2-}$ with $\cdot OH$ is lower than that with $HCO_3^-$. On this basis, we then examined whether the pH of the mineral dust surface can be sustained to maintain fast $SO_2$ oxidation triggered by sufficient $CO_3^{.-}$ in the typical lifespan of mineral dust after accumulation of sulfate production. Considering this, we thus plotted the heterogeneous sulfate production over $TiO_2$ and $TiO_2+CaCO_3$ particles versus equivalent exposure time (Fig. R6).

The heterogeneous reaction of $SO_2$ on $TiO_2$ in the presence of $CO_2$ as well as on $TiO_2+CaCO_3$ mixtures was investigated by *in situ* DRIFTS technique. Similar to the $TiO_2+Air+CO_2$ system, we also plotted the heterogeneous sulfate production over $TiO_2$ and $TiO_2+CaCO_3$ particles versus equivalent exposure time assuming that the atmospherically relevant concentration of $SO_2$ is 20 ppb (Fig. R6). The equivalent exposure time determined for these two sets of experiments is across one day to nearly two weeks. Clearly, the sulfate yield builds up steadily during the two-week equivalent exposure time, suggesting that the regime of $CO_3^{.-}$ initiated $SO_2$ oxidation over $TiO_2$ and $TiO_2$-$CaCO_3$ particles is slightly affected by the decrease of surface pH due to the accumulation of sulfate production over the course of equivalent exposure time. In the atmosphere, the lifetime of typical mineral dust particles ranges from several days to weeks (Bauer and Koch, 2005), and the equivalent exposure time considered in this study (nearly 2 weeks) falls right within the characteristic lifespan range of mineral dust particles. This leads us to deduce that pH variation due to persistent growth of sulfate during the reaction shows a negligible effect on $CO_3^{.-}$ initiated $SO_2$ oxidation channel proposed in this work.

[Figure]

**Fig. R6.** *In situ* DRIFTS of S(IV) and S(VI) species on $TiO_2$ and $TiO_2+CaCO_3$ mixtures (wt./wt. = 50/50) upon irradiation as function of equivalent exposure time. Reaction conditions: RH = 30 %, Light intensity (I) = 30 mW cm$^{-2}$, Total flow rate = 52.5 mL min$^{-1}$, and $CO_2$ = 400 ppm.

***Detailed correction in the manuscript:***

*"pH is an important factor within aqueous chemical reaction processes and is likely to alter the reaction scheme. Yet so far adjusting the pH of particle surfaces is quite tough, and exploring the role of dust surface pH in the reactivity of $CO_3^-$ is not easily achieved. Notwithstanding, the increase of pH in $TiO_2$ suspension was observed to promote the production of $CO_3^-$, further strengthening the oxidation capability of dust particles. In contrast, decreasing pH is expected to reduce the yield of $CO_3^-$ since the reaction rate of $CO_3^{2-}$ with ·OH is nearly two orders of magnitude higher than that with $HCO_3^-$. On this basis, we then examined whether the surface pH of mineral dust can be sustained to maintain fast $SO_2$ oxidation triggered by $CO_3^-$ in the typical lifespan of mineral dust. Considering this, we thus plotted the heterogeneous sulfate production over $TiO_2$ and $TiO_2+CaCO_3$ particles versus equivalent exposure time (Fig. S15). Clearly, the sulfate yield builds up steadily during the two-week equivalent exposure time (see more detailed discussion on determining equivalent exposure time in supplementary text 6), suggesting that the regime of $CO_3^-$ initiated $SO_2$ oxidation over $TiO_2$ and $TiO_2+CaCO_3$ particles are slightly affected by the possible decrease of surface pH due to accumulation of sulfate production over entire reaction course. In the atmosphere, the lifetime of mineral dust particles ranges from several days to weeks (Bauer and Koch, 2005), and the equivalent exposure time considered in this study (nearly 2 weeks) falls right within the characteristic lifespan range of mineral dust particles. This leads us to deduce that persistent growth of sulfate shows a negligible effect on $CO_3^-$ initiated $SO_2$ oxidation scheme proposed in this work."* ***(Main Text, Page 14, Line 352-367)***

Other Concerns:

1.  The concentrations of $SO_2$ used are much higher than the ambient atmospheric concentration.

**Response to Q₄:** Thanks for your question. We also note the large gap between the $SO_2$ concentration applied in our lab study and the $SO_2$ concentration measured in field observations. Therefore, we considered the reaction order of the heterogeneous reaction of $SO_2$ on the dust particles $TiO_2$. It follows the pseudo-first-order kinetics in the range of 400 ppb-20000 ppb, covering all $SO_2$ concentrations applied in this study. While we note nearly one order of magnitude of $SO_2$ concentration gap lies between experimental studies and field observations, we assume this gap may slightly affect the findings in this study, and the proposed mechanism remains valid.

2.  Line 228: it is difficult to understand this sentence "ESR data (Fig. 3d) further confirms the increase of $SO_3^{\cdot-}$ after 2 min UV irradiation in the presence of carbonate ion" since the change is not very obvious.

**Response to Q₅:** We have repeated our ESR measurements to solidify the point that $SO_3^{\cdot-}$ is increased in the presence of carbonate radical ions. For visual clarity, we also provided the integrated areas for two ESR spectra. In Fig. R7, the presence of carbonate ions in the $TiO_2+S(IV)$ system evidently promotes the generation of $SO_3^{\cdot-}$, which verifies our proposed mechanism.

[Figure]

**Fig. R7.** ESR spectrometry of [DMPO–$SO_3\cdot$] intermediate formed in a solution of d $TiO_2$ (3 mg ~ 4 mL) + 0.1 M $Na_2SO_3$ and $TiO_2$ (3 mg ~ 4 mL) + 0.5 M $Na_2CO_3$ + 0.1 M $Na_2SO_3$. For visual clarity, the integrated areas of ESR profiles were also presented for direct comparison. Exp. and Sti. stand for experimental results and corresponding fitting results using software Isotropic Radicals.

3.  Sampling in field observation. The samples collected in daytime and nighttime did not mean they are always in dark and illuminated conditions. The samples collected in nighttime may also have undergone multiple daytime photochemical processes.

**Response to Q₆:** Thank you for your thoughtful question. Generally, the aerosol lifetime is on the order of less than an hour to days (Koelemeijer et al., 2006), highly depending on particle size. For example, the lifetime of $PM_{10}$ ranges from minutes to hours, and its travel distance, in general, is

less than 10 km (Agustine et al., 2018). When PM downsizes to 2.5 μm, $PM_{2.5}$ has a lifetime prolonged to nearly one day or longer (Liu et al., 2020). Therefore, in our sampling, $PM_{3.3}$-$PM_{9.0}$ are expected to have a relatively long lifetime, on the order of several hours or more, which enables the heterogeneous reaction process to become a more important contributor to overall sulfate ions measured in $PM_{3.3}$-$PM_{9.0}$ than that in $PM_{\geq 9.0}$. This is supported by our observations where during the daytime hours the correlation coefficients for $PM_{3.3}$-$PM_{9.0}$, i.e. 0.56 (sulfate vs carbonate) and 0.61 (sulfate vs bicarbonate), are higher than that of $PM_{\geq 9.0}$, i.e. 0.489 (sulfate vs carbonate) and 0.36 (sulfate vs bicarbonate), respectively.

We also note that their correlations are not high in part because the concentration of water-soluble ions determined in these samples may not come from the net contribution of processes in day-time and night-time periods. As you mentioned, some undesired processes that take place during the day(nigh)-night(day) shifts may also contribute to the concentration of sulfate ions in separate sampling hours. Nevertheless, the negative correlations between the mass concentrations of sulfate ions and (bi)carbonate ions are observed in the nighttime hours, consistent with the suppression of sulfate formation by $CO_2$ in the dark experiments. Instead, positive correlations are seen for those ions within PM sampled during the daytime hours. This matches with the scenarios in which sulfate production upon irradiation in the presence of (bi) carbonate ions is increased over both model and authentic dust particles. Taken the lifetime of $PM_{\geq 3.3}$ as well as the distinct trends observed during day-time and night-time periods, it is plausible that in this study the ambient PM collected separately in the daytime and nighttime hours, to some extent, are likely to reflect aerosol particles that mainly go through heterogeneous reaction under dark and irradiation, analogous to the scenario where we considered in lab simulations.

***Detailed correction in the manuscript:***

"*However, under the low wind speed (0.76 ± 0.73), correlation coefficients $R^2$ obtained for relationship between bi(carbonate) and sulfate ions are not promising, 0.56 (sulfate vs carbonate) and 0.61 (sulfate vs bicarbonate) for $PM_{3.3}$-$PM_{9.0}$ during daytime hours. A plausible explanation is that although less significant, local primary emission source also brings certain bias and uncertainty to the correlation analysis. Shanghai is a coastal city, and sulfate species such as $K_2SO_4$ and $Na_2SO_4$ from the sea salt contribute to the local sulfate emission as well (Long et al., 2014). On the other hand, this novel $SO_2$ oxidation channel is in the infant stage, and only active mineral dust components have been considered in this work whereas other components found in the coarse mode of PM such as organic matter, elemental carbon as well as sea salt (Cheung et al., 2011) are likely to involve this mechanism and alter the response of sulfate yield to $SO_2$ heterogeneous uptake. In addition, the concentration of water-soluble ions determined in these samples (relatively small size) may not come from the net contribution of heterogenous reaction processes in absolute day-time and night-time periods. Some of the undesired processes that take place during day(nigh)-night(day) shifts may also contribute to the concentration of sulfate ions in separate sampling hours.*"
***(Main Text, Page 16, Line 413-424)***

4. In addition, as proposed by Sullivan et al. (Atmos. Chem. Phys., 7, 1213–1236, 2007), oxidation of S(IV) to S(VI) by iron in the aluminosilicate dust is a possible explanation for the enrichment of sulphate in Asian mineral dust. So, how to exclude the effect of Fe in this study?

[Figure]

**Fig. R8.** (a) Determination of sulfite and sulfate concentration after exposure to air flow under irradiation in $Fe_2O_3$-$CaCO_3$ particles for 20 min. (b) Band positions of typical active mineral dust components (at pH = 7 in aqueous media), with highlights on the oxidation capability and generation of reactive oxygen species. Reaction conditions: RH = 30 %, Light intensity (I) = 30 mW cm$^{-2}$, Total flow rate = 52.5 mL min$^{-1}$.

**Response to Q$_7$:** Thanks for your suggestion. We thus considered the role of iron promoters in our proposed mechanism by using the alpha-$Fe_2O_3$+$CaCO_3$ mixture. Similar to experiments using $TiO_2$+$CaCO_3$ mixture, alpha-$Fe_2O_3$+$CaCO_3$ are prepared by grinding alpha-$Fe_2O_3$ and $CaCO_3$. The ratio of iron oxide and calcium components is fixed at 3:4 according to EDS mapping analysis results. In panel a, our results show that alpha-$Fe_2O_3$ can not trigger fast $SO_2$ oxidation in the presence of carbonate ions upon irradiation, which is distinguished from results we derived from the $TiO_2$+$CaCO_3$ mixture. This can be explained by the fact that $Fe_2O_3$ shows a lower redox activity compared to $TiO_2$ regardless of the anatase phase and rutile phase (panel b). We collected the redox potential vs NHE (V) of $O_2/O_2^{.-}$, $CO_3^{.-}/H^+/HCO_3^{.-}$, and $CO_3^{.-}/CO_3^{2-}$ in the previous literature, with both valence band and conduction band information of each dust particle, i.e. semiconductors, shown in panel a. Owing to the strong redox capability of $TiO_2$ particles, where the photo-induced electrons and holes are able to form $O_2^{.-}$ and ·OH radical ions, respectively. In stark contrast, mineral dust component $Fe_2O_3$ has a rather narrow band gap, with its valence band and conduct band lying at -0.18 and at 1.68 V vs. NHE at pH = 7, lower than the redox potential required for generating $CO_3^{.-}/H^+/HCO_3^{.-}$, and $CO_3^{.-}/CO_3^{2-}$, which thus can not produce $CO_3^{.-}$.

Nevertheless, we are aware of the inconsistency between our lab results and the reported results in the literature (Li et al., 2019; Toledano and Henrich, 2001). Toledano et al. observed a UV-induced increase in adsorption of $SO_2$ over alpha-$Fe_2O_3$ (0001) using the XPS technique. The difference is likely to correlate to the different light sources and dust sources. 30 mW cm$^{-2}$ of photon flux was applied using a solar simulator in our lab study, corresponding to 0.3 times of AM 1.5 G solar irradiance while they employed a focused 200 W Hg(Xe) lamp, which provides a strong light source, with $hv > E_{gap}$ (2.2 eV) roughly 70 times of the solar flux in that wavelength range. On the other hand, we adopted commercially available alpha-$Fe_2O_3$ nanoparticles and the chemical

properties of which are believed to be different from that of single-crystal $Fe_2O_3$ with a pure (0001) surface. It is believed that crystal plane, morphology, and size can modulate the inherent band gap (the position of conduction band relative to the position of valence band) of semiconductors (Alivisatos, 1996; Xu et al., 2013; Xu et al., 2015). In Li's work, they synthesized four types of $Fe_2O_3$ nanomaterials with different morphologies, which have a various abundance of each crystal facet. This leads to distinct photochemical properties compared to pristine $Fe_2O_3$ nanoparticles.

In addition, we observed a slight decrease of sulfite and sulfate yield upon irradiation compared to dark experiments, consistent with Du's work (Du et al., 2019), in which a more evidently decrease is found in the initial reaction stage. A plausible explanation for this observation is that while we applied xenon lamp as a light source for experiments, elevated temperature in the chamber is likely to decrease the $SO_2$ uptake over dust particles, more evidently for those particles with dark colors. Our earlier study shows that $SO_2$ uptake over $Fe_2O_3$ particles are sensitive to temperature (Wang et al., 2018), and persistent increase in temperature hinders $SO_2$ adsorption, and consequently reduce sulfate yield. Overall, we show that ferric oxide can not initiate fast $SO_2$ oxidation by generating $CO_3^-$ ions where we considered in this study due to its poor photo activity although ferric chemistry is important in secondary sulfate formation in the atmosphere.

***Detailed correction in the manuscript:***

*"$Fe_2O_3$ is also one of the crucial components found in the authentic mineral dust (El Zein et al., 2013), and it has been reported to produce ROS under solar irradiation (Li et al., 2019), thus likely involving the reaction mechanism proposed in this work. Similar to experiments using $TiO_2+CaCO_3$ mixture, alpha-$Fe_2O_3+CaCO_3$ are prepared by grinding alpha-$Fe_2O_3$ and $CaCO_3$. In Fig. S9a, our results show that alpha-$Fe_2O_3$ can not trigger fast $SO_2$ oxidation in the presence of carbonate ions upon irradiation, which is distinguished from results we derived from the $TiO_2+CaCO_3$ mixture. This can be explained by the fact that $Fe_2O_3$ shows a lower redox activity relative to $TiO_2$ (Fig. S9b), where its strong redox capability essentially enables photo-induced electrons and holes to produce $O_2^-$ and ·OH radical ions. In stark contrast, the valence band and conduct band of $Fe_2O_3$ lie at -0.18 and at 1.68 V vs. NHE (pH = 7), lower than the redox potential required for generating $O_2^-$, ·OH as well as $CO_3^-$ (Li et al., 2016). Hence, no promoted sulfate production is seen for $Fe_2O_3+CaCO_3$ particles under irradiation. More discussion on the inconsistency between our study and the previous results regarding the response of $SO_2$ oxidation to solar irradiation can be found in supplementary text 3.*

*Overall, we show that upon irradiation atmospherically relevant content of $TiO_2$ (nearly 1 %) found in authentic dust simulants is able to interact with carbonate ions to launch a fast $SO_2$ oxidation channel, which is beyond the conventional regime of alkaline neutralization of $H_2SO_4$. Unlike $TiO_2$, alpha- $Fe_2O_3$ can not initiate fast $SO_2$ oxidation by generating $CO_3^-$ due to its limited photo activity although ferric chemistry is important in secondary sulfate formation in the atmosphere (Sullivan et al., 2007; Yermakov and Purmal, 2003)."*
***(Main Text, Page 7, Line 173-189)***

5.  Conclusion and atmospheric implications: this study only found the enhanced sulfate formation in mixed $TiO_2$ and $CaCO_3$ particles compared to individual $TiO_2$ or $CaCO_3$ particles. However, the hypotheses of $CO_2^-$ derived carbonate species and carbonate salt works as the precursor of $\cdot CO_3^-$ is exaggerated. As seen in this study, $TiO_2$ is necessary but its content in atmospheric

particulate matter is very low. Considering the unclear role of •CO$_3^-$, as well as the high concentration of SO$_2$ used, its implications even on sulfate formation is limited. Consequently, the extension of its atmospheric implications to fine PM concentration, human health, and climate is not meaningful.

Response to Minor Concern 5: Thanks for your comments. We have performed several experiments to prove the significance of the novel SO$_2$ oxidation channel proposed in this work. The detailed discussion is shown as follows:

**1. CO$_3^-$ production from the atmospherically-relevant concentration of CO$_2$**

In this work, we investigated not only the synergistic effect between CaCO$_3$ and TiO$_2$ but the role of atmospheric atmospherically-relevant concentration of CO$_2$ in promoting SO$_2$ oxidation. By using a series of authentic dust particles such as Arizona test dust (ATD), clays IMt-2 (Illite, Mont., USA), and K-Ga-2 (Kaolin, Georgia, USA), we prove that CO$_2$-derived carbonate radical ions can increase sulfate production, especially for K-Ga-2 due to its enriched TiO$_2$ content ($\approx$ 3 %). It almost doubles the sulfate yield in the presence of CO$_2$+air upon irradiation compared to that in the air flow upon irradiation. This result indicates that our proposed scheme does exist and plays role in sulfate formation in the atmosphere.

***Detailed correction in the manuscript:***

*"As another step toward a real scenario in the atmosphere, experimental trials employing authentic dust particles, i.e. Arizona test dust (ATD), clays IMt-2 (Illite, Mont., USA) and K-Ga-2 (Kaolin, Georgia, USA), were implemented (Table S2). In Fig. 4, K-Ga-2 clay exhibits the most marked promotional effect on sulfate yield (by nearly 100 % increased sulfate production in the CO$_2$-involved case under irradiation). This correlates with its considerable TiO$_2$ contents (3.43 %) in the K-Ga-2 clay, in which active intermediates are readily evolved from TiO$_2$ and (bi)carbonate species upon irradiation. However, the promotional effect of CO$_2$ on sulfate production under irradiation is weak for IMt-2 (the content of TiO$_2$ $\approx$ 0.99 %) and ATD (the content of TiO$_2$ $\approx$ 0.46 %) as compared to K-Ga-2 particles. This may correlate to their higher mass fraction of alkaline earth metal oxide (denoted as A.E.), which enables dust particles to possess a large number of (bi)carbonate species in the natural environment where they have experienced long-term exposure to atmospheric CO$_2$ during the regional transport. Therefore, the aforementioned synergetic effect takes effect over IMt-2 and ATD particles even without exposure to CO$_2$ due to the presence of abundant carbonate formed, and a less evident increase of sulfate yield is observed."*

*__(Main Text, Page 8, Line 200-210)__*

**2. Observation of enhanced sulfate formation over synthetic mineral dust proxy (SiO$_2$: Al$_2$O$_3$: CaCO$_3$: TiO$_2$ = 81:9.6:7.7:1.0)**

*"To generalize our finding to a more real condition, the rapid SO$_2$ oxidation pathway was further probed by employing mineral dust simulants where two dominant crust constituents SiO$_2$ and Al$_2$O$_3$ were introduced into TiO$_2$-CaCO$_3$ particles to mimic the authentic mineral dust particles in the atmosphere, with specific component and corresponding ratio information shown in Table S1. It is worth mentioning that the determination of the ratio of each component in the simulants relies on the EDS mapping results of ATD particles. In Fig. 2, the introduction of TiO$_2$ components ($\approx$ 1 % wt.) into SiO$_2$-Al$_2$O$_3$ leads to 81.6 % enhancement of sulfate production while merely 24.8 % wt. increase of sulfate yield was observed once $\approx$ 8 % wt. of CaCO$_3$ was incorporated into SiO$_2$-Al$_2$O$_3$*

*dust particles. Surprisingly, mixing of ≈ 1 % mass fraction of $TiO_2$ and ≈ 8 % wt. of $CaCO_3$ into $SiO_2$-$Al_2O_3$ gives rise to a 235 % increase of sulfate formation relative to that of $SiO_2$-$Al_2O_3$. Hence, the synergistic effect on heterogeneous oxidation of $SO_2$ is likely to take effect in the atmosphere. Overall, we show that even using the mass ratio of $SiO_2$, $Al_2O_3$, $CaCO_3$, and $TiO_2$ detected in authentic particles, evident acceleration of sulfate production is observed. Hence, the proposed mechanism in this study is prone to play role in the atmosphere." (Main Text, Page 8, Line 164-173)*

**3. Investigation of the reaction order of $SO_2$ uptake over dust particles ($SO_2$ concentration ranges from 400 ppb to 20000 ppb)**

We note that a relatively high concentration of $SO_2$ is applied for experiments in this work, and this concentrate gap may bring uncertainty to the reaction kinetics and applicability of the proposed reaction scheme in the atmosphere. To properly describe the reaction efficiency of gas-surface interactions, sulfate formation rates as a function of $SO_2$ concentration were initially determined to verify its reaction order in the selected concentration range (Fig. S10 A-E), which is a crucial step to give a credible estimation of $SO_2$ uptake coefficient. Based on a prior study (Shang et al., 2010b), $SO_2$ uptake on particle surfaces depends on the $SO_2$ concentrations and active sites, which thus could be described by the following equation (Eq. **R6**):

$$\frac{d[SO_4^{2-}]}{dt} = k[SO_2]^m[CO_2]^l[TiO_2]^n[H_2O]^p \qquad \textbf{[R6]}$$

where $[SO_4^{2-}]$ refers to the sulfate concentration on $TiO_2$ surfaces, $[SO_2]$( $[CO_2]$) to $SO_2$ ($CO_2$) gas concentration employed in the system, $[TiO_2]$ to the concentration of active sites on the $TiO_2$ particle surfaces, and $[H_2O]$ represents for surface water concentration, and m, l, n, and p are the reaction orders of corresponding species. Clearly, the steady growth of sulfate on $TiO_2$ particles in the presence of $9.83 \times 10^{15}$ molecules cm$^{-3}$ $CO_2$ made a clear indication (panel A-E) that the decrease in surface active sites is negligible. Meanwhile, mass flow controllers provide stable gas flow and maintain the constant concentrations of humidified air and $CO_2$, which allows us to simplify the Eq. **R6** to Eq. **R7** through a logarithm function.

$$\lg \frac{d[SO_4^{2-}]}{dt} = \lg k + m \lg [SO_2] + C \qquad \textbf{[R7]}$$

where C stands for $l\lg[TiO_2] + n\lg[TiO_2] + p\lg[H_2O]$, and $[SO_2]$ for the concentration of $SO_2$ where particles are exposed. We then plotted the sulfate formation rate against exposed $SO_2$ concentration. Linear fitting analysis for those points resulted in 1.13 order for the reaction, with $R^2 ≈ 0.99$ (Fig. S10 F). So far, 400 ppb is the lowest concentration that we are able to apply for the uptake measurements due to the limitation of the current experimental setup. The prior work has demonstrated that atmospheric $SO_2$ concentration reaches up to 40 ppb (Franchin et al., 2015). We note that a difference of nearly a factor of 10 remains for the $SO_2$ concentration employed in laboratory studies and that measured in field observations. Nevertheless, we have already verified its pseudo-first-order kinetic in the wide range of 400-20000 ppb. This leads us to assume that uptake coefficients estimated under ppm level remain valid, and those datasets derived from laboratory chambers are able to be generalized to the atmosphere condition. *(Supporting Information, Page S5)*

**4. Observation of the ejection of gas-phase $CO_3^-$ on the dust particle surface upon irradiation.**

Through probe molecular aniline, we determined the steady concentration of $[\cdot OH]_{ss}$ and $[CO_3^{-}]_{ss}$ released from $TiO_2$ particles in the absence and presence of $CO_2$, respectively. Specifically, the steady-state concentration of carbonate radicals was determined to be $1.39 \times 10^{-13}$ M for the $TiO_2+Air+CO_2$ system, which is much higher than that of hydroxyl radicals measured in the $TiO_2+Air$ system ($2.15 \times 10^{-15}$ M). Previously, it has been demonstrated that the gas-phase hydroxyl radical produced from $TiO_2$ has a great impact on sulfate formation. Our study unfolds that the production of gas-phase carbonate radical ions in the presence of $CO_2$ over mineral dust upon irradiation. To be important, since $CO_3^{-}$ enter into the gas phase, they will promote the oxidation of $SO_2$ in the gas phase to form external sulfate aerosol, which is known to serve as cloud condensation nuclei and play a role in the global climate by scattering solar radiation. Hence, the rapid $SO_2$ oxidation pathway proposed in this work shows its non-negligible atmospheric implications to fine PM concentration, human health, and climate.

***Detailed correction in the manuscript:***

*"When $CO_2$ (atmospheric relevant concentration) is introduced into the home-made flow-cell chamber, with an intervening gap between $TiO_2$-coated film and probe molecule solution fixing at nearly 2 mm, and a short distance of which allows possible gaseous ROS to diffuse and react with aniline molecular (None, 2013). An increased degradation rate of aniline was seen, which can be attributed to the generation of active carbonate radical ions (Fig. S17). The maximum concentration of steady-state $CO_3^{-}$ radical ions supplied by partition processes between gas phase and solid-liquid phases (humified dust particles) was determined to be $1.39 \times 10^{-13}$ M for the $TiO_2+Air+CO_2$ system, which is over one order of magnitudes higher than that of $\cdot OH$ for $TiO_2+Air+$ system ($2.15 \times 10^{-15}$ M). This observation matches with the earlier study where the concentration of carbonate radical can be two orders of magnitudes than $\cdot OH$ over the water surface (Sulzberger et al., 1997a)."* ***(Supporting Information, Page 15, Line 373-380).***

---

## Author Comment (AC2)

**Response Letter (For Referee 2)**

**Comment from Referee 2:**

"The authors provide a strong case that carbonate radical reactions could contribute substantially to the atmospheric production of sulfate from $SO_2$. I have no argument with all of the very extensive laboratory and theoretical studies. It makes good sense. However, when we get to the field study data, the hypothesis does not seem to hold up. In Figure 5 we are presented with the experimental "correlation" of sulfate and carbonate ions in daylight and at night. To call the correlation "weak" is being very generous. In fact, I would argue there is no demonstrated correlation at all. I think the authors ought to re-write their paper conclusion to highlight the lack of any observed correlation, even though there ought to be one based on all of the laboratory work. They could speculate on why the field study failed to find the expected relationship, and suggest more field studies to resolve the issue."

**Author general reply:**

Thanks for your valuable suggestions, which greatly helped us to improve the manuscript. According to your comments, we have revised the discussion on field measurement and rewritten the conclusion section to highlight the significance of the findings in this work. Besides, we would like to give a special thank you for your careful check on our manuscript, pointing out the grammar issues listed as follows. We have revised all of the problems that you've pointed out, and they were all fully revised and highlighted in blue in the current version of the manuscript.

**Response to the weak correlation between sulfate and carbonate ions in the field observation:**

We note that weak correlation coefficients were obtained from current field observations with regard to the relationship between sulfate and bi(carbonate) ions in PM. This is because sulfate ions found in PM come from both primary sources and secondary sources, locally or non-locally. To confirm which source is more dominant in our PM sample, we collected and analyzed meteorological information from the open-access database (https://www.aqistudy.cn/). During the sampling period, the wind scale mainly varies from 0 to 1, corresponding to the wind speed ranging from 0 to 1.5 m s$^{-1}$ (Fig. R8). All plots shown in Fig. R1 give rise to a statistical wind speed of 0.76 ± 0.73, which represents the weak dispersion of pollutants at low wind speed (not exceeding 2.5 m s$^{-1}$)(Witkowska et al., 2016; Wu et al., 2020). On this basis, the local source is expected to be the dominant contributor to local air pollution and the mass growth of collected PM.

It is generally accepted that under stagnant meteorological conditions (wind speed < 1.5 m s$^{-1}$), for the coarse-mode (2.5 μm ~ 10 μm) of sulfate, the heterogeneous reaction of $SO_2$ on the dust surfaces is believed to be a major contributor (Liu et al., 2017). In this case, for ambient PM considered in this study (3.3 μm ~ 9.0 μm) under the low wind speed (0.76 ± 0.73), local heterogeneous chemical pathway mainly contributes to sulfate ions, while long-term transport is less important. Nevertheless, in our field observations, correlation coefficients $R^2$ obtained for the relationship between bi(carbonate) and sulfate ions are not promising, 0.56 (sulfate vs carbonate) and 0.61 (sulfate vs bicarbonate) for $PM_{3.3}$-$PM_{9.0}$, respectively, during daytime hours. A plausible explanation is that although less significant, local primary emission source also brings bias and uncertainty to the correlation analysis. Shanghai is a coastal city, and sulfate species such as $K_2SO_4$

and $Na_2SO_4$ from the sea salt contribute to the local PM as well (Long et al., 2014). On the other hand, this novel $SO_2$ oxidation channel is in the infant stage, and only active mineral dust components have been considered in this work whereas other components found in the coarse mode of PM such as organic matter, elemental carbon as well as sea salt (Cheung et al., 2011) are likely to involve this mechanism and alter the response of sulfate yield to $SO_2$ heterogeneous uptake.

For those large particles (LP), that refer to the particles with a diameter large than 9 μm in this work, sulfate ions show a rather weak or even no correlation to (bi)carbonate ions during the night-time and day-time hours (Fig 7a and 7c). This is likely correlated to the short lifetime of LP. Generally, the aerosol lifetime is on the order of less than an hour to several days (Koelemeijer et al., 2006), highly depending on particle size. For example, the lifetime of $PM_{10}$ ranges from minutes to hours, and its travel distance, in general, is less than 10 km (Agustine et al., 2018). As a consequence, secondary sulfate formation through chemical reaction over LP is not significant with respect to in situ emissions. When PM downsizes to 2.5 μm, $PM_{2.5}$ has a lifetime prolonged to nearly one day or longer (Liu et al., 2020). Therefore, $PM_{3.3}$-$PM_{9.0}$ are expected to have a relatively long lifetime, on the order of several hours or longer, which enables the heterogeneous reaction process to become a more important contributor to overall sulfate ions measured in $PM_{3.3}$-$PM_{9.0}$ than that in $PM_{\geq 9.0}$. This is supported by our observations where during the daytime hours the correlation coefficients for $PM_{3.3}$-$PM_{9.0}$, i.e. 0.56 (sulfate vs carbonate) and 0.61 (sulfate vs bicarbonate), are higher than that of $PM_{\geq 9.0}$, i.e. 0.489 (sulfate vs carbonate) and 0.36 (sulfate vs bicarbonate), respectively. Similarly, higher correlation coefficients are also observed for $PM_{3.3}$-$PM_{9.0}$ than $PM_{\geq 9.0}$ in the sample collected during the nighttime period.

Taken all, a weak correlation between sulfate ions and (bi)carbonate ions observed for $PM_{3.3}$-$PM_{9.0}$ in this work correlates to non-chemical primary emission and complicated $CO_3^-$ regime of heterogenous sulfate production in the atmosphere. For LP, they have a relatively short lifetime, and thus further weaken the $CO_3^-$ initiated $SO_2$ oxidation channel to occur, leaving the correlation coefficient much low. While we note that the correlation coefficients between sulfate and (bi)carbonate are not promising in this work, field measurements of sulfate and (bi)carbonate ions shed light on their distinct correlations during the daytime and nighttime hours. This is the first time that relationships between those ions are explored separately in these two periods. Overall, carbonate radical is likely to impact the sulfate formation in the atmosphere during daytime hours. Detailed and systematic $SO_2$ oxidation triggered by $CO_3^-$ needs further investigations to enable a better interpretation of correlations between these inorganic ions at the given meteorological conditions and physico-chemical properties.

[Figure]

**Fig. R1.** Daily variations of wind scales from 20 September to 28 September in 2018, Yangpu Sipiao Station, Shanghai.

***Detailed correction in the manuscript:***

[revised manuscript text omitted]

**Response to the request for rewiring the conclusion section:**

We have fully revised the conclusion section, with detailed modifications shown as follows:

***Detailed correction in the manuscript:***

[revised manuscript text omitted]

The paper is well written, but I found some English grammar issues:

**Response to grammar issue:** We have revised the all mentioned grammar issues ($G_1$-$G_7$) listed below, and the revised part has been highlighted in the current version of manuscript.

$G_1$: Line 35 "such as" amines. **Response:** Revised

$G_2$: Line 92 and elsewhere: "phy-chemical" ought to be physico-chemical. **Response:** Revised

$G_3$: Line 95 change "better applicability in" to "simulate" **Response:** Revised

$G_4$: Line 97 "This result allows us to consider that in addition to alkaline environment alternative important force resulting in the remarkable increase of sulfate yield upon irradiation is expected within the carbonate-containing system." Change to "This result allows us to assert that the carbonate-containing system contains another important mechanism for sulfate generation beyond the production of an alkaline environment."

$G_5$: Line 223 "stemmed' should be "produced" **Response:** Revised

$G_6$: Line 238 "none" should be "no"Line 270 "complement" should be "complementary" **Response:** Revised

$G_7$: Line 292 "photo-response" should be "photo-generated. **Response:** Revised

Reference:

[revised manuscript text omitted]

---

## Author Response (AR2)

**Response Letter**

**Review 3:**

In this manuscript, Liu and co-authors examine $SO_2$ uptake by carbonate-containing mineral dust, and explore the potential role of carbonate radical-mediated processes in enhancing sulfate formation. The premise of the article is interesting and worthwhile; at the same time, the manuscript in its current form is very difficult to follow and the experimental conditions are sufficiently different from those in the ambient atmosphere that the reported conclusions require much more in the way of discussion and qualification of limitations than is currently provided. Given these concerns, I think that the manuscript requires significant (major) revisions prior to publication. I hope that the authors find my comments useful; I have aimed to be constructive throughout.

Author general reply:

We are very appreciative of your valuable suggestions, which enable us to improve the manuscript. According to your comments, we have noticed shortcomings mentioned in the comment list, especially for the aspect of atmospheric chemical relevance and the way of discussion. We improved the structure of the manuscript, and revised the original manuscript to clarify more convincing explanations to the readers, allowing them to follow the string of reasoning. We carefully consider all your comments posted to the previous version of the manuscript, and the detailed point-by-point revisions are presented as follows.

d1—Writing style

The manuscript is extremely difficult to follow, primarily because data and results text are largely in the supplemental information. As written, the supplemental reads like a point-by-point response to reviewers rather than a clear and concise summary of supporting information. As a result of this separation of information, important points in the text are quickly glossed over. For example, to me, Figure S3 seems to show that there is substantial production of sulfate at the surface of $TiO_2$–CaO upon illumination. Isn't it possible that the difference between $CaCO_3$/CaO is related to other factors? Specific surface area differences? Hygroscopicity? Solubility? In my opinion, the manuscript requires significant reorganizational effort to address this issue, since I (as a relative expert in the field) am finding it difficult to be sure that I am convinced by the data and the results as presented.

Response to Q: We carefully consider your suggestion regarding writing style and have placed our great efforts to change the layout of supporting information. Some supporting information of significance to support our statement has been moved to the main text. In addition, we have reorganized the data demonstration to convince readers of our results.

The CaO particles applied in our experiment are commercially available from the SINOPHARM company, and their BET surface area is determined to be 2.54 $m^2$ $g^{-1}$ in the literature (Tang et al., 2016b). On the other hand, the BET surface area for $CaCO_3$ particles in this study was determined to be 1.35 $m^2$ $g^{-1}$ (Quantachrome). Hence, $CaCO_3$ particles disadvantage in BET surface area compared to CaO, and it would not be the factor that promotes sulfate yield.

We compared the hygroscopicity of $CaCO_3$ and CaO by specifying RHs that allow them to form a monolayer of water. A prior study shows that a monolayer of water formed on the surface of $CaCO_3$ particles when RH is over 52 % (Li et al., 2010). On the other hand, RH of 27 % enables CaO to

form a monolayer of water (Goodman et al., 2001). While those results are collected from different literature and a bias may come from different measurement systems. However, this evident gap leads us to conclude that CaO exhibits a stronger hygroscopicity than $CaCO_3$ does. Following this, we believe at least the hygroscopicity is not the force to produce more sulfate over $TiO_2+CaCO_3$ relative to $TiO_2+CaO$ upon irradiation. Additionally, this comparison also gives an explanation for substantial sulfate production at the surface of $TiO_2+CaO$ under illumination due to its strong hygroscopicity.

We obtained solubility information from "CRC CHEMISTRY and PHYSICS HANDBOOK (97th Edition, 2016)", where calcium carbonate (calcite) is 6.6 mg/100 g $H_2O$ (Section "Physical Constants of Inorganic Compounds", page 4-53). However, the specific solubility of CaO is not available in this handbook or literature we have been searching for so far. Nevertheless, calcium oxide (CaO) is likely to have a higher solubility than calcium carbonate since it can react with $H_2O$ to form $Ca(OH)_2$ and form a saturated solution (160 mg/100 g $H_2O$) (Chrzan, 1987).

Collectively, we show that $CaCO_3$ shows humble physical properties including BET surface area, hygroscopicity as well as solubility relative to CaO while $TiO_2+CaCO_3$ particles have higher sulfate production than $TiO_2+CaO$ particles do under irradiation. Together with the observation of sulfate production over $TiO_2/CaCO_3/TiO_2+CaCO_3$, in combination with the analysis of sulfur species over $TiO_2+CaO/TiO_2+CaCO_3$, active intermediates derived from $TiO_2+CaCO_3$ are speculated to account for the increased sulfate production, as we stated in the main text. We have put the relevant discussion in the main text to solidify our argument.

2—General comments and concerns
Why would the authors expect that carbonate radical would promote sulfate oxidation more than hydroxyl radical produced by $TiO_2$? Perhaps I am missing something here, but if the idea is that carbonate is scavenging OH to produce carbonate radical, then why would the overall S(IV) to S(VI) conversion be any different?
Response to Q: Thanks for your thoughtful question. DFT calculations produce theoretical evidence that carbonate radical ions decrease the energy barrier for $SO_3^-$ formation, and its reaction with $SO_3^{2-}$ is thus faster than that with hydroxyl radical (Fig. 8 b and c). Considering this, $CO_2$ and carbonate ions severs as a precursor of $CO_3^-$, and thus increase sulfate production. Therefore, overall S(IV) to S(VI) conversion mediated by carbonate radicals is faster than that by hydroxyl radicals.

The experiments shown in the main experimental figure (Figure 1) were performed at almost 10 ppm $SO_2$. The mismatch between these experimental conditions and those relevant for the actual atmosphere warrants discussion. What challenges do the authors anticipate when using these results to make predictions about behaviour under more realistic conditions?
Response to Q: We understand your concern about the gap between the lab and realistic conditions regarding $SO_2$ concentration. Therefore, we conducted the $SO_2$ concentration dependence experiments, and the reaction order of $SO_2$ in "$TiO_2+CaCO_3+SO_2$" reaction system in the range of 400-20000 ppb is determined to be 0.80 (**Fig. R1.**), indicating that the uptake coefficients obtained at ppm level of $SO_2$ would somewhat overestimate the real one obtained at atmosphere relevant $SO_2$ concentration (ppb level of $SO_2$), and the uptake coefficients should be calibrated before being

employed for model simulation for sulfate.

[Figure]

**Fig. R1.** Reaction order determination. The Lg-Lg curve of the sulfate production rate of $TiO_2$ mixed with $CaCO_3$ (50 wt. %) upon varied $SO_2$ concentration exposure (400-20000 ppb, RH = 30 %) under irradiation (30 mW cm$^{-2}$) plotted against the concentrations of $SO_2$ molecules exposed.

Abstract

L18—Is there direct evidence for the proposed mechanistic pathways?
Response to Q: We have removed this statement from the abstract.

L20—Is there direct evidence for production of gas-phase carbonate radical?
Response to Q: We have changed our statement in a conservative way; that is "Importantly, upon irradiation mineral dust particles are speculated to produce gas-phase carbonate radical ions when the $CO_2$ of atmospherically relevant concentration presents"

Introduction

L29—These references seem somewhat out of date.
Response to Q: We have updated the citations.

L31—What does "unique chemical activity" mean? Are nanometer-sized particles relevant for dust chemistry?
Response to Q: This term means that the chemical activity of water-adsorbed particles is higher than that of dry particles since radical-initiated reactions occur over humidified particles.

Strong nucleation events are initiated by the strong dust events, and a high concentration of nano-sized particles along with aerosol nucleation is observed (Dupart et al., 2012; Guo et al., 2014; Sun et al., 2014).

Nano-sized particles are important constituents found in the atmospheric aerosol, with lifetimes of

up to several days and can be transported over thousands of kilometers (Tang et al., 2016a), which is a major contributor to ambient air pollution (Whiteside and Herndon, 2018) and employed for understanding heterogeneous sulfate production within dust chemistry (Ma et al., 2019; Shang et al., 2010; Wang et al., 2018).

However, we want to avoid to arise the undesired dispute due to the above argument, thus have changed to a conservative expression by replacing "nanometer-sized" with "humidified"; that is "However when atmospheric chemical reactions occur over humified particles at ambient conditions."

L33—Are these references for the aquatic environment? Or for the atmosphere? What does "over the water surface" mean?
Response to Q: While these references refer to an aquatic environment, the media of which can be also found in the atmosphere, e.g. cloud/fog drops and aerosol liquid water.

Water surface refers to the surface of aquatic/aqueous media.

L37—Higher selectivity than what? What selectivity does the carbonate radical have?
Response to Q: This term is redundant here, and we thus removed it from the current manuscript.

L40—Phenol is presumably only one organic compound that is degraded by carbonate radical—why was this one chosen to highlight here?
Response to Q: Thanks for your suggestion. We have removed specific highlight of an organic compound from the current manuscript.

L41—Are porphyrins relevant in an atmospheric context? Is this rate constant relevant? Higher than what?
Response to Q: Porphyrin is observed in the atmosphere (Hodgson and Baker, 1969) and is known as a functional pigment of biology (Kay and Gratzel, 1993).

While rate constant in the unit of $M^{-1}$ $s^{-1}$ is usually applied for describing the rate of reactions that occur in aqueous media, the order of $10^9$ stands for a fast reaction process, somehow reflecting the great oxidative potential of $CO_3^{-}$ in the atmospherically relevant aqueous media, e.g. fog, cloud, and aerosol water.

We revised the term "higher" to "high" to fulfill the context.

L42—"great oxidation capability that may trigger atmospherically relevant chemical reactions"—this is vague/unclear.
Response to Q: We change to "great oxidation capability that may trigger atmospherically relevant chemical reactions, e.g. secondary inorganic species formation."

L44—These references seem out of date as well. Are there any tropospheric modelling-type papers (CAPRAM? GAMMA?) that provide carbonate radical rate constants?

Response to Q: Thanks for your suggestion. We have updated citations, adding recent modeling works in the current main text.

Ge and coworkers investigated the effect of in-cloud aqueous-phase chemistry on $SO_2$ oxidation using Community Earth System Model version 2 (CESM2), the improved run of which incorporates aqueous $CO_3^-$ chemistry to estimate sulfate production (Ge et al., 2021). It is noted that the most of aqueous $CO_3^-$ chemistry and corresponding rate constants applied in the study are derived from many early works (before the year 2000). Another work using CAPRAM modeling (Herrmann et al., 2000) also reported the production and the loss flux of the carbonate radical anion in the troposphere, and a series of sulfur chemistry have been considered.

However, some of the rate constants in the above two works are much lower than that are derived from theoretical calculations in our study. Besides, this estimation does not incorporate the role of mineral dust particles, which can significantly increase the production of $CO_3^-$, as suggested in this study and our early work (Fang et al., 2021).

L53—Where does this underproduction occur? Under what sorts of conditions? Also, what about other sulfate production pathways, e.g., the DMS chemistry being explored in Timothy Bertram's group?
Response to Q: In this work, sulfate underproduction during the dust storm episode is the scenario we are mainly concerned about (Dong et al., 2016; Huang et al., 2014; Yu et al., 2020). During the dust storm period, mineral dust components are abundant and the photochemical process under low RH conditions can also produce substantial sulfate. A more detailed discussion is as follows:

Long-term field studies have suggested underlying enhancements of sulfate production due to mineral aerosols and found a high consistency between sulfate and calcium in Asia dust particles (Arimoto et al., 2004; Li et al., 2013; Wang et al., 2012). Besides, it is believed that the evident increase of $Ca^{2+}$ in Beijing and Tangshan relates to dust storm events (Li et al., 2013; Wang et al., 2005). Many early studies have shown that carbonate-containing particles with high alkalinity are prevalent in the troposphere during dust storm episodes (Abou-Ghanem et al., 2020; Li et al., 2014; Tang et al., 2016a). $TiO_2$ was found at mass mixing ratios ranging from 0.1 to 10 % depending on the exact location where dust particles were uplifted (Chen et al., 2012; Hanisch and Crowley, 2003).

The photochemical process that occurs during dust storm events can alter atmospheric constituents (Liu et al., 2022). By employing Atmospheric Mineral Aerosol Reaction (AMAR) model, Jang's group highlights the significant contribution of heterogenous photochemical reaction that occurs over dust particles to overall sulfate formation (Yu and Jang, 2018; Yu et al., 2017), accounting for 79-93 % of total sulfate production, while the sulfate production that comes from the gas-phase oxidation and aqueous-phase oxidation or heterogeneous reactions in the dark case accounts for the residual of overall sulfate yield (Yu et al., 2017).

Besides, during dust storm episodes, the air mass is usually in low relative humidity, reported to be 25-35 % (Al-Salihi and Mohammed, 2015) (Csavina et al., 2014; Najafpour et al., 2020).

Given above, a detailed revision of the main text presents as follows:

"Atmospheric models fail to capture the key feature of atmospheric observations of high sulfate production during dust storm episodes in the troposphere (Dong et al., 2016; Yu et al., 2020), where an evident increase of $Ca^{2+}$ (Li et al., 2013; Wang et al., 2005), carbonate-containing particles with high alkalinity (Abou-Ghanem et al., 2020; Li et al., 2014; Tang et al., 2016a) as well as photoactive mineral components (Nie et al., 2012; Ta et al., 2003) are prevalent. Air mass is usually in low relative humidity, reported being 25-35 % (Al-Salihi and Mohammed, 2015) (Csavina et al., 2014; Najafpour et al., 2020), in these events, during which photochemical process is able to alter atmospheric constituents (Liu et al., 2022). Consequently, there are unknown heterogeneous reaction pathways of significance and previously unconsidered promoters that have great potential to accelerate sulfate formation in the dust storm relevant conditions."

Dimethylsulfoniopropionate (DMS) chemistry is illustrated to play important role in sulfate formation over the marine atmosphere (Kilgour et al., 2022; Mayer et al., 2020). Once emitted into the marine atmosphere, DMS is oxidized by gas-phase hydroxyl radical and halogen radicals to form lower volatility products, which can contribute to new particle formation after further oxidation to sulfate (Novak and Bertram, 2020). However, herein we mainly focus on the continental sulfate production over dust particles.

L55–60—"unconsidered heterogeneous mechanism is very likely to narrow the gap …" — to me, this seems speculative, if only one aerosol study is cited (Zheng 2015). Is it certain that heterogeneous pathways will close this gap?
Response to Q: Thanks for your suggestion. We have cited more references here; that is, "However, a remarkable missing sulfate budget emerges for the atmospheric modeling (Huang et al., 2019; Itahashi et al., 2018; Liu et al., 2021), which significantly underpredicts $SO_4^{2-}$ with respect to observational results when heterogeneous aerosol chemistry is not considered (Feng et al., 2018; Wu et al., 2021; Zheng et al., 2015)."

L66—Carbonate salt is enriched over what? What does "authentic dust aerosol" mean here?
Response to Q: As carbonate salt is one of the key constituents in mineral dust particles, reported to reach over 10 % wt. of Asian dust particles. Instead of being enriched, the word "abundant" is more appropriate in the context. Besides, we attempt to express the mineral dust aerosol found in the atmosphere using authentic mineral dust. This term, however, seems to be controversial and odd to readers. Taken together, we thus correct this sentence to "Carbonate salt is abundant in mineral dust particles".

L70—How does carbonate alkalinity "favour sulfate formation"? What exactly does this mean?
Response to Q: It is believed that the increase in alkalinity leads to an increase in $SO_2$ adsorption and subsequent oxidation of $SO_2$ to sulfate in the presence of oxidants. We have noted that the expression may cause difficulties for readers to follow, and we thus have corrected this expression in the current manuscript; that is "…intrinsic alkalinity, which buffers aerosol acidity and increases $SO_2$ adsorption and corresponding sulfate production in the presence of oxidants (Al-Hosney and Grassian, 2005; Bao et al., 2010; Kerminen et al., 2001; Li et al., 2007; Yu et al., 2018a)."

L71—Where is this information provided in the literature (that $CO_2$/carbonate produces carbonate radical)?

Response to Q: Thanks for your reminder, and we have supplied the reference accordingly.

For reference (Ervens et al., 2003), the relevant information is visible on page 15, section "4.5.7. Carbonate Chemistry".

For reference (Graedel and Weschler, 1981b), the relevant information is visible on page 510, section "3. Inorganic Carbon Chemistry".

L73—What does "fast $SO_2$ oxidation" mean here? Why is it likely to be a driving force?

Response to Q: Acceleration of $SO_2$ oxidation. In our early study (Fang et al., 2021), we provide experimental evidence that carbonate radicals produced over the mineral dust surface are able to promote $NO_2$ oxidation to secondary nitrate formation. Therefore, we tentatively believe that this active intermediate can serve as a driving force to accelerate $SO_2$ oxidation as well.

Here, we note that some information is missing to bridge "acceleration of $SO_2$ oxidation" and "a driving force", and we thus correct the sentence as follows:

"Our early study shows that carbonate radicals serve as an active oxidant to accelerate $NO_2$ oxidation over mineral dust particles, allowing us to consider the possibility that fast heterogeneous $SO_2$ oxidation can be triggered by this active intermediate as well."

L76—What does "extend their ability" mean?

Response to Q: "have the ability", and we have revised the expression in the current manuscript.

Methods

General—As I noted at the beginning of my comments, is there a reason why all of the techniques, etc., are in the supplement? It would be very helpful to have information regarding the samples (which "authentic dust" samples and which "authentic simulants"? What does "authentic" mean here?) as well as the experimental set-up in the main text. A reference for the carbonate radical measurement strategy is needed, as well (aniline as probe molecule).

Response to Q: We understand your concern about the inconvenience of reading flow when relevant techniques and information were placed in SI.

However, it would be tedious if we put all of the technical descriptions in the main text. Maybe cover 9-10 pages in length in the main text.

Considering this, we put part of the technique characterizations, which you have emphasized in the comments, into the section of "Experimental Methods". Supporting information on authentic dust and authentic simulants as well as experiment set-up have been moved into the main text. The term "authentic" refers to the "real", i.e. what is found in the atmosphere.

We have cited relevant references in the current manuscript.

Field Observations—Where is the department? Was sulfate determined in all size fractions? More information is needed here. Citations for the "ionization balance approach" are needed. Which

sulfate is expected to be non-water-soluble? Were the authors concerned about S(IV) to S(VI) conversion in the sample extracts? How was this addressed? Some of this information may be in the supplement, but as a reader, I do not want to have to shift back and forth between documents to find it.

Response to Q: More geographical information for sampling is available in our early study (Liu et al., 2020b) and we have mentioned the information and cited the reference. We have added relevant information to the main text.

We only considered the coarse mode of sulfate since the heterogeneous reaction of $SO_2$ on the dust surfaces is believed to be a major contributor (Liu et al., 2017). This correlates to the fact that a large mass fraction of mineral dust is abundant in coarse-model particulate matter (PM) (Fang et al., 2017; Miller-Schulze et al., 2015). Sulfate and (bi)carbonate in PM (3.3 μm ≤ Size ≤ 9 μm) were our major focus. We also determined the concentrations of sulfate and (bi)carbonate in PM (Size ≥ 9 μm) and performed a correlation analysis for comparison.

Citations for the "ionization balance approach" are now available in the current submission.

To our humble knowledge, non-water-soluble sulfate (non-WSS) is not usually considered in field observation probably due to its low content found in the particulate matter (Canelli and Husain, 1982). Therefore, we do not pay much attention to these species.

The content of S(IV) species is low relative to S(VI) species. In the early study, Dixon and Aasen determined both S(IV) including sulfite and HMS as well as S(VI) species in collected PM (Dixon and Aasen, 1999). They extracted the sample using methanol mixed with a preserving solution to prevent the unexpected conversion of S(IV) to S(VI). In all their considered sample, nearly 1/3 of samples are beyond detection limit or not detected. For others, S(IV) species are on the order of ng $m^{-3}$ whereas S(VI) species are on the order of μg $m^{-3}$, thus giving rise to more than two orders of magnitudes gap between S(IV) species and S(VI) species. This is consistent with observation where S(IV) is not detectable in the rain water due to a rather low sulfite concentration, which is beyond the detection limit of ion chromatography (Jin et al., 2020). The above results suggest the content of S(IV) is much lower than that of S(VI).

Hence, while we did not employ preservation procedures to prohibit the conversion of S(IV) to S(VI), we believe the unexpected S(IV) oxidation during the sampling/extraction steps gives a slight influence on determining S(VI) species.

Results

L102—At this point, no information about the mineral dust proxies has been presented. How are their properties "consistent with earlier studies"?

Response to Q: While we note that straightforward data shown in the main text would help readers to follow the conclusion we draw, it may distract readers from focusing on the main discussion flow instead. After all, they are just supplementary data to support our statement and the limited information we attempt to expand to discuss. For the characterization of mineral dust proxies, kindly you may understand our concern, allowing us to reserve the original arrangement. We have also

updated relevant information in the supporting information to support our argument in the main text.

In the supporting information, BET surface area, crystal phase, and structure have been characterized by using $N_2$ adsorption-desorption isotherm, XRD as well as Raman spectra. Our results match well with the early works.

L105—What is the experiment being discussed here? What kind of experimental conditions were employed? "Upon irradiation" of what? What were the $SO_2$ concentrations here? What was the humidity?

Response to Q: We have clarified the experimental conditions in the main text; that is "Upon solar irradiation under RH of 30 % $SO_2/N_2+O_2$ flow ([$SO_2$] $=2.21\times10^{14}$ molecules $cm^{-3}$), the sulfate production on $TiO_2+CaCO_3$ mixture particles (50 wt. % $TiO_2$ and 50 wt. % $CaCO_3$), measured by IC, is significantly enhanced by 7 times and 23 times compared to that of $TiO_2$ and $CaCO_3$ (Fig. 2a), respectively.".

L106—How are the authors distinguishing between S(VI) and S(IV) in these experiments? What quality assurance/quality control-type experiments were performed? How did the authors prevent S(IV) to S(VI) oxidation during extraction/prior to analysis? Was S(IV) quantified in these experiments? I would assume that it would have been present in all extracts … why are these data not shown in Figure 1?

Response to Q: For *in situ* DRIFTs technique, distinguishing S(VI) from S(IV) species over $TiO_2$ particles relies upon the position of the IR bands according to the assignment of previous literature (Nanayakkara et al., 2014; Wu et al., 2011).

For the IC technique, distinguishing S(VI) from S(IV) species relies on the retention time of the sulfur species. These two species can be distinguished by employing the optimized measurement method.

Our group accumulates a lot of measurement experience for *in situ* DRIFTs and IC techniques, especially for determining the concentration of S(IV) and S(VI) species. We have already benchmarked the measurement methodologies for each technique that has been applied for heterogeneous reaction studies (Liu et al., 2020b; Liu et al., 2020c; Wang et al., 2020a; Wang et al., 2020b; Wang et al., 2018).

In the experiments where we conducted heterogeneous reactions of $SO_2$ over $TiO_2$, $CaCO_3$, and $TiO_2+CaCO_3$ (50 wt. % $TiO_2$ and 50 wt. % $CaCO_3$) particles, we applied isopropanol to avoid unexpected conversion of S(IV) to S(VI).

We have put the quantified S(IV) results in the current submission. Sulfite species is only observable in $CaCO_3$ particles because it has no photoactivity while sulfite species is negligible in $TiO_2$ and $TiO_2+CaCO_3$ particles.

[Figure]

**Fig. R2.** Sulfate or sulfite concentrations quantified by IC on mineral dust particles after exposure to gaseous $SO_2$ under irradiation for 60 min. Reaction conditions: RH = 30 %, Light intensity (I) = 30 mW $cm^{-2}$, Total flow rate = 52.5 mL $min^{-1}$ and $SO_2$ = 2.21×$10^{14}$ molecules $cm^{-3}$.

L109—What does "it remains unclear for the role of carbonate salt …" mean?

Response to Q: Thanks for your kinder reminder. Our point is that a great discrepancy regarding sulfate production is observed between dark and irradiation experiments in which we conducted heterogeneous reactions of $SO_2$ over $TiO_2$+$CaCO_3$ particles. Therefore, $CaCO_3$ may play a different role in the dark and irradiation cases. However, this set of experiments is not solid enough to justify our speculation that carbonate salt plays a distinct role in these two scenarios.

We note that suddenly jumping into the argument "unclear for the role of carbonate salt" would be hard for readers to follow. We have corrected the original sentence to "Great discrepancies in sulfate production over $TiO_2$+$CaCO_3$ particles between dark and irradiation experiments suggest that carbonate salt may play a different role in these two scenarios."

L110—"There is a prevailing view" — reference? What does this statement mean?

Response to Q: In the current manuscript, we have revised this sentence to "The alkalinity of carbonate salt favors $SO_2$ adsorption (Al-Hosney and Grassian, 2005; Yu et al., 2018b) and photo-oxidation process assisted by $TiO_2$ particles can strengthen the oxidation efficiency of adsorbed $SO_2$ (Chen et al., 2012; Shang et al., 2010), which is a plausible explanation for rapid $SO_2$ oxidation over $TiO_2$+$CaCO_3$ particles."

L113—These data ($CaO$/$CaCO_3$ comparison) need to be presented clearly in the main text, since they are part of the authors' overall argument. "as they are likely to present similar physical and chemical properties" — what surface pH do they present? Are they comparable? Do they have similar hygroscopicities?

Response to Q: Thanks for your suggestion, and we have moved this data to the main text.

Our early study measured the pH of leaching solution of 0.625 mg $CaCO_3$ + 0.625 mg $TiO_2$ and

0.625 mg CaO + 0.625 mg $TiO_2$, and they are 9.27 and 11.26, respectively (Fang et al., 2021). On this basis, we could deduce that the surface pH of CaO is higher than $CaCO_3$.

You have raised the hygroscopicity issue in the early comment. For your convenience, we copy the answer, as follows:
"We compared the hygroscopicity of $CaCO_3$ and CaO by the specifying RHs that allow them to form a monolayer of water. A prior study shows that a monolayer of water formed on the surface of $CaCO_3$ particles when RH is over 52 % (Li et al., 2010). In contrast, RH of 27 % enables CaO to form a monolayer of water (Goodman et al., 2001). While those results are collected from different literature and the bias may come from different measurement systems. However, this evident gap leads us to conclude that CaO exhibits a stronger hygroscopicity than $CaCO_3$ does."

L118—I find this use of "theoretical" and "experimental" confusing.
Response to Q: While they are not commonly-used expression in the publications, we have defined them in the legend of Figure 2; that is "the production of sulfur species in theoretical $TiO_2$+$CaCO_3$ mixtures refers to $0.5 \times$ K-M bands of sulfur species of $TiO_2$ + $0.5 \times$ K-M bands of sulfur species of $CaCO_3$ while that for experimental $TiO_2$+$CaCO_3$ mixtures refers to $1 \times$ K-M bands of sulfur species of $TiO_2$+$CaCO_3$ mixtures (wt./wt. = 50/50).

To clarify these terms, allowing readers to follow, we have revised the original sentence in the main text to "The "theoretical" is calculated based on the DRIFTS experiments of pristine $TiO_2$ and $CaCO_3$ particles through a simple linear superposition whereas the "experimental" is directly derived from DRIFTS experiment of $TiO_2$+$CaCO_3$ (wt./wt. = 50/50) particles."

L125—Again, these data need to be presented in the text. The need to flip back and forth between the main text and the supplementary information is very frustrating for the reader.
How were these uptake coefficients determined? Are they surface-area scaled? How does this pathway compare to other heterogeneous $SO_2$ oxidation mechanisms? This statement (that this is "likely a potential driving force to trigger fast $SO_2$ oxidation …"seems like an overreach to me in the absence of supporting calculations.
Response to Q: Thanks for your suggestion and we have put relevant data in the main text. Detailed methodology for uptake coefficient calculation is available in the experimental section.

These uptake coefficients are derived from dividing the observed sulfate production rate by a total number of surface collisions per unit time ($Z$), and $Z$ is scaled by surface area.

Since we do not do the calculation of the relative contribution of this pathway to overall sulfate production or make a comparison with other pathways, we employed quite a mild tone to express that "the photochemical pathway associated with carbonate species is likely a potential driving force to increase sulfate production in the atmosphere."

Figure 1—How many trials were performed here? Why was this DRIFTS experiment not performed for $TiO_2$/CaO, to account for any changes in speciation after extraction? To me, the data presented in Figure 1b–c don't necessarily show that $CaCO_3$ is enhancing the photochemistry over that of

TiO$_2$ alone via carbonate radical … how does the surface area of these mixed films compare to that of the single-component films? What is the surface water content of each? Is it possible that the surface reaction environment is different? What is the S(VI)/S(IV) ratio under each condition? What does "yield" mean in the caption?

Response to Q: No less than three times unless the first two trails are highly consistent.

IC data we supplied in the supporting information (previous version) is complement evidence to support our argument in the main text. Together with DRIFTS analysis of sulfur species over TiO$_2$, CaCO$_3$, TiO$_2$+CaCO$_3$, in combination with IC analysis of sulfur species over TiO$_2$+CaO and TiO$_2$+CaCO$_3$, carbonate ions are speculated to play a different role in enhancing sulfate production in dark and irradiation cases. Therefore, we do not further perform the DRIFTS experiments for TiO$_2$/CaO particles.

In our experiments, we applied grounded TiO$_2$, grounded CaCO$_3$ as well as grounded TiO$_2$+CaCO$_3$ particles for all experiments and BET measurements. Our BET measurement results show that the BET surface area of grounded TiO$_2$ (50 % wt.) +CaCO$_3$ (50 % wt.) particles (23.52 m$^2$ g$^{-1}$) is slightly lower than the averaged surface area of TiO$_2$ (56.44 m$^2$ g$^{-1}$) and CaCO$_3$ particles (1.25 m$^2$ g$^{-1}$). Therefore, a total exposed surface area of 50 % wt. TiO$_2$ + 50 % wt. CaCO$_3$ is almost comparable to that sum of 50 % of TiO$_2$ and 50 % of CaCO$_3$, and surface area is not the fundamental force to increase sulfate production.

Similarly, we specify RHs that enable to form the monolayer water over various particles to compare hygroscopicity among different types of particles. For TiO$_2$, over 11 % of RH can have the particles form monolayer water (Haghighatmamaghani et al., 2019) while a monolayer of water forms on the surface of CaCO$_3$ particles when RH is over 52 % (Li et al., 2010). On this basis, increased water content due to the presence of CaCO$_3$ is the not reason that increases the sulfate production over the TiO$_2$+CaCO$_3$ mixture upon irradiation.

While we can not ensure that there is no change of properties during the grinding process, a mixture of 50 % wt. TiO$_2$ + 50 % wt. CaCO$_3$ is prone to present properties that combine both TiO$_2$ and CaCO$_3$ particles. Considering this, the DRIFTS analysis compares the spectra of 50 % wt. TiO$_2$ + 50 % wt. CaCO$_3$ mixture and linear superposition of spectra of TiO$_2$ and CaCO$_3$ components can reflect the underlying synergistic effect between TiO$_2$ and CaCO$_3$ particles.

By analyzing IC data, the ratios of S(IV) to S(VI) over TiO$_2$, CaCO$_3$, 50 % wt. TiO$_2$ + 50 % wt. CaCO$_3$ were determined to be 0, 1.94, and < 0.003, respectively. On the other hand, we compare the net S(VI) production for each particle, and TiO$_2$: CaCO$_3$: TiO$_2$ (50 % wt.) + CaCO$_3$ (50 % wt.) $\approx$ 1.87:0.28:14.60. Our results suggest that CaCO$_3$ and TiO$_2$ produce much less S(IV)+S(VI) than TiO$_2$: CaCO$_3$: 50 % wt. TiO$_2$ + 50 % wt. CaCO$_3$, indicating that there is an abnormal reaction channel that significantly increases sulfate production when these two components contact with each other.

L151—What does "a strong interaction" mean? What does "weak interplay" mean? These terms are vague/imprecise. Why did grinding not lead to a decrease in size of the CaCO$_3$ particles?

Response to Q: "A strong interaction" means that $TiO_2$ particles strongly adhere to $CaCO_3$ particles whereas "weak interplay" means that $TiO_2$ particles poorly adhere to $CaCO_3$ particles. We have changed those two terms to "Compact contact" and "Loose contact".

Consistent with the SEM observations in the early study (Christidis et al., 2004), the continuous grinding process brings a negligible change in crystal size of calcite within 15 min, probably due to the instinct hardness of calcite. In our study, particles were ground for 10 min for each mixture. Therefore, grinding does not lead to an evident change in the size of $CaCO_3$ particles.

We have changed the term "yield" to "production".

L155—Is it possible that the overall exposed surface area of the system was larger after grinding, or that aggregates were disrupted, and that this was the cause of the results in Figure S8 rather than anything relating to the specific interface/interactions between $TiO_2$ and $CaCO_3$?
Response to Q: Thanks for your thoughtful comments. We have considered this issue already. In our experiments, we applied ground $TiO_2$, ground $CaCO_3$ as well as ground $TiO_2+CaCO_3$ particles for all experiments and BET measurements. Our results show that the BET surface area of ground $TiO_2+CaCO_3$ particles is almost identical to the average value of BET surface area of ground $TiO_2$ and ground $CaCO_3$ particles. Therefore, the total exposed surface area is almost not changing and it is not the fundamental force to increase sulfate production.

L157—What is "fast production", exactly? This term is used throughout the manuscript, but in the absence of kinetic data / multiple timepoints, I do not have a sense of what "fast" means here.
Response to Q: We calculated $SO_2$ uptake coefficients for both $CaCO_3$, $TiO_2$, and $TiO_2+CaCO_3$, which were determined by the sulfate production rate using multiple timepoints. More details could be found in the experimental section. Seven and Ten timepoints during the heterogeneous reaction were applied for calculating the kinetics in "$TiO_2 +SO_2+(CO_2)$" and "$TiO_2+(CaCO_3)+SO_2$" reaction systems, respectively. Hence, we mentioned the concept "fast" in the context.

L164—What exactly is meant by "the rapid $SO_2$ oxidation pathway"? This echoes my previous comment (L157).
Response to Q: We have changed "the rapid $SO_2$ oxidation pathway" to "the increased sulfate production".

L169—What mechanism do the authors propose for the enhancement observed upon addition of $CaCO_3$ to $SiO_2/Al_2O_3$? Overall, I find this paragraph confusing; there are a lot of results presented in this manuscript that are briefly discussed / glossed over, and to fully convince the reader that all possibilities have been considered, all results should be discussed in terms of how they fit / do not fit with the overall/big-picture interpretation presented by the authors.
Response to Q: Thanks for your suggestion, we thus put more effort to discuss the results shown in our graph and give more interpretation to convince the readers that we have fully considered the possibility that would affect the conclusion/big-picture we deduced.

For this data, more discussion is presented as "In Fig. 4, the introduction of $TiO_2$ components ($\approx$

1 % wt.) into $SiO_2$-$Al_2O_3$ leads to 81.6 % enhancement of sulfate production because of the photolabile ROS. On the other hand, merely 24.8 % wt. increase of sulfate yield was observed once $\approx$ 8 % wt. of $CaCO_3$ was incorporated into $SiO_2$-$Al_2O_3$ dust particles. This can be attributed to the alkaline environment created by $CaCO_3$, which is believed to increase $SO_2$ adsorption (Al-Hosney and Grassian, 2005) and sulfate production accordingly. Surprisingly, mixing of $\approx$ 1 % mass fraction of $TiO_2$ and $\approx$ 8 % wt. of $CaCO_3$ into $SiO_2$-$Al_2O_3$ gives rise to a 235 % increase in sulfate formation relative to that of $SiO_2$-$Al_2O_3$. It represents that there is nearly an extra 100 % enhancement of sulfate production due to the presence of $TiO_2$ and $CaCO_3$ in the atmospherically relevant mass fraction. These results lead to the hypothesis that the observed synergistic effect on heterogeneous oxidation of $SO_2$ is likely to take effect in the atmosphere."

L174—"thus likely involving the reaction mechanism proposed in this work"—I do not follow this statement.

Response to Q: We attempt to express that alpha-$Fe_2O_3$ may produce electron-hole pairs upon irradiation, and further react with (bi)carbonate ions to form carbonate radicals. In this case, the $CO_3^{-}$ initiated $SO_2$ oxidation is also likely to occur in alpha-$Fe_2O_3$ particles.

L184—Recent work by Abou-Ghanem and co-workers (ES&T 2020) has shown that Ti in mineral dust differs substantially from commercial $TiO_2$. In this context, I don't think it is reasonable to state that 1% $TiO_2$ is "atmospherically relevant" without qualification of some sort. I also don't know what "authentic dust simulants" means here.

Response to Q: Thanks for your thoughtful comments. We have gone through the literature you recommended and it indeed helps us to have a better understanding of titanium-containing dust particles (Abou-Ghanem et al., 2020). While we may not ensure that the mass percentage (%) of $TiO_2$ in Ti-containing mineral dust particles always goes above 1%, the mass percentage of $TiO_2$ in authentic PM collected from aerosol sampling sites was analyzed and determined to be 5 % (Engelbrecht et al., 2009). On this basis, we believe 1 % of $TiO_2$ is plausible to be found in the atmosphere.

On the other hand, we also considered the crystal phase of $TiO_2$. Anatase and rutile are two of the three naturally occurring $TiO_2$ polymorphs whereas the third brookite is an uncommon phase of $TiO_2$ (Abou-Ghanem et al., 2020; Jaffe and Howard, 1996). In our experiment, commercial-available $TiO_2$ particles comprise both anatase and rutile. Combing with a possible 5 % of $TiO_2$ found in the PM, we suppose 1 % mass fraction of anatase and rutile $TiO_2$ is likely to exist in the PM.

We have changed the term "authentic dust simulants" to "mineral dust simulants", which refers to the simulants that include major crustal components, i.e. $SiO_2$ and $Al_2O_3$.

L186—What does "fast oxidation channel" mean? What does "beyond the conventional regime of alkaline neutralization of $H_2SO_4$" mean?

Response to Q: We have changed to "increased sulfate production". We want to say that the conventional reaction mechanism involving alkaline carbonate salt increases sulfate production mainly through the neutralization process. In this study, we propose that an alkaline environment is

not the fundamental driving force to increase sulfate production, and carbonate salt serves as the precursor of carbonate radical, contributing to the increased sulfate yield. This is beyond the conventional reaction scheme.

L190—Which "mineral dust" was used here? Sulfate "yield" on what surface? The $CO_2$ is atmospherically relevant, but what about the $SO_2$ concentrations? This (the concentration dependence) should be discussed in the main text, rather than shown only in the supplementary information, because the experiments shown in Figure 1 were conducted at 10 ppm $SO_2$. Why was this the case, and how might the results obtained under these highly unrealistic conditions differ from those in the real atmosphere?

Response to Q: Mineral dust proxies $TiO_2$ were used here, and we have clarified the information in the main text; that is "Its influence on photochemical $SO_2$ uptake on mineral dust proxies $TiO_2$ was thus studied."

[Figure]

**Fig. R3.** Reaction order determination. The lg-Lg curve of the sulfate production rate of $TiO_2$ in the presence of $CO_2$ (400 ppm) upon varied $SO_2$ concentration exposure (400-20000 ppb, RH= 30 %) under irradiation (30 mW cm$^{-2}$) plotted against the concentration of $SO_2$ molecules exposed.

Thanks for your suggestion. We understand your concern about the gap between the lab and realistic conditions regarding $SO_2$ concentration. Therefore, we also conducted the $SO_2$ concentration dependence experiments for "$TiO_2+CO_2+SO_2$" system, and the pseudo-first reaction order (1.13) was determined in the selected concentration range (400-20000 ppb), indicating that the reaction kinetic obtained in the ppm level is slightly overlooked compared to that conducted in the ppb range. While we note that the difference in $SO_2$ concentration between the lab and atmospheric conditions remains, employing hundreds of ppb $SO_2$ in the laboratory simulation to obtain the kinetic parameter of sulfate formation is acceptable (Liu and Abbatt, 2021; Liu et al., 2020a). Taken above, we tentatively believe that uptake coefficients estimated under ppm level are valid, and these datasets derived from laboratory chambers are able to be generalized to the atmosphere condition.

For 10 ppm we employed in "$TiO_2+CaCO_3$" reaction system, we have performed $SO_2$ dependence experiment as well. We have replied to this question in your early comment.

L193—I don't follow these arguments regarding the effects of gas-phase $CO_2$. How can these observations be directly related to observations with carbonate particles? I would think that the effect of $CO_2$ might relate to a competitive adsorption effect rather than being directly comparable to the effect of pre-existing (solid) carbonate on $SO_2$ uptake …

Response to Q: Thanks for your suggestion. We have changed the original sentence to "$CO_2$ suppresses both S(IV) and S(VI) products under the dark probably due to the competitive adsorption effect, as we observed over $Al_2O_3$ particles (Liu et al., 2020b)."

L203—What is the speciation of this $TiO_2$? Elemental Ti content is not necessarily an accurate predictor of photoreactivity.

Response to Q: Thanks for your comments. Though we do not do the specimen of $TiO_2$ in these mineral dust particles, the results from previous literature are able to provide some useful information.

Only the anatase phase of $TiO_2$ is observable in Kaolin (K-Ga-2) through differential thermal analysis, and chemical analysis shows that $TiO_2$ accounts for 2.08 % wt. of the total mass of K-Ga-2 (Johnson et al., 1982), consistent with early observation where anatase is general phase found in Kaolin clay (Weaver and Minerals, 1976). For Illite (IMt-2), $TiO_2$ is in rutile phase and accounts for 0.87 % wt. of total mass (Gailhanou et al., 2012). $TiO_2$ phases of anatase and rutile altogether account for Ti-containing components in Arizona test dust (ATD) particles (Joshi et al., 2017), where the 0.3 % wt. of $TiO_2$ is determined in the work of Joshi and coworkers and 0.5-1 % of $TiO_2$ is provided by supplier.

Anatase $TiO_2$ exhibits more efficient production of hydroxyl radicals than rutile $TiO_2$ in the presence of adsorbed hydroxyl groups and water layers (Buchalska et al., 2015), with more efficient production of carbonate radicals correspondingly. This gives an alternative explanation why in the presence of $CO_2$ under irradiation Kaolin (K-Ga-2) exhibits a pronounced increase of sulfate production than Illite (IMt-2) and Arizona test dust (ATD). The content and proportion of the active phase of $TiO_2$ altogether contribute to a pronounced increase of sulfate production, as you suggested in this comment in which Ti content is not necessarily an accurate predictor of photoreactivity. We have added relevant discussion in the main text; that is "On the other hand, $TiO_2$ content is not necessarily an accurate predictor of photoreactivity, the content and proportion of active phase of $TiO_2$ in K-Ga-2 altogether contribute to a more pronounced increase in sulfate production relative to other two clays (see detailed discussion in supplement text 18)."

L205–210—I find this argument difficult to follow. Are these results scaled to the surface area presented by each of these samples? I do not see how they can be quantitatively compared in this manner otherwise.

Response to Q: Although a comparison of sulfate production among these three dust particles is not the scope and intention of this graph, we understand your concern.

To address the problem, we made a rearrangement of the graph (Fig. R4), and sulfate production in each mineral dust is scaled by the mass for a quantitative comparison.

[Figure]

**Fig. R4.** Laboratory studies of sulfate production on authentic dust and clay membranes **(a)** K-Ga-2 **(b)** ATD as well as **(c)** IMt-2 under the dark and irradiation (30 mW cm$^{-2}$) upon exposure to 4.91×10$^{14}$ molecules cm$^{-3}$ SO$_2$/N$_2$+O$_2$ and 2.46×10$^{18}$ molecules cm$^{-3}$ CO$_2$+ 4.91×10$^{14}$ molecules cm$^{-3}$ SO$_2$/N$_2$+O$_2$ at RH of 30 %. Noting that sulfate yield in three cases was normalized by the mass of dust particles employed for heterogeneous reaction.

L231—Where is the evidence that "two water layers" "absorb" (adsorb?) on dust particles? On line 225, I find the terms being used here imprecise … what do "capture SO$_2$ in the gas phase first" and "then stabilize it as adsorbed S(IV)" mean?
Response to Q: Thanks for your careful check on our manuscript. It should be "adsorb".

In the early context, we have added relevant citations to support our point that two water layers are prone to adsorb on dust particles; that is "As the RH increases beyond 10 % -15 %, multilayer water coverage occurs, reaching approximately two monolayers at RH of 30 % (Mogili et al., 2006)." In our experiments, employing RH of 30 % is tentatively assumed to provide two water layers over dust particles.

As we do not have direct evidence to validate the two water layers adsorbed on the dust surface, we thus put this sentence in a mild tone; that is "two water layers are likely to absorb on dust particles."

We have revised the expression to "the resulting hydroxyl groups react with gaseous SO$_2$ to form adsorbed S(IV)$_{ad}$ species."

L247—These aniline results should be presented in the main text. What does "a promoted degradation" mean?
Response to Q: Thanks for your suggestion, and we have put this result in the main text. The degradation rate of aniline is increased. We have changed it to "An increased degradation rate of aniline".

L245—I find the paragraph starting on this line extremely difficult to follow. Specifically, I can't decipher the argument relating to the scavengers. In addition, how do these conditions (dust suspension in water) compare to the conditions explored in the previous sections of the manuscript (solid particles)? Is it reasonable to use information gathered in aqueous suspensions to interpret results obtained at the surface of solid mixtures? The potential limitations/biases in this approach should be addressed in the text.

Response to Q: We understand your concern about the inconsistency between aqueous media and water layers on humidified dust particles. We have noted this issue, and thus introduce the relevant background information in this paragraph, i.e. "As the RH increases beyond 10 % -15 %, multilayer water coverage occurs, reaching approximately two monolayers at RH of 30 % (Mogili et al., 2006). Under these circumstances, the amount of water adsorbed onto the surface of the dust particles is believed to be sufficiently large that it is liquid-like in its physical and chemical properties (Cwiertny et al., 2008) (Peters and Ewing, 1997). In this work, heterogenous $SO_2$ oxidation over mineral dust proxies proceeds at the RH of 30 %, and two water layers are likely to attach to dust particles. Thus, radical ions are anticipated to play a key role in fast $SO_2$ oxidation, and mechanism studies performed in the aqueous phase are persuasive to some extent."

Following this, we performed scavenger experiments to further validate our hypothesis that carbonate radical ions provide an alternative pathway to enhance sulfate production. Isopropanol (i-PrOH), known as the scavenger of hydroxyl radicals, was applied. In the absence of carbonate ions in $TiO_2$ suspension, hydroxyl radicals are a major contributor to oxidizing S(IV) to S(VI). Therefore, a great loss of sulfate production can be observed when carbonate ions are absent in the reaction system after adding i-PrOH. On the contrary, the introduction of carbonate ions reduces the loss of sulfate production, indicating that carbonate radical ions take effect and provide an alternative route to oxidize S(IV) to S(VI) even with i-PrOH.

L305—I do not understand the logic underlying these proposed reactions (in particular, equations 6–8). Where is the evidence for these species?
Response to Q: Through NTAS and ESR analysis, we produce experimental evidence that $SO_3^{-}$ increases due to the presence of $CO_3^{-}$. In fact, $SO_3^{-}$ chemistry has been extensively investigated by numerous studies looking at atmospheric chemistry (Gankanda et al., 2016; Hung and Hoffmann, 2015; Hung et al., 2018). When both oxygen and aqueous medium are available, these chain reactions (Eqs. 6–8) that we proposed in the main text are the most plausible pathway to take place in the reaction system of our concern.

$$SO_3^{-} + O_2 \rightarrow SO_5^{-} \tag{Eq. 6}$$

$$SO_5^{-} + SO_3^{2-} \rightarrow SO_4^{-} + SO_4^{2-} \tag{Eq. 7}$$

$$SO_4^{-} + SO_3^{2-} \rightarrow SO_4^{2-} + SO_3^{-} \tag{Eq. 8}$$

In Gankanda's work, they proposed the chain reactions (Eqs. 6–8) over mineral dust particles although they do not observe intermediate $SO_3^{-}$ and $SO_5^{-}$ (Gankanda et al., 2016). In the review of Grassian's group (Rubasinghege et al., 2010), they also suggest $SO_3^{-}$ chemistry, including chain reactions (equations 6–8), can account for sulfate production over mineral dust particles.

While we do not provide direct evidence for these species, previous relevant works allow us to speculate on these reaction steps. We noted that we should introduce necessary background information on $SO_3^{-}$ chemistry to readers. This would help them to follow the reaction mechanism we proposed in the main text. Specially, we have added background information in the main text;

that is "Based on the above results, one may deduce that the interplay between carbonate radical and sulfite ions is a crucial step giving rise to the increased $SO_3^-$ (Eq. 5), which is reported to account for rapid atmospheric sulfate formation through chain propagation reactions (Eqs. 6-8) (Hung and Hoffmann, 2015; Hung et al., 2018). Additionally, this sulfite radical ion chemistry is believed to drive fast sulfate formation over mineral dust particles (Gankanda et al., 2016; Rubasinghege et al., 2010)."

L325—I do not at all follow how these concentrations (for carbonate/hydroxyl radicals) were chosen here. The selected references do not make sense to me—one title is "The carbonate radical is a site-selective oxidizing agent of guanine in double-stranded oligonucleotides." How is this relevant to the argument regarding the relative concentrations of these oxidizing species?

Response to Q: Thanks for your careful check on our manuscript, and we have removed this reference from the current manuscript.

We have cited another reference relevant to our argument; that is "Oxidative Transformations of Contaminants in Natural and in Technical Systems". In this work, Sulzberger and coworkers suggest that the concentration of carbonate radicals can be two orders of magnitude higher than that of hydroxyl radicals in aqueous media, consistent with the concentration gap between carbonate radicals and hydroxyl radicals through the partitioning process from gas-phase determined in our reaction system (over 1.8 orders of magnitude, Fig. 9). While the net concentrations of carbonate hydroxyl radicals in the water layers of humified particles are very likely to be different from that found in the bulk aqueous media, concentration inputs of two radicals with the gap of two orders could reflect the relative contribution of carbonate radicals and hydroxyl radicals to sulfate production based on literature and our experimental trails.

L350–367—I think that this section would be much strengthened by addition of discussion regarding the way(s) in which performing experiments at such elevated $SO_2$ concentrations may have altered the surface pH of the particles employed.

Response to Q: Thanks for your suggestion. The detailed discussion has been added to the main text as follows:

"Considering that $SO_2$ concentration employed in this work is higher than that in the real atmosphere, the concept of "equivalent exposure time" is introduced to evaluate the influence of pH on the $SO_2$ oxidation pathway initiated by $CO_3^-$ (see a more detailed discussion on determining equivalent exposure time in supplementary text 22)."

"Besides, 20 ppb is assumed to be atmospherically relevant concentration to calculate "equivalent exposure time" in this study whereas an even low concentration (several or a few tens ppb of $SO_2$) is monitored in the field observation (He et al., 2014; Watanabe et al., 2020). Therefore, reduction of dust surfaces pH would be more moderate than we now considering and even little influence of surface pH on our proposed reaction scheme would have."

L369—What does a "non-negligible contribution to sulfate aerosol formation" mean? Compared to what?

Response to Q: We have changed the statement to "underlying pathway for sulfate aerosol formation"

L375—This information should be presented in the methods.
Response to Q: Thx for your suggestion, and we have put the relevant information into the method section.

L378—I do not understand how the maximum OH and carbonate radical concentrations were determined here (in this paragraph). Also, are there any other possible reasons (other than carbonate radical release) why the aniline loss may have been larger in the presence of $CO_2$?
Response to Q: In the reaction system we have been considering, carbonate radical ions are the most plausible oxidants that lead to the degradation of aniline.

In fact, without the involvement of $TiO_2$ particles, the degradation rate for "Aniline+Photolysis+$CO_2$+Air+Irradiation" system ($-6.33\times10^{-5}$ $s^{-1}$) is slightly slower than that for "Aniline+Photolysis+Air+Irradiation" system ($-7.18\times10^{-5}$ $s^{-1}$). While we can not explain the difference, this result however suggests the carbonate radical ion that is derived from the interaction between $TiO_2$ and (bi)carbonate under irradiation is the driving force in accelerating the loss of aniline. After all, carbonate radical is reported to react fast with aniline. If the presence of $CO_2$ prohibits aniline photolysis or air stripping processes, that means aniline degradation due to carbonate radicals contributes more than we are now considering.

More methodologies, sample preparation as well as measurement details including experimental set-up and description of measurements could be found in the experimental section. The major discussion on determining the maximum concentrations of hydroxyl and carbonate radical ions were demonstrated as follows:

"In the reaction system containing $TiO_2$ film upon irradiation (the UV wavelength = 310 nm) in the presence of humidified air (RH = 30 %), when operated in a continuous mode, the overall degradation rate of probe molecular in the presence of $TiO_2$ film can be described by Eq. **4** (Wang et al., 2004):

$$k_{obs}=\frac{d[An]}{dt}=r_A+r_U+r_{ROS}=r_A+r_U+k_{ROS+AN}[ROS][An] \qquad \textbf{[4]}$$

Where $k_{obs}$ is the observed degradation rate of aniline, [An] is the concentration of aniline, denoted as [An] hereafter, and $r_A$, $r_U$, $r_{ROS}$ stand for aniline removal rates resulting from air stripping, UV photolysis, ROS oxidation. $k_{ROS, An}$ are the overall second-order reaction rate constants for An with ROS.

Reference experiments without the introduction of ROS were also conducted to measure $r_A+r_U$ in each reaction system, e.g. "An+$TiO_2$+Air+Irradiation", "Aniline+$TiO_2$+$N_2$+Irradiation", etc. Upon irradiation, the dust proxy $TiO_2$ produces hole-electron pairs, further forming ·OH radicals and superoxide radicals ($O_2^-$) in the presence of absorbed water and oxygen molecules. Thus, a complement experiment using $N_2$ was adopted to investigate the role of $O_2^-$ in consuming aniline. As shown in Fig. 9a, a slight change in the degradation rate of aniline after stripping oxygen from the air, indicating that $O_2^-$ shows quite a smaller contribution than ·OH. This result agrees well with the finding reported by Durán et al. (Duran et al., 2019), where the removal of $O_2^-$ by adding

benzoquinone (BQ) into $TiO_2$ suspension results in the negligible change of An degradation rate. Taken above, ·OH radicals are assumed to be the only active ROS that accounts for the An degradation. Hence, the maximum steady concentration of ·OH radicals can be given by the following expression (Eqs. 5-7):

$$-\frac{d[An]}{dt}=k_{exp}[An]=k_{·OH, An}[·OH]_{ss-max}[An] \tag{5}$$

Integration of Eq. **S3** yields

$$-\ln\frac{[An]_t}{[An]_0}=k_{exp}t \tag{6}$$

$$k_{exp}=k_{·OH, An}[·OH]_{ss-max} \tag{7}$$

Together with the reported second-order rate constant ($k_{·OH, An} = 6.5\times10^9$ $M^{-1}s^{-1}$) (Samuni et al., 2002), the steady-state OH radical concentration $[·OH]_{ss-max}$ in buffered An solution can be calculated from eq. (1). The observed degradation rate constant of $k_{exp}$ can be obtained from the slope of the semi-log plot of An degradation as shown in Eq. (2). The maximum steady-state concentration of ·OH radical ions supplied by the partitioning process from gas phases was thus estimated to be $2.15 \times10^{-15}$ M for the $TiO_2$+Air system.

When $CO_2$ (400 ppm, atmospheric relevant concentration) is introduced into a flow-cell chamber, an increased degradation rate of An is seen, which can be attributed to the generation of active carbonate radical ions (Fig. 9b). Similar to the method we adopted for the estimation of $[·OH]_{ss-max}$, reference experiments were conducted to determine the rates for air stripping and UV photolysis processes in the "$TiO_2$+Air+$CO_2$" system. In the next step, we quenched the hydroxyl radicals by adding tertiary butanol (TBA). This is because it reacts rapidly with hydroxyl radicals (Li et al., 2020) $k_{·OH, TBA}= (6 \times10^8$ $M^{-1}$ $s^{-1})$ while showing a rather low reaction rate with carbonate radicals (Liu et al., 2015) ($k_{CO_3^{·-},TBA}<1.6 \times10^2$ $M^{-1}$ $s^{-1}$). Subsequently, we determined $[CO_3^{·-}]_{ss}$ using the previous protocol (Huang and Mabury, 2000) with known $k_{CO_3^{·-}, AN}$ $(5.4 \times10^8$ $M^{-1}$ $s^{-1})$ (Wojnarovits et al., 2020). In the extreme case, assuming that all hydroxyl radical ions were fully trapped by absorbed and dissolved $HCO_3^-/CO_3^{2-}$ over $TiO_2$ film and gaseous water molecular in the humidified airflow, the maximum steady-state $CO_3^{·-}$ radical concentration was determined to be $1.39 \times 10^{-13}$ M for "$TiO_2$+Air+$CO_2$" system, matching well with the early study where the concentration of carbonate radical can be two orders of magnitudes than ·OH over the water surface (Sulzberger et al., 1997)."

L391—I don't understand how carbonate radical ions "strengthen" the oxidative capacity of $TiO_2$-containing dust particles.

Response to Q: Here is a typo, and we have changed "carbonate radical ions" to "(Bi)carbonate ion". In this work, we show that increased sulfate production can be attributed to the formation of carbonate radical ions over dust particles that contain active component $TiO_2$ in the presence of bi(carbonate) ions upon irradiation. Following this, we deduce that (bi)carbonate ions strengthen the oxidative capacity of $TiO_2$-containing dust particles with regard to $SO_2$ oxidation.

L410—How were S(IV)/S(VI) ratios preserved after sampling/during extraction/analysis?

Response to Q: Thanks for your question. You have raised the question in the early comment. For your convince, we have pasted the answer here.

"The content of S(IV) species is low relative to S(VI) species. In the early study, Dixon and Aasen determined both S(IV) including sulfite and HMS as well as S(VI) species in collected PM (Dixon and Aasen, 1999). They extracted sample using mixing methanol with a preserving solution to prevent the unexpected conversion of S(IV) to S(IV). In all their considered sample, nearly 1/3 of samples are beyond detection limit or not detected. For others, S(IV) species are on the order of ng $m^{-3}$ whereas S(VI) species are on the order of μg $m^{-3}$, thus giving rise to more than two orders of magnitudes gap between S(IV) species and S(VI) species. This is consistent with observation where S(IV) is not detectable in the rain water due to a rather low sulfite concentration, which is beyond the detection limit of ion chromatography (Jin et al., 2020). The above results suggest the content of S(IV) is much lower than that of S(VI).

Hence, while we did not employ specific preservation procedures to prohibit the conversion of S(IV) to S(VI), we believe the unexpected S(IV) oxidation during the sampling/extraction steps gives a slight influence on determining S(VI) species."

L422—What does "undesired processes" mean?
Response to Q: We attempt to express some collected samples may experience a heterogeneous reaction process during the mixed periods that combine both daytime and nighttime hours, which is the undesired case. This is because it will bring bias and uncertainty to examining the relationship between sulfate and (bi)carbonate ions in two periods.

We note this is a confusing phrase that would frustrate readers to follow, and we thus have revised the sentence to "Some of the collected sample, experiencing heterogeneous reaction that takes place during the day(nigh)-night(day) shifts periods, inevitably being assigned to the sulfate ions measured in separate sampling hours, thus reducing the correlation coefficients."

L446—I do not understand how the negative correlations discussed here and the $CO_2$ suppression results are related to one another.
Response to Q:
In our sampling, we mainly focus on the PM collected from the first four sampling stages (particles size ≥3.3 μm). Previous field observation suggests mineral dust particles are dominant (45 %) in coarse inorganic mass fraction of PM (2.5 μm ≤ particles size ≤ 10 μm) (Bougiatioti et al., 2013).

In some sense, competitive adsorption between $CO_2$ and $SO_2$ over PM is supported by our early observations where $CO_2$ inhibits the $SO_2$ uptake on $Al_2O_3$ particles regardless of low and high water content formed, more precisely upon exposure of humified $CO_2+SO_2/N_2+O_2$ flow (5% and 95 %) (Liu et al., 2020b). $Al_2O_3$ is the major crust constituent found in the mineral dust (≈15 %) (Usher et al., 2003). We produce experimental evidence that there is competitive adsorption between $CO_2$-derived (bi)carbonate and sulfate species, and this gives rise to the reduction of $SO_2$ uptake on humified $Al_2O_3$ particles. Besides, $SiO_2$ particles are the most dominant constituent of mineral dust particles (≥ 60 %) (Ji et al., 2015). We supplied the data showing that $CO_2$ of atmospherically relevant concentration decreases the sulfate production over $SiO_2$ particles in the presence of gaseous $H_2O_2$ (Fig. R5). $H_2O_2$ is introduced into the reaction system as $SiO_2$ has rather lower $SO_2$ uptake coefficients ( $< 1\times10^{-7}$ ) than other crust constituents, e.g. (α-$Al_2O_3$ ≈ $1.6\times10^{-4}$ and α-$Fe_2O_3$

$\approx 7.0 \times 10^{-5}$) (Crowley et al., 2010; Usher et al., 2002). The difference in sulfate production in two reaction systems ("$SiO_2+SO_2$" and "$SiO_2+CO_2+SO_2$") can not be easily observed otherwise. The heterogeneous reaction of $SO_2$ has been investigated over black carbon particles as well, and active sites are believed to drive sulfate production (He et al., 2017; Xu et al., 2015). While less knowledge is available for heterogeneous reactions on organic fraction of PM, our early study focusing on the sulfate production over atmospheric Humic-Like Substances (HULIS) reported that $SO_2$ consumes active sites of HULIS (Liu et al., 2020c). On this basis, $CO_2$ is likely to behave the same as $SO_2$ does and competitive adsorption between $CO_2$ and $SO_2$ is expected over HULIS as well, or maybe over other organic matters with similar physicochemical properties.

Taken from the above discussion, we speculate the competitive adsorption between different trace gases is likely to occur over particulate matter, especially for trace gas $SO_2$ of several to dozens of ppb level in the presence of $CO_2$ of 400 ppm level.

[Figure]

**Fig. R5.** Sulfate concentration quantified by IC on mineral dust particles after exposure to gaseous $SO_2$ under irradiation for 30 min in presence of $H_2O_2$ gas flow. Reaction conditions: Total flow rate = 52.5 mL $min^{-1}$, $SO_2$ = $2.21 \times 10^{14}$ molecules $cm^{-3}$ and $CO_2$ = $9.83 \times 10^{15}$ molecules $cm^{-3}$. To produce gaseous $H_2O_2$ flow, an air flow was humidified in a bubbler loaded with 100 mM $H_2O_2$. The detailed protocol is similar to the one we applied for sulfate production over $TiO_2$ particles (Supplement Text 3). We performed two sets of experiments with a duration time of 30 min and 60 min, respectively.

L469—I don't understand the statement regarding "$CO_2$-derived bicarbonate species". Would the authors expect $CO_2$ to occupy PM surface sites? What about $CO_2$ dissolving in an aqueous layer? How do the authors know that the bicarbonate species are $CO_2$-derived?
Response to Q: Yes, we suppose the $CO_2$-derived bicarbonate species occupy PM surface sites, giving rise to the negative correlation between sulfate and (bi)carbonate ions.

When liquid water content (LWC) of particulate matter is sufficiently low, sites of particle surface dominate the $SO_2$ uptake. On the other hand, aqueous-like media largely determines the $SO_2$ oxidation rate if LWC over PM is high (Wang et al., 2020). Competitive adsorption occurs in both above two cases, where the former can be explained by the maximum surface coverage that allows

for gas uptake (Al-Hosney et al., 2005; Grassian, 2008). In the latter one, PM with high LWC sometimes comes from the PM with low LWC. Fresh particulate matter is usually dry when emitted into the atmosphere and suppression of $SO_2$ adsorption and sulfate formation correspondingly in the early emission stage. These particles become wet by adsorbing water over time (Khlystov et al., 2005). Adsorbed sulfate under dry conditions enters into the semi-aqueous layer, i.e. water layers over humidified PM. Since bi(carbonate) species hinder sulfate production in the early dry stage, the overall negative correlations to sulfate species are thus observed even after considering the later wet period, where $CO_2$ shows a negligible effect on $SO_2$ adsorption and subsequent oxidation to sulfate.

Alkaline (bi)carbonate salt particles behave unlike adsorbed/dissolved (bi)carbonate species regarding $SO_2$ uptake. The former favors $SO_2$ adsorption (Al-Hosney and Grassian, 2005) and subsequent oxidation by oxidants, e.g. $O_3$ (Li et al., 2006), and is expected to positively correlate to sulfate production. On the contrary, competitive adsorption between adsorbed $CO_2$ ($HCO_3^-/CO_3^{2-}$) and $SO_2$ reduces the sites over dust particles, likely showing the negative correlation to sulfate.

Considering that negative correlations between sulfate and (bi)carbonate ions are observed, we deduce that $CO_2$-derived anions are the dominant bi(carbonate) source observed in the collected sample. We have added relevant discussion into the current manuscript, allowing readers to follow.

"This is also supported by lab-based observations where $CO_2$-derived (bi)carbonate species are demonstrated to suppress sulfate production over two dominant mineral dust components aluminum oxide (Liu et al., 2020b) and silicon dioxide (Fig. S15 and supplement text 23). Alternatively, while $CO_2$-derived (bi)carbonate may slightly affect sulfate accumulation over PM with high water content in the dark scenario, fresh PM is usually dry when emitted into the atmosphere. Due to the competitive adsorption, the occurrence of suppression of $SO_2$ adsorption and sulfate formation is possible in the early emission stage before PM becomes wet, thus contributing to the overall negative correlation."

L493—To what extent do the authors think that gas-phase carbonate radicals contribute to overall atmospheric oxidative capacity? I am not entirely convinced of its production in these experiments (as the evidence is indirect).
Response to Q: Thanks for your question. You raise a really good suggestion and we would like to place our great effort on directly probing the gas-phase $CO_3^{\cdot-}$ coming from dust particle surfaces in the future work.

At present, direct observation of gas-phase $CO_3^{\cdot-}$ ejected from dust particles is not available for our lab experiment set-up. Therefore, indirect measurements using probe molecules of carbonate radicals were conducted to validate our assumption.

In the reaction system in which we designed to probe gas-phase $CO_3^{\cdot-}$; that is probe molecule (phosphate buffer solution pH = 7.0 + aniline) beneath $TiO_2$-coated film upon 310 nm UV irradiation in the presence of air flow or air+$CO_2$ flow (the intervening gap between probe molecules and $TiO_2$-coated film is less than 2 mm)), the formation of new active intermediates due to $CO_2$ is very likely

to account for accelerated degradation of aniline. While we can not provide a direct observation of $CO_3^{-}$ (g), at least in the current reaction system we are now considering, $CO_3^{-}$ (g) is the most plausible active intermediate to account for promoted aniline degradation.

Taken above, in combination with current knowledge of the formation scheme of $CO_3^{-}$, we thus speculate that gas-phase $CO_3^{-}$ ejected from dust particles contribute to the increased atmospheric oxidative capacity.

Similarly, we put this conclusive statement in a conservative tone to avoid disputed arguments coming from the community; that is "To be important, gas-phase carbonate radical ions are speculated to be formed the in the atmospherically relevant $CO_2$ concentration (400 ppm) when mineral dust particles are irradiated."

Reference:

Abou-Ghanem M, Oliynyk AO, Chen ZH, Matchett LC, McGrath DT, Katz MJ, et al.: Significant Variability in the Photocatalytic Activity of Natural Titanium-Containing Minerals: Implications for Understanding and Predicting Atmospheric Mineral Dust Photochemistry. Environ. Sci. Technol., 54,13509-13516, https://doi.org/10.1021/acs.est.0c05861, 2020.

Al-Hosney HA, Carlos-Cuellar S, Baltrusaitis J, Grassian VH: Heterogeneous uptake and reactivity of formic acid on calcium carbonate particles: a Knudsen cell reactor, FTIR and SEM study. Phys. Chem. Chem. Phys., 7,3587-3595, https://doi.org/10.1039/b510112c, 2005.

Al-Hosney HA, Grassian VH: Water, sulfur dioxide and nitric acid adsorption on calcium carbonate: A transmission and ATR-FTIR study. Phys. Chem. Chem. Phys., 7,1266-1276, https://doi.org/10.1039/b417872f, 2005.

Al-Salihi AM, Mohammed TH: The effect of dust storms on some meteorological elements over Baghdad , Iraq: Study Cases. IOSR Journal of Applied Physics, 7,Ver. II PP 01-07, https://doi.org/10.9790/4861-07220107, 2015.

Arimoto R, Zhang XY, Huebert BJ, Kang CH, Savoie DL, Prospero JM, et al.: Chemical composition of atmospheric aerosols from Zhenbeitai, China, and Gosan, South Korea, during ACE-Asia. J. Geophys. Res., 109,D19S04, https://doi.org/10.1029/2003jd004323, 2004.

Bao H, Yu S, Tong DQ: Massive volcanic $SO_2$ oxidation and sulphate aerosol deposition in Cenozoic North America. Nature, 465,909-912, https://doi.org/10.1038/nature09100, 2010.

Bougiatioti A, Zarmpas P, Koulouri E, Antoniou M, Theodosi C, Kouvarakis G, et al.: Organic, elemental and water-soluble organic carbon in size segregated aerosols, in the marine boundary layer of the Eastern Mediterranean. Atmos. Environ., 64,251-262, https://doi.org/10.1016/j.atmosenv.2012.09.071, 2013.

Buchalska M, Kobielusz M, Matuszek A, Pacia M, Wojtyla S, Macyk W: On Oxygen Activation at Rutile- and Anatase-$TiO_2$. Acs. Catal., 5,7424-7431, https://doi.org/10.1021/acscatal.5b01562, 2015.

Canelli E, Husain L: Determination of Total Particulate Sulfur at Whiteface Mountain, New-York, by Pyrolysis Microcoulometry. Atmos. Environ., 16,945-949, https://doi.org/10.1016/0004-6981(82)90179-2, 1982.

Chen HH, Nanayakkara CE, Grassian VH: Titanium Dioxide Photocatalysis in Atmospheric Chemistry. Chem. Rev., 112,5919-5948, https://doi.org/10.1021/cr3002092, 2012.

Christidis GE, Makri P, Perdikatsis V: Influence of grinding colour properties of on the structure and talc, bentonite and calcite white fillers. Clay Minerals, 39,163-175, https://doi.org/10.1180/0009855043920128, 2004.

Chrzan K: Conductivity of Aqueous Dust Solutions. Ieee. T. Dielect. El. In., 22,241-244, https://doi.org/10.1109/Tei.1987.298984, 1987.

Crowley JN, Ammann M, Cox RA, Hynes RG, Jenkin ME, Mellouki A, et al.: Evaluated kinetic and photochemical data for atmospheric chemistry: Volume V - heterogeneous reactions on solid substrates. Atmos. Chem. Phys., 10,9059-9223, https://doi.org/10.5194/acp-10-9059-2010, 2010.

Csavina J, Field J, Felix O, Corral-Avitia AY, Saez AE, Betterton EA: Effect of wind speed and relative humidity on atmospheric dust concentrations in semi-arid climates. Sci.Total. Envrion., 487,82-90, https://doi.org/10.1016/j.scitotenv.2014.03.138, 2014.

Cwiertny DM, Young MA, Grassian VH: Chemistry and photochemistry of mineral dust aerosol. Annu. Rev. Phys. Chem., 59,27-51, https://doi.org/10.1146/annurev.physchem.59.032607.093630, 2008.

Dixon RW, Aasen H: Measurement of hydroxymethanesulfonate in atmospheric aerosols. Atmos.

Environ., 33,2023-2029, https://doi.org/10.1016/S1352-2310(98)00416-6, 1999.

Dong XY, Fu JS, Huang K, Tong D, Zhuang GS: Model development of dust emission and heterogeneous chemistry within the Community Multiscale Air Quality modeling system and its application over East Asia. Atmos. Chem. Phys., 16,8157-8180, https://doi.org/10.5194/acp-16-8157-2016, 2016.

Dupart Y, King SM, Nekat B, Nowak A, Wiedensohler A, Herrmann H, et al.: Mineral dust photochemistry induces nucleation events in the presence of $SO_2$. Proc. Natl. Acad. Sci. USA, 109,20842-20847, https://doi.org/10.1073/pnas.1212297109, 2012.

Duran A, Monteagudo JM, Martin IS, Merino S, Chen X, Shi X: Solar photo-degradation of aniline with rGO/$TiO_2$ composites and persulfate. Sci.Total. Envrion., 697,134086-, https://doi.org/10.1016/j.scitotenv.2019.134086, 2019.

Engelbrecht JP, Mcdonald EV, Gillies JA, Quot RKM, Jay, quot, et al.: characterizing mineral dusts and other aerosols from the middle east-part 2: grab samples and re-suspensions. 21,327-336, https://doi.org/10.1080/08958370802464299, 2009.

Fang T, Guo H, Zeng L, Verma V, Nenes A, Weber RJ: Highly Acidic Ambient Particles, Soluble Metals, and Oxidative Potential: A Link between Sulfate and Aerosol Toxicity. Environ. Sci. Technol., 51,2611-2620, https://doi.org/10.1021/acs.est.6b06151, 2017.

Fang X, Liu Y, Kejian, Tao W, Yue D, Yiqing F, et al.: Atmospheric Nitrate Formation through Oxidation by carbonate radical. ACS Earth Space Chem., 5,1801–1811, https://doi.org/10.1021/acsearthspacechem.1c00169, 2021.

Feng T, Bei NF, Zhao SY, Wu JR, Li X, Zhang T, et al.: Wintertime nitrate formation during haze days in the Guanzhong basin, China: A case study. Environ. Pollut., 243,1057-1067, https://doi.org/10.1016/j.envpol.2018.09.069, 2018.

Gailhanou H, Blanc P, Rogez J, Mikaelian G, Kawaji H, Olives J, et al.: Thermodynamic properties of illite, smectite and beidellite by calorimetric methods: Enthalpies of formation, heat capacities, entropies and Gibbs free energies of formation. Geochim. Cosmochim. Ac., 89,279-301, https://doi.org/10.1016/j.gca.2012.04.048, 2012.

Gankanda A, Coddens EM, Zhang YP, Cwiertny DM, Grassian VH: Sulfate formation catalyzed by coal fly ash, mineral dust and iron(III) oxide: variable influence of temperature and light. Environmental Science-Processes & Impacts, 18,1484-1491, https://doi.org/10.1039/c6em00430j, 2016.

Ge WD, Liu JF, Yi K, Xu JY, Zhang YZ, Hu XR, et al.: Influence of atmospheric in-cloud aqueous-phase chemistry on the global simulation of $SO_2$ in CESM2. Atmos. Chem. Phys., 21,16093-16120, https://doi.org/10.5194/acp-21-16093-2021, 2021.

Goodman AL, Bernard ET, Grassian VH: Spectroscopic study of nitric acid and water adsorption on oxide particles: Enhanced nitric acid uptake kinetics in the presence of adsorbed water. J. Phys. Chem. A, 105,6443-6457, https://doi.org/10.1021/jp003722l, 2001.

Grassian VH: When Size Really Matters: Size-Dependent Properties and Surface Chemistry of Metal and Metal Oxide Nanoparticles in Gas and Liquid Phase Environments. J. Phys. Chem. C, 112,18303-18313, https://doi.org/10.1021/jp806073t, 2008.

Guo S, Hu M, Zamora ML, Peng JF, Shang DJ, Zheng J, et al.: Elucidating severe urban haze formation in China. Proc. Natl. Acad. Sci. USA, 111,17373-17378, https://doi.org/10.1073/pnas.1419604111, 2014.

Haghighatmamaghani A, Haghighat F, Lee CS: Performance of various commercial $TiO_2$ in photocatalytic degradation of a mixture of indoor air pollutants: Effect of photocatalyst and operating parameters. Science and Technology for the Built Environment, 25,600-614,

https://doi.org/10.1080/23744731.2018.1556051, 2019.

Hanisch F, Crowley JN: Ozone decomposition on Saharan dust: an experimental investigation. Atmos. Chem. Phys. Disscuss., 3,119-130, https://doi.org/10.5194/acp-3-119-2003, 2003.

He J, Xu HH, Balasubramanian R, Chan CY, Wang CJ: Comparison of $NO_2$ and $SO_2$ Measurements Using Different Passive Samplers in Tropical Environment. Aerosol Air Qual. Res., 14,355-363, https://doi.org/10.4209/aaqr.2013.02.0055, 2014.

He X, Pang SF, Ma JB, Zhang YH: Influence of relative humidity on heterogeneous reactions of $O_3$ and $O_3/SO_2$ with soot particles: Potential for environmental and health effects. Atmos. Environ., 165,198-206, https://doi.org/10.1016/j.atmosenv.2017.06.049, 2017.

Herrmann H, Ervens B, Jacobi HW, Wolke R, Nowacki P, Zellner R: CAPRAM2.3: A chemical aqueous phase radical mechanism for tropospheric chemistry. J. Atmos. Chem., 36,231-284, https://doi.org/10.1023/A:1006318622743, 2000.

Hodgson GW, Baker BL: Porphyrins in Meteorites - Metal Complexes in Orgueil, Murray, Cold Bokkeveld, and Mokoia Carbonaceous Chondrites. Geochim. Cosmochim. Ac., 33,943-&, https://doi.org/10.1016/0016-7037(69)90105-7, 1969.

Huang JP, Mabury SA: Steady-state concentrations of carbonate radicals in field waters. Environ. Toxicol. Chem., 19,2181-2188, https://doi.org/10.1002/etc.5620190906, 2000.

Huang L, An JY, Koo B, Yarwood G, Yan RS, Wang YJ, et al.: Sulfate formation during heavy winter haze events and the potential contribution from heterogeneous $SO_2 + NO_2$ reactions in the Yangtze River Delta region, China. Atmos. Chem. Phys., 19,14311-14328, https://doi.org/10.5194/acp-19-14311-2019, 2019.

Huang X, Song Y, Zhao C, Li MM, Zhu T, Zhang Q, et al.: Pathways of sulfate enhancement by natural and anthropogenic mineral aerosols in China. J. Geophys. Res., 119,14165-14179, 10.1002/2014jd022301, 2014.

Hung HM, Hoffmann MR: Oxidation of Gas-Phase $SO_2$ on the Surfaces of Acidic Microdroplets: Implications for Sulfate and Sulfate Radical Anion Formation in the Atmospheric Liquid Phase. Environ. Sci. Technol., 49,13768-13776, https://doi.org/10.1021/acs.est.5b01658, 2015.

Hung HM, Hsu MN, Hoffmann MR: Quantification of $SO_2$ oxidation on interfacial surfaces of acidic micro-droplets: Implication for ambient sulfate formation. Environ. Sci. Technol., 52,9079-9086, https://doi.org/10.1021/acs.est.8b01391, 2018.

Itahashi S, Yamaji K, Chatani S, Hayami H: Refinement of Modeled Aqueous-Phase Sulfate Production via the Fe- and Mn-Catalyzed Oxidation Pathway. Atmosphere, 9,132, https://doi.org/10.3390/atmos9040132, 2018.

Jaffe, Howard W. Crystal Chemistry and Refractivity New York: Dover Publications, Inc.: Mineola, 1996.

Ji YM, Wang HH, Li GY, An TC: Theoretical investigation on the role of mineral dust aerosol in atmospheric reaction: A case of the heterogeneous reaction of formaldehyde with NO2 onto SiO2 dust surface. Atmos. Environ., 103,207-214, https://doi.org/10.1016/j.atmosenv.2014.12.044, 2015.

Jin X, Fang M, Ji H, Nesterenko PN, Chen M: Migration of Organic Acids in Atmospheric Wet Deposition in Hangzhou Monitored by Ion Chromatography. Environ. Chem., 39,2287-2295, https://doi.org/10.7524/j.issn.0254-6108.2019053101, 2020.

Johnson SM, Pask JA, Moya JS: Influence of Impurities on High-Temperature Reactions of Kaolinite. J. Am. Ceram. Soc., 65,31-35, https://doi.org/10.1111/j.1151-2916.1982.tb09918.x, 1982.

Joshi N, Romanias MN, Riffault V, Thevenet F: Investigating water adsorption onto natural mineral dust particles: Linking DRIFTS experiments and BET theory. Aeolian Res., 27,35-45,

https://doi.org/10.1016/j.aeolia.2017.06.001, 2017.

Kay A, Gratzel M: Artificial Photosynthesis .1. Photosensitization of $TiO_2$ Solar-Cells with Chlorophyll Derivatives and Related Natural Porphyrins. Journal of Physical Chemistry, 97,6272-6277, https://doi.org/10.1021/j100125a029, 1993.

Kerminen VM, Hillamo R, Teinilä K, Pakkanen T, Allegrini I, Sparapani R: Ion balances of size-resolved tropospheric aerosol samples: implications for the acidity and atmospheric processing of aerosols. Atmos. Environ., 35,5255-5265, https://doi.org/10.1016/S1352-2310(01)00345-4, 2001.

Khlystov A, Stanier CO, Takahama S, Pandis SN: Water content of ambient aerosol during the Pittsburgh air quality study. J. Geophys. Res., 110,D07S10, https://doi.org/10.1029/2004jd004651, 2005.

Kilgour DB, Novak GA, Sauer JS, Moore AN, Dinasquet J, Amiri S, et al.: Marine gas-phase sulfur emissions during an induced phytoplankton bloom. Atmos. Chem. Phys., 22,1601-1613, https://doi.org/10.5194/acp-22-1601-2022, 2022.

Li BQ, Ma XY, Li QS, Chen WZ, Deng J, Li GX, et al.: Factor affecting the role of radicals contribution at different wavelengths, degradation pathways and toxicity during UV-LED/chlorine process. Chem. Eng. J., 392,124552, https://doi.org/10.1016/j.cej.2020.124552, 2020.

Li HJ, Zhu T, Zhao DF, Zhang ZF, Chen ZM: Kinetics and mechanisms of heterogeneous reaction of $NO_2$ on $CaCO_3$ surfaces under dry and wet conditions. Atmos. Chem. Phys., 10,463-474, https://doi.org/10.5194/acp-10-463-2010, 2010.

Li L, Chen ZM, Zhang YH, Zhu T, Li JL, Ding JJAC, et al.: Kinetics and mechanism of heterogeneous oxidation of sulfur dioxide by ozone on surface of calcium carbonate. 6,125-139, https://doi.org/10.5194/acp-6-2453-2006, 2006.

Li L, Chen ZM, Zhang YH, Zhu T, Li S, Li HJ, et al.: Heterogeneous oxidation of sulfur dioxide by ozone on the surface of sodium chloride and its mixtures with other components. J. Geophys. Res., 112,D18301, https://doi.org/10.1029/2006jd008207, 2007.

Li WJ, Shao LY, Shi ZB, Chen JM, Yang LX, Yuan Q, et al.: Mixing state and hygroscopicity of dust and haze particles before leaving Asian continent. J. Geophys. Res., 119,1044-1059, https://doi.org/10.1002/2013jd021003, 2014.

Li XR, Wang LL, Ji DS, Wen TX, Pan YP, Sun Y, et al.: Characterization of the size-segregated water-soluble inorganic ions in the Jing-Jin-Ji urban agglomeration: Spatial/temporal variability, size distribution and sources. Atmos. Environ., 77,250-259, https://doi.org/10.1016/j.atmosenv.2013.03.042, 2013.

Liu JR, Ning A, Liu L, Wang HX, Kurten T, Zhang XH: A pH dependent sulfate formation mechanism caused by hypochlorous acid in the marine atmosphere. Sci.Total. Envrion., 787,147551, https://doi.org/10.1016/j.scitotenv.2021.147551, 2021.

Liu TY, Abbatt JPD: Oxidation of sulfur dioxide by nitrogen dioxide accelerated at the interface of deliquesced aerosol particles. Nat. Chem., 13,1173-+, https://doi.org/10.1038/s41557-021-00777-0, 2021.

Liu TY, Clegg SL, Abbatt JPD: Fast oxidation of sulfur dioxide by hydrogen peroxide in deliquesced aerosol particles. Proc. Natl. Acad. Sci. USA, 117,1354-1359, https://doi.org/10.1073/pnas.1916401117, 2020a.

Liu XC, Tang WJ, Chen HN, Guo JM, Tripathee L, Huang J: Observational Study of Ground-Level Ozone in the Desert Atmosphere. B. Environ. Contam. Tox., 108,219-224, https://doi.org/10.1007/s00128-021-03444-9, 2022.

Liu Y, Wang T, Fang X, Deng Y, Cheng H, Fu H, et al.: Impact of greenhouse gas $CO_2$ on the

heterogeneous reaction of SO$_2$ on Alpha-Al$_2$O$_3$. Chinese Chem. Lett., 31,2712-2716, https://doi.org/10.1016/j.cclet.2020.04.037, 2020b.

Liu YQ, He XX, Duan XD, Fu YS, Dionysiou DD: Photochemical degradation of oxytetracycline: Influence of pH and role of carbonate radical. Chem. Eng. J., 276,113-121, https://doi.org/10.1016/j.cej.2015.04.048, 2015.

Liu YY, Wang T, Fang XZ, Deng Y, Cheng HY, Bacha AUR, et al.: Brown carbon: An underlying driving force for rapid atmospheric sulfate formation and haze event. Sci.Total. Envrion., 734,139415, https://doi.org/10.1016/j.scitotenv.2020.139415, 2020c.

Liu ZR, Xie YZ, Hu B, Wen TX, Xin JY, Li XR, et al.: Size-resolved aerosol water-soluble ions during the summer and winter seasons in Beijing: Formation mechanisms of secondary inorganic aerosols. Chemosphere, 183,119-131, https://doi.org/10.1016/j.chemosphere.2017.05.095, 2017.

Ma QX, Wang L, Chu BW, Ma JZ, He H: Contrary Role of H$_2$O and O$_2$ in the Kinetics of Heterogeneous Photochemical Reactions of SO$_2$ on TiO$_2$. J. Phys. Chem. A, 123,1311-1318, https://doi.org/10.1021/acs.jpca.8b11433, 2019.

Mayer KJ, Wang XF, Santander MV, Mitts BA, Sauer JS, Sultana CM, et al.: Secondary Marine Aerosol Plays a Dominant Role over Primary Sea Spray Aerosol in Cloud Formation. Acs Central Science, 6,2259-2266, https://doi.org/10.1021/acscentsci.0c00793, 2020.

Miller-Schulze JP, Shafer M, Schauer JJ, Heo J, Solomon PA, Lantz J, et al.: Seasonal contribution of mineral dust and other major components to particulate matter at two remote sites in Central Asia. Atmos. Environ., 119,11-20, https://doi.org/10.1016/j.atmosenv.2015.07.011, 2015.

Mogili PK, Kleiber PD, Young MA, Grassian VH: Heterogeneous uptake of ozone on reactive components of mineral dust aerosol: an environmental aerosol reaction chamber study. J. Phys. Chem. A, 110,13799-807, https://doi.org/10.1021/jp063620g, 2006.

Najafpour N, Afshin H, Firoozabadi B: Dust concentration over a semi-arid region: Parametric study and establishment of new empirical models. Atmos. Res., 243,104995, https://doi.org/10.1016/j.atmosres.2020.104995, 2020.

Nanayakkara CE, Larish WA, Grassian VH: Titanium Dioxide Nanoparticle Surface Reactivity with Atmospheric Gases, CO$_2$, SO$_2$, and NO$_2$: Roles of Surface Hydroxyl Groups and Adsorbed Water in the Formation and Stability of Adsorbed Products. J. Phys. Chem. C, 118,23011-23021, https://doi.org/10.1021/jp504402z, 2014.

Nie W, Wang T, Xue LK, Ding AJ, Wang XF, Gao XM, et al.: Asian dust storm observed at a rural mountain site in southern China: chemical evolution and heterogeneous photochemistry. Atmos. Chem. Phys., 12,11985-11995, https://doi.org/10.5194/acp-12-11985-2012, 2012.

Novak GA, Bertram TH: Reactive VOC Production from Photochemical and Heterogeneous Reactions Occurring at the Air-Ocean Interface. Accounts of Chemical Research, 53,1014-1023, https://doi.org/10.1021/acs.accounts.0c00095, 2020.

Peters SJ, Ewing GE: Water on Salt: An Infrared Study of Adsorbed H$_2$O on NaCl (100) under Ambient Conditions. J. Phys. Chem. B, 101,10880-10886, https://doi.org/10.1021/jp972810b, 1997.

Rubasinghege G, Elzey S, Baltrusaitis J, Jayaweera PM, Grassian VH: Reactions on Atmospheric Dust Particles: Surface Photochemistry and Size-Dependent Nanoscale Redox Chemistry. J. Phys. Chem. Lett., 1,1729-1737, https://doi.org/10.1021/jz100371d, 2010.

Samuni A, Goldstein S, Russo A, Mitchell JB, Krishna MC, Neta P: Kinetics and mechanism of hydroxyl radical and OH-adduct radical reactions with nitroxides and with their hydroxylamines. J. Am. Chem. Soc., 124,8719-8724, https://doi.org/10.1021/ja017587h, 2002.

Shang J, Li J, Zhu T: Heterogeneous reaction of $SO_2$ on $TiO_2$ particles. Sci. China Chem., 53,2637–2643, https://doi.org/10.1007/s11426-010-4160-3, 2010.

Sulzberger B, Canonica S, Egli T, Giger W, Klausen J, von Gunten U: Oxidative transformations of contaminants in natural and in technical systems. Chimia, 51,900-907, https://doi.org/10.1051/epjconf/20101105003, 1997.

Sun YL, Jiang Q, Wang ZF, Fu PQ, Li J, Yang T, et al.: Investigation of the sources and evolution processes of severe haze pollution in Beijing in January 2013. J. Geophys. Res., 119,4380-4398, https://doi.org/10.1002/2014jd021641, 2014.

Ta WQ, Xiao Z, Qu JJ, Yang GS, Wang T: Characteristics of dust particles from the desert/Gobi area of northwestern China during dust-storm periods. Environmental Geology, 43,667-679, https://doi.org/10.1007/s00254-002-0673-1, 2003.

Tang MJ, Cziczo DJ, Grassian VH: Interactions of Water with Mineral Dust Aerosol: Water Adsorption, Hygroscopicity, Cloud Condensation, and Ice Nucleation. Chem. Rev., 116,4205-4259, https://doi.org/10.1021/acs.chemrev.5b00529, 2016a.

Tang Y, Li LL, Wang SS, Cheng QT, Zhang J: Tricomponent coupling biodiesel production catalyzed by surface modified calcium oxide. Environ. Prog. Sustain., 35,257-262, https://doi.org/10.1002/ep.12194, 2016b.

Usher CR, Al-Hosney H, Carlos-Cuellar S, Grassian VH: A laboratory study of the heterogeneous uptake and oxidation of sulfur dioxide on mineral dust particles. J. Geophys. Res., 107,4713, https://doi.org/10.1029/2002jd002051, 2002.

Usher CR, Michel AE, Grassian VH: Reactions on mineral dust. Chem. Rev., 103,4883-4939, https://doi.org/10.1021/cr020657y, 2003.

Wang JF, Li JY, Ye JH, Zhao J, Wu YZ, Hu JL, et al.: Fast sulfate formation from oxidation of $SO_2$ by $NO_2$ and HONO observed in Beijing haze. Nat. Commun., 11,2844, https://doi.org/10.1038/s41467-020-16683-x, 2020.

Wang T, Liu YY, Deng Y, Cheng HY, Fang XZ, Zhang LW: Heterogeneous Formation of Sulfur Species on Manganese Oxides: Effects of Particle Type and Moisture Condition. J. Phys. Chem. A, 124,7300-7312, https://doi.org/10.1021/acs.jpca.0c04483, 2020a.

Wang T, Liu YY, Deng Y, Cheng HY, Yang Y, Li KJ, et al.: Irradiation intensity dependent heterogeneous formation of sulfate and dissolution of ZnO nanoparticles. Environ. Sci. Nano., 7,327-338, https://doi.org/10.1039/c9en01148j, 2020b.

Wang T, Liu YY, Deng Y, Fu HB, Zhang LW, Chen JM: The influence of temperature on the heterogeneous uptake of $SO_2$ on hematite particles. Sci.Total. Envrion., 644,1493-1502, https://doi.org/10.1016/j.scitotenv.2018.07.046, 2018.

Wang XF, Wang WX, Yang LX, Gao XM, Nie W, Yu YC, et al.: The secondary formation of inorganic aerosols in the droplet mode through heterogeneous aqueous reactions under haze conditions. Atmos. Environ., 63,68-76, https://doi.org/10.1016/j.atmosenv.2012.09.029, 2012.

Wang XM, Huang X, Zuo CY, Hu HY: Kinetics of quinoline degradation by $O_3$/UV in aqueous phase. Chemosphere, 55,733-741, https://doi.org/10.1016/j.chemosphere.2003.11.019, 2004.

Wang Y, Zhuang GS, Tang AH, Yuan H, Sun YL, Chen SA, et al.: The ion chemistry and the source of PM2.5 aerosol in Beijing. Atmos. Environ., 39,3771-3784, https://doi.org/10.1016/j.atmosenv.2005.03.013, 2005.

Watanabe K, Yang L, Nakamura S, Otani T, Mori K: Volcanic Impact of Nishinoshima Eruptions in Summer 2020 on the Atmosphere over Central Japan: Results from Airborne Measurements of

Aerosol and Trace Gases. Sola, 17,109-112, https://doi.org/10.2151/sola.2021-017, 2020.

Weaver CEJC, Minerals C: The nature of $TiO_2$ in kaolinite. Clay Clay Miner., 24,215–218, https://doi.org/10.1346/CCMN.1976.0240501, 1976.

Whiteside M, Herndon J: Coal Fly Ash Aerosol: Risk Factor for Lung Cancer. Journal of Advances in Medicine and Medical Research, 25,1-10, https://doi.org/10.9734/jammr/2018/39758, 2018.

Wojnarovits L, Toth T, Takacs E: Rate constants of carbonate radical anion reactions with molecules of environmental interest in aqueous solution: A review. Sci.Total. Envrion., 717,137219, https://doi.org/10.1016/j.scitotenv.2020.137219, 2020.

Wu LY, Tong SR, Wang WG, Ge MF: Effects of temperature on the heterogeneous oxidation of sulfur dioxide by ozone on calcium carbonate. Atmos. Chem. Phys., 11,6593-6605, https://doi.org/10.5194/acp-11-6593-2011, 2011.

Wu Q, Tang X, Kong L, Dao X, Lu MM, Liu ZR, et al.: Evaluation and Bias Correction of the Secondary Inorganic Aerosol Modeling over North China Plain in Autumn and Winter. Atmosphere, 12,578, https://doi.org/10.3390/atmos12050578, 2021.

Xu W, Qian L, Jing S, Jia L, Xiang F, Tong ZJJoES: Heterogeneous oxidation of $SO_2$ by $O_3$ aged black carbon and its dithiothreitol oxidative potential. J. Environ. Sci. China, 36,56-62, https://doi.org/10.1016/j.jes.2015.02.014, 2015.

Yu T, Zhao D, Song X, Zhu T: $NO_2$-initiated multiphase oxidation of $SO_2$ by $O_2$ on $CaCO_3$ particles. Atmos. Chem. Phys., 18,6679-6689, https://doi.org/10.5194/acp-18-6679-2018, 2018a.

Yu T, Zhao DF, Song XJ, Zhu T: NO2-initiated multiphase oxidation of SO2 by O-2 on CaCO3 particles. Atmos Chem Phys, 18,6679-6689, 10.5194/acp-18-6679-2018, 2018b.

Yu ZC, Jang M: Simulation of heterogeneous photooxidation of $SO_2$ and $NO_X$ in the presence of Gobi Desert dust particles under ambient sunlight. Atmos. Chem. Phys., 18,14609-14622, https://doi.org/10.5194/acp-18-14609-2018, 2018.

Yu ZC, Jang M, Park J: Modeling atmospheric mineral aerosol chemistry to predict heterogeneous photooxidation of $SO_2$. Atmos. Chem. Phys., 17,10001-10017, https://doi.org/10.5194/acp-17-10001-2017, 2017.

Yu ZC, Jang MS, Kim S, Bae C, Koo BY, Beardsley R, et al.: Simulating the Impact of Long-Range-Transported Asian Mineral Dust on the Formation of Sulfate and Nitrate during the KORUS-AQ Campaign. ACS Earth Space Chem., 4,1039-1049, https://doi.org/10.1021/acsearthspacechem.0c00074, 2020.

Zheng B, Zhang Q, Zhang Y, He KB, Wang K, Zheng GJ, et al.: Heterogeneous chemistry: a mechanism missing in current models to explain secondary inorganic aerosol formation during the January 2013 haze episode in north China. Atmos. Chem. Phys., 15,2031-2049, https://doi.org/10.5194/acp-15-2031-2015, 2015.